# Distributed snow and rock temperature modelling in steep rock walls using Alpine3D

Anna Haberkorn[1,2], Nander Wever[1,3], Martin Hoelzle[2], Marcia Phillips[1], Robert Kenner[1], Mathias Bavay[1], Michael Lehning[1,3]

[1]WSL Institute for Snow and Avalanche Research SLF, CH-7260 Davos, Switzerland

[2]Department of Geosciences, Unit of Geography, University of Fribourg, CH-1700 Fribourg, Switzerland

[3]CRYOS, School of Architecture, Civil and Environmental Engineering, EPFL, CH-1015 Lausanne, Switzerland

*Correspondence to*: Anna Haberkorn (haberkorn@slf.ch)

**Abstract**. In this study we modelled the influence of the spatially and temporally heterogeneous snow cover on the surface energy balance and thus on rock temperatures in two rugged, steep rock walls on the Gemsstock ridge, central Swiss Alps. The heterogeneous snow depth distribution in the rock walls was introduced to the distributed, process based energy balance model Alpine3D with a precipitation scaling method based on snow depth data measured by terrestrial laser scanning. The influence of the snow cover on rock temperatures was investigated by comparing a snow-covered model scenario (precipitation input provided by precipitation scaling) with a snow-free (zero precipitation input) one. Model uncertainties are discussed and evaluated at both the point- and spatial-scale against 22 near-surface rock temperature measurements and high-resolution snow depth data from winter terrestrial laser scans.

In the rough rock walls, the heterogeneously distributed snow cover was moderately well reproduced by Alpine3D with mean absolute errors ranging between 0.47 and 0.77 m. However, snow cover duration was reproduced well and consequently near-surface rock temperatures were modelled convincingly. Uncertainties in rock temperature modelling were found to be around 1.6 °C. Errors in snow cover modelling and consequently in rock temperature simulations are explained by inadequate snow settlement due to linear precipitation scaling, missing lateral heat fluxes in the rock, as well as by errors caused by interpolation of shortwave radiation, wind and air temperature into the rock walls.

Mean annual near-surface rock temperature increases were both measured and modelled in the steep rock walls as a consequence of a thick, long lasting snow cover. Rock temperatures were 1.3-2.5 °C higher in the shaded and sunny rock walls, while comparing snow-covered to the snow-free simulations. This helps to assess the potential error made in ground temperature modelling when neglecting snow in steep bedrock.

**Keywords**: snow depth distribution, Alpine3D, distributed energy balance modelling, impact of snow on rock temperatures, steep rock walls

## 1. Introduction

In the European Alps, numerous rock fall events were observed in permafrost rock faces during the last decades (e.g. Fischer et al., 2012; Gruber et al., 2004b; Phillips et al., 2016b; Ravanel et al., 2010, 2013). Rock fall can be attributed to various triggering factors (Fischer et al., 2012; Krautblatter et al., 2013), including a fast reaction of rock faces to climate change expressed in rapid active layer thickening and permafrost degradation (e.g. Allen and Huggel, 2013; Deline et al., 2015; Gruber and Haeberli, 2007; Ravanel and Deline, 2011; Sass and Oberlechner, 2012). Rock wall instability is a risk to the safety of local communities and infrastructure in the densely populated Alps (Bommer et al., 2010). Measuring rock wall temperatures (e.g. Gruber et al., 2004a;

Haberkorn et al., 2015a, Hasler et al., 2011; Magnin et al., 2015, PERMOS, 2013) and in a further step modelling the spatial permafrost distribution in steep rock walls is therefore of great importance.

Numerical model studies simulating rock temperatures of idealized rock walls have been realised e.g. by Gruber et al. (2004a), Noetzli et al. (2007) and Noetzli and Gruber (2009). These studies assumed a lack of snow in steep rock exceeding slope angles of 50°, which is based on the general assumption that wind and gravitational

transport (avalanching or sloughing) remove the snow from steep rock exceeding 50°-60° (e.g. Blöschl and Kirnbauer, 1992; Gruber Schmid and Sardemann, 2003; Winstral et al., 2002). They therefore suggested that air temperature and solar radiation are sufficient to model rock surface temperatures in near-vertical, compact, homogeneous rock walls. Rock walls are, however, often variable inclined, heterogeneous, fractured and thus partly snow-covered (Haberkorn et al., 2015a; Hasler et al., 2011; Sommer et al., 2015). Beside three-

dimensional (3d) subsurface heat flow and transient changes in steep bedrock thermal modelling (Noetzli et al., 2007; Noetzli and Gruber, 2009), the strongly variable spatial and temporal rock surface boundary conditions therefore also need to be taken into account. The spatially variable snow cover is one of these driving factors.

The influence of the snow cover on the rock thermal regime has recently been studied in steep bedrock (Haberkorn et al., 2015a,b; Hasler et al., 2011; Magnin et. al., 2015). The highly variable spatial and temporal

distribution of the snow cover strongly influences the ground thermal regime of steep rock faces (Haberkorn et al., 2015a,b; Magnin et al., 2015) due to the high surface albedo and low thermal conductivity of the snow cover, as well as energy consumption during snow melt (Bernhard et al., 1998; Keller and Gubler, 1993; Zhang, 2005). In gently inclined, blocky terrain, effective ground surface insulation from cold atmospheric conditions were observed and modelled for snow depths exceeding 0.6 to 0.8 m (Hanson and Hoelzle, 2004; Keller and Gubler,

1993; Luetschg et al., 2008). In contrast, Haberkorn et al. (2015a) found that snow depths exceeding 0.2 m were enough to have an insulating effect on steep, bare bedrock. Such amounts are likely to accumulate in steep, high rock walls with a certain degree of surface roughness. Indeed, a warming effect of the snow cover on mean annual ground surface temperature (MAGST) was observed by Haberkorn et al. (2015a) and Magnin et al. (2015) in shaded rock walls, whilst in moderately inclined (45°-70°) sun-exposed rock walls Hasler et al. (2011)

suggest a reduction of MAGST of up to 3 °C compared to estimates in near-vertical, compact rock, due to snow persistence during the months with most intense radiation. Those observations emphasize the need to account for the strongly varying snow cover in thermal modelling of steep rock walls. Myhra et al. (2015) and Pogliotti (2011) simulated the potential thermal effect of snow on steep bedrock temperatures, while changing snow depths arbitrarily in one-dimensional (1d) (Pogliotti, 2011) and two-dimensional (Myhra et al., 2015) numerical

model runs. Both authors provided evidence of a considerable influence of snow on the rock thermal regime, but could not verify their results with measurements due to a lack of snow depth observations in steep rock walls. Nevertheless, the relative influence of snow on the rock thermal regime was evaluated by Pogliotti (2011) by comparing point simulations without snow to those with virtual snow.

Recent studies based on terrestrial laser scanning (TLS) not only confirmed that snow accumulates in steep,

rough rock walls with rock ledges (Haberkorn et al., 2015a; Sommer et al., 2015; Wirz et al., 2011), but also provided accurate snow depth distribution measurements for both rock temperature modelling and model verification. This is of great importance, since an accurately modelled snow cover evolution and its spatial patterns are crucial to correctly model the ground thermal regime (Fiddes et al., 2015; Hoelzle et al., 2001; Stocker-Mittaz et al., 2002) and assess contrasting influences of a heterogeneous snow cover on the ground

thermal regime.

To capture the strong spatial variability of the local surface energy balance and consequently of the ground thermal regime in moderately inclined terrain (Gubler et al., 2011; Riseborough et al., 2008), as well as in steep, rough rock walls (Haberkorn et al., 2015a; Hasler et al., 2011) it is necessary to account for the complex micro-topography and its influence on local shading effects, lateral heat fluxes at the rock surface caused by pronounced temperature gradients, small-scale snow distribution patterns and rock temperatures. The 1d modelling approach used by Haberkorn et al. (2015b) to investigate the influence of the snow cover on the rock thermal regime is therefore not sufficient, although the ability of the 1d SNOWPACK model (Lehning et al., 2002a,b; Luetschg et al., 2003; Wever et al., 2015) to simulate the effect of a snow cover on rock temperatures could clearly be demonstrated. High-resolution and spatially distributed physics-based simulations of land surface processes are needed.

We therefore present a spatially distributed model study of the influence of the snow cover on the surface energy balance and consequently on near-surface rock temperatures (NSRT) in steep north-west and south-east oriented rock walls using the physics-based 3d atmospheric and surface process model Alpine3D (Lehning et al., 2006). The distribution of the spatially and temporally heterogeneous snow cover in the steep terrain (up to 85°) was provided to the model using a precipitation scaling approach. This was based on a combination of snow depth measurements from the on-site flat field automatic weather station (AWS) and high-resolution (0.2 m) snow depth distribution data obtained using TLS. The challenge of integrating representative precipitation input (e.g. Imhof et al., 2000; Fiddes et al., 2015; Stocker-Mittaz et al., 2002) in the rock walls and its redistribution by wind (Mott and Lehning, 2010), as well as gravitational transport (Bernhardt and Schulz, 2010; Gruber, 2007) was thus accounted for. Model performance for simulating snow depth distribution and consequently the influence on rock temperatures was tested against a dense network of validation measurements of snow depth and NSRTs at both the point- and the spatial-scale. After quantifying model uncertainties, a sensitivity study was performed in order to assess the effects of the snow cover on the rock thermal regime. High-resolution (0.2 m) simulations were carried out, either providing snow cover distribution to the model (by precipitation scaling) or fully neglecting the presence of a snow cover in the rock walls. Thus the potential error induced by neglecting the snow cover in steep rock face thermal modelling for slope angles >50° can be estimated. This is necessary, since it has in general been assumed that wind and gravitational transport remove the snow from steep rock in slopes >50–60° (e.g. Blöschl and Kirnbauer, 1992; Gruber Schmid and Sardemann, 2003; Winstral et al., 2002) and rock temperatures were often modelled without snow for idealized rock walls >50° (e.g. Gruber et al., 2004a; Noetzli and Gruber, 2009; Noetzli et al. 2007).

## 2. Study site

The Gemsstock mountain ridge (46° 36' 7.74" N; 8° 36' 41.98" E; 2961 m a.s.l.) is located on the main divide of the Western Alps, central Switzerland (Fig. 1). Precipitation at Gemsstock is affected by both northerly and southerly airflows, resulting in enhanced orographic precipitation (Haberkorn et al. 2015a). The rocky ridge consists of Gotthard paragneiss and granodiorite, with veins of quartz. The site is at the lower fringe of mountain permafrost. Permafrost distribution is patchy in the north-west facing rock wall, whereas there is no permafrost in the south-east facing wall of the ridge (PERMOS, 2013).

This study focuses on a specific area on the north-west and south-east facing rocky flanks of the ridge, which for simplicity are henceforth referred to as the N and S slopes. The 40 m high slopes (2890–2930 m a.s.l.) are 40° to 70° steep, with vertical to overhanging (>90°) sections (Fig. 1a). The N facing scarp slope is intersected by a

series of parallel joints dipping south-eastwards at 70° (Phillips et al., 2016a). These joints form 0.3 to 3 m wide horizontal ledges within the N facing rock wall and alternate with steep to vertical parts. In contrast, the S facing dip slope has a rather smooth rock surface. We investigate the 2 year study period between 1 September 2012 and 31 August 2014.

## 3. Methods

Applying the Alpine3D model chain for spatially distributed steep rock wall thermal modelling requires various input data and computing steps. In Fig. 2 a brief synopsis of the methods used in this study are shown. Based on Fig. 2 first the distributed numerical model used in this study is introduced. Then the data and model settings required to drive the model are specified, followed by a description of the computation of the precipitation input, which is essential in order to introduce varying snow depths to the extremely steep terrain. Finally the validation data-sets used to evaluate the model performance are introduced.

### 3. 1 Distributed energy balance modelling

#### 3.1.1   The Alpine3D model

The fully distributed physics-based surface process model Alpine3D (Lehning et al., 2006, 2008; Kuonen et al., 2010) was used to simulate the influence of the heterogeneously distributed snow cover on the thermal regime of the Gemsstock rock ridge. To do this it is essential to model the surface energy balance as shown in Eq. 1, which is determined by the exchange of energy between the atmosphere and the surface. The energy flux $Q_{snow}$ available for warming and melting or cooling and freezing of the snowpack or the ground is calculated in Alpine3D as the sum of all energy balance components [W m$^{-2}$] at the respective surface (Armstrong and Brun, 2008):

$$Q_{snow} = Q_{net} + Q_{sensible} + Q_{latent} + Q_{rain} + Q_{ground} \quad (1)$$

Where $Q_{net}$ is the sum of the net fluxes of short- and longwave radiation, $Q_{sensible}$ and $Q_{latent}$ are the turbulent fluxes of sensible and latent heat through the atmosphere, $Q_{rain}$ is the rain energy flux and $Q_{ground}$ is the 1d conduction of heat into the ground. In Alpine3D energy fluxes are considered positive when directed towards the snowpack surface (energy gain).

Meteorological data, a digital elevation model (DEM) and a land-use model are required to run Alpine3D (Fig. 2). In the setup used here Alpine3D consists of a 3d radiation model, which is based on the view factor approach to calculate short- and longwave radiation in complex terrain, including shortwave scattering and longwave emission from the terrain (Helbig et al., 2009). The 3d atmospheric processes are coupled to the 1d energy balance model SNOWPACK (Wever et al., 2014). The latter is based on the assumption that there is no lateral exchange in these media. SNOWPACK simulates the temporal evolution of the vertical transport of mass and energy, as well as phase-change processes for a variety of layers within the seasonal snowpack and in the ground for each single grid cell (Luetschg et al., 2003, 2008; Wever et al., 2015). A bulk Monin-Obukhov formulation is used to parameterize the latent and sensible heat fluxes at the surface. The water flow in the snow and rock is solved using a simple bucket type approach, which is suitable for daily and seasonal time-scales (Wever et al., 2014).

The 3d snow drift module (Lehning et al., 2008; Mott and Lehning, 2010) was not included in the simulations, although snow redistribution due to wind was observed at Gemsstock (Haberkorn et al., 2015a), because there is currently no model that convincingly reproduces 3d wind fields over extremely steep, heterogeneous rock walls. In addition the mass conserving computation of gravitational transport and deposition of snow (Bernhardt and Schulz, 2010; Gruber, 2007) is not included in simulations, although sloughing and avalanching were observed in the field (Haberkorn et al., 2015a) and have been suggested as the main process involved in the redistribution of snow in steep rock walls by Sommer et al. (2015). To account for the effects of snow redistribution on the snow depth distribution, we used measured snow depth data from a TLS campaign to scale precipitation grids (Sect. 3.1.3).

### 3.1.2 Model setup

The model was driven by meteorological data measured by the on-site AWS Gemsstock (Fig. 1a, 2869 m a.s.l.). Air temperature, relative humidity, wind speed and direction, as well as incoming short- and longwave radiation data were pre-processed, as well as spatially interpolated and parameterized with the MeteoIO library (Bavay and Egger, 2014). Precipitation was provided to the model as described in Sect. 3.1.3. Gaps in meteorological data were corrected according to Haberkorn et al. (2015b).

The DEM is derived from high-resolution TLS, carried out at Gemsstock in the snow-free N and S facing rock walls in summer using a RIEGL *VZ6000* scanner at a grid resolution of 0.2 m and with a domain size of 4460 m$^2$ (Figs. 1a, b). Based on the DEM, the land-use classification was divided into two groups with varying rock properties, depending on whether the grid cells were N or S facing. The rock is simulated to 20 m depth, divided into 24 layers of varying thickness ranging from 0.02 m at the surface to 4 m at the bottom of the substrate. For each classification, the rock layers were initialized with different layer temperatures based on borehole rock temperatures measured on-site (Fig. 1c, PERMOS 2013). The rock was assumed to be 99 % solid with 1 % pore space containing ice (N facing grid cells) or water (S facing grid cells) to account for near-surface fracture space to a depth of 0.5 m. Unfractured rock with a solid content of 100 % was assumed between 0.5 and 20 m depth. The physical properties of the granodiorite bedrock were based on Cermák and Rybach (1982): with a rock density of 2600 kg m$^{-3}$, a specific heat capacity of 1000 J kg$^{-1}$K$^{-1}$, a thermal conductivity of 2.8 W m$^{-1}$K$^{-1}$ (S facing grid cells) respectively 1.9 W m$^{-1}$K$^{-1}$ (N facing grid cells), as discussed in Haberkorn et al. (2015b). The rock albedo is assumed to be 0.15 and an aerodynamic roughness length of 0.002 m over snow is used for simulations. Although the geothermal heat flux is most likely negligible in the narrow, steep and complex Gemsstock ridge due to strong topographic (Kohl, 1999) and 3d thermal effects (Noetzli et al., 2007), a constant upward ground heat flux had to be applied. $Q_{ground}$ is assumed to be 0.001 W m$^{-2}$ at 20 m depth to ensure a marginal impact of the lower boundary condition on the analysed rock thermal regime close to the surface.

All simulations were run in parallel mode on the same computer cluster as a 32 core process, requiring around 15 days for a two-year simulation. Simulations were also performed for coarser resolutions (1 m, 5 m) to analyse the loss of model accuracy for lower computational costs.

### 3.1.3 Precipitation input for Alpine3D

*Terrestrial laser scanning (TLS)*

Snow depths acquired from TLS were used as input data for the precipitation scaling approach. Snow depth distribution was measured at different times in the winters 2012-2013 and 2013-2014 using a RIEGL *VZ6000*

long-range laser scanner. A total of 4 high-resolution scans were carried out, i.e. two per winter. The high spatial and temporal variability of snow depth distribution in the rock walls were determined by comparing the data to that obtained in snow-free summer scans of the rock walls. The shortest distance from each terrain point to the point cloud at the snow surface was calculated with a point resolution of 0.2 m (Haberkorn et al. 2015a,b). The snow depth determined perpendicular to the surface was both more representative regarding the impact on ground temperatures (Haberkorn et al., 2015b) and more accurate than conventional vertical snow depths in extremely steep terrain (Sommer et al., 2015). Snow depth gaps in the laser scans result from blind areas behind ridges or rocky outcrops. The measurement error made using TLS for snow depth measurements was found to be ±0.08 m (Haberkorn et al., 2015b) and is therefore similar to other observations in steep rock (Sommer et al., 2015).

*Precipitation scaling*

To model the snow cover in steep rock walls the high-resolution spatially explicit snow depth distribution data provided by TLS were used. A precipitation scaling algorithm was used to drive the Alpine3D model, which only uses precipitation as input data. As this was not available for Gemsstock, precipitation was first calculated from the snow depth measured at the on-site AWS using a stand-alone SNOWPACK simulation. By using the snow depth driven mode of the SNOWPACK model, the snow depth measurements were used to determine the timing and amount of snowfall by interpreting increases in snow depth as fresh snowfall. According to Lehning et al. (1999) and Wever et al. (2015), SNOWPACK converts snowfall to precipitation while calculating both snow settlement and snow density based on a statistical model. To complete the resulting precipitation series, summer liquid precipitation was used from the nearby MeteoSwiss AWS Gütsch (2287 m a.s.l., 6 km north of Gemsstock; Haberkorn et al. 2015b).

Secondly, for each grid cell, scaling factors were calculated based on the ratio between measured snow depth at the AWS and the snow depth of each grid cell measured by TLS at the date of the TLS campaign. These scaling factors were then used to scale the two-year precipitation time series for each grid cell of the DEM. We refer to this method as precipitation scaling, which provides grids of spatially distributed precipitation amounts for Alpine3D input. Data gaps in the TLS lead to data gaps in the precipitation scaling grid, resulting in erroneously modelled snow depths and rock temperatures at these locations. For the analysis of the Alpine3D grid output those grid cells have not been used.

Thirdly, model runs were carried out using scaled precipitation of each of the four TLS campaigns. The modelled snow depth and NSRT data coincided best with validation data when using scaled precipitation from snow depth data based on the TLS data obtained on 19 December 2012. Henceforth, the modelled results analysed and discussed here are only based on this TLS data. The use of an early winter TLS is preferred, since the early winter snow depth distribution best represents winter snowfall events. TLS data obtained in spring already contain ablation processes. Fig. 3 provides justification for the choice of the TLS used. Here the distribution of the ratio modelled to measured snow depth is shown for the 4 TLS available. The TLS data measured on 19 December 2012 is centred around 1, as well as the snow depth curves on the dates of two (7 June 2013, 28 January 2014) of the other three TLS campaigns. Those TLS show satisfactory agreement between modelled and measured snow depths. In contrast, simulations using the TLS on 11 December 2013 overestimate snow depths and have a wider spread compared to the ratio of the other scans. This may be due to snow depth distribution differences due to varying wind conditions in the rock walls. In early winter 2013-2014

a large proportion of snowfall events occurred with southerly winds, whereas in general snowfalls were accompanied by northwesterly flows. A quantitative analysis of the precipitation scaling approach is currently being evaluated (Voegeli et al., submitted). The use of only one TLS is additionally justified by annually recurring snow depth distribution patterns caused by the micro-topography, which have been observed in steep rock walls by Haberkorn et al. (2015a), Sommer et al. (2015) and Wirz et al. (2011).

## 3.2 Sensitivity study

A sensitivity study is performed in order to assess the bias made while neglecting snow in thermal modelling of steep rock walls, which has often been done for ideal, compact rock walls with slope angles >50° (e.g. Fiddes et al., 2015; Gruber et al., 2004a; Noetzli and Gruber, 2009; Noetzli et al. 2007). The sensitivity study comprises a rock temperature comparison between a model run with snow (precipitation input from precipitation scaling) and a model run without snow. For the model run without snow, precipitation input was forced to be zero. Alpine3D simulations were thus carried out for two contrasting scenarios in the rock walls (Fig. 2): one accounting for snow accumulation (henceforth referred to as 'snow-covered' scenario) and one neglecting snow (henceforth referred to as 'snow-free' scenario).

## 3. 3 Model validation

Uncertainties in modelling the snow depth distribution and the near-surface rock thermal regime in steep rock walls were assessed statistically, using the mean bias error (MBE), the mean absolute error (MAE) and the coefficient of determination ($r^2$). An error calculation was performed between observations (snow depth, NSRT) and model predictions at the corresponding grid cells for both the snow-covered and the snow-free scenarios for the years 2012-2013 (1 September 2012 – 31 August 2013) and 2013-2014 (1 September 2013 – 31 August 2014). The validation data sets (Fig. 2) will subsequently be explained.

*Near-surface rock temperature (NSRT) data*

The spatially variable thermal regime of the rock slopes was studied using a two-year time series of near-surface rock temperatures. NSRTs were measured in 0.1 m deep boreholes using Maxim iButtons® DS1922L (Maxim Integrated, 2013) temperature loggers. After calibration in an ice-water mixture, instrument accuracy was ±0.25 °C at 0 °C (Haberkorn et al., 2015b). 30 of these temperature loggers were distributed in a linear layout over the N and S facing rock walls (Fig. 1) with a vertical spacing of approximately 3 m.

A detailed statistical point-to-point analysis between modelled and measured NSRTs has been performed at 22 of 30 NSRT locations with a temporal resolution of two hours. 11 of these locations are N facing and 11 are S facing (Appendix: Table 1A). Data from 8 locations were disregarded due to data gaps in the TLS, as discussed in Section 3.1.2. All 22 points were used to evaluate the spatial model performance for each individual rock wall (Section 4.4). Therefore all measured or modelled NSRT data were averaged within the slopes depending whether the grid cells are N or S facing. In addition to the spatial analysis, an absolute point analysis between measured and modelled NSRT evolutions has been carried out for 4 loggers (Section 4.3). These 4 NSRT loggers were chosen in order to represent snow-rich and snow-free locations and thus contrasting NSRT conditions in the N and S facing rock walls. Logger N3 is located in a vertical sector near the top of the N facing rock wall, whereas logger N7 is located in vertical rock 12 m lower at the foot of this rock wall sector, 0.1 m above a ledge. On the S side of the ridge, logger R2 is in 58° steep rock 15 m above a ledge, whereas logger S9

is located in 70° steep terrain close to the gently inclined foot of a rock outcrop on the S facing rock wall (Table 1). Pronounced daily NSRT amplitudes indicate that N3 and R2 were generally snow-free (Figs. 7d, e). Although logger N7 and S9 are located in steep rock, wide ledges below allow the accumulation of a thick snow cover in winter, causing strong NSRT damping during the snow-covered period, as well as a zero curtain in spring (Figs. 7b, c).

*Snow depth data*

The rock thermal regime strongly depends on the timing, depth and duration of the snow cover. An accurately modelled snowpack is essential for the correct modelling of the rock thermal regime. The modelled snowpack was therefore validated against measured snow depth data from three independent TLS campaigns, which were not used for precipitation scaling. This was done at the rock wall-scale for all grid cells of the entire N and the S facing slopes on the date of the three TLS campaigns, as well as at the point-scale for grid cells corresponding to the 22 NSRT validation measurement locations. As for the 4 NSRT loggers' data presented in detail, the modelled 2-year snow depth evolution is only presented for the same 4 grid cells.

## 4 Results

In this section only measured and modelled results are presented, while model uncertainties will be discussed in Section 5. First the measured and modelled snow cover accumulating in the rock walls is described at both the spatial- and the point-scale (4 selected locations). The accumulation of snow changes the surface energy balance of the rock walls, which is discussed in Section 4.2, where the surface energy balance is presented for both the virtually snow-free and the snow-covered scenario at two NSRT locations accumulating snow. Changes in the surface energy balance are mirrored in the rock temperatures. The rock thermal regime close to the surface is firstly presented at the 4 selected NSRT locations (Section 4.3), followed by the spatial analysis (all 22 NSRT locations) of measurements and model results of both the snow-covered and the snow-free scenario (Section 4.4). Finally the accuracy of model results for coarser resolutions (1 m, 5 m) is evaluated in Section 4.5. Mean annual near-surface rock temperature (MANSRT), $r^2$, MAE and MBE are always given for the study years 2012-2013/2013-2014, separated by a slash (e.g. MANSRT for 2012-2013/MANSRT for 2013-2014).

### 4.1 Spatial snow cover variability

### 4.1.1 Measured snow cover variability

Similar inter-annual patterns of snow depth distribution were observed using TLS (Figs. 4a-c). However, the variability of the snow depth distribution and thus of snow cover onset and disappearance at certain locations was high over both the N and S facing rock walls. Areas accumulating a thick snow cover can be in the immediate vicinity of snow-free areas due to strongly varying micro-topographic effects. The snow cover was more homogeneous and thicker on the smoother S facing dip slope than on the steeper and rougher N facing scarp slope. Steep to vertical areas far above ledges or areas close to the ridge were usually snow-free, as was the case for the N3 and R2 loggers (Figs. 4a-c). Locations close to the foot of the rock wall and steep areas just above flat ledges accumulated mean snow depths up to 3.5 m.

Inter-annual snow depth variations are illustrated in Fig. 5 for both the four locations discussed in detail and for the flat field AWS. Snow depths were on average 1 m lower at both the AWS and NSRT logger locations in 2013-2014 compared to 2012-2013, resulting in snow disappearance up to 4 weeks earlier in 2014.

### 4.1.2 Modelled snow cover variability

The evaluation of the snow depth distribution modelled using Alpine3D (Figs. 4d-f) against data from three independent TLS revealed a reasonably well reproduced snow depth distribution with $r^2 = 0.52\text{-}0.95$ (Figs. 4j-l), while absolute snow depth differences were in the range of +1.5 m to -1 m (Figs. 4g-i). Considering the area around NSRT locations modelled snow depths are often underestimated (Fig. 5), whereas they are overestimated while averaging over the entire model domain (MBE = 0.31 – 0.81 m). The MAE of the N and S slopes varied between 0.47 – 0.77 m and always indicated higher deviations between snow depth observations and predictions in the heterogeneous N facing slope (Table 2).

In Fig. 5 the evolution of modelled snow depths for the four selected locations within both the N (N3, N7) and the S (R2, S9) facing slopes is shown. While R2 and N3 lacked snow, snow accumulated for 7.5 to 9 months per year at N7 and S9. The modelled winter snowpacks are compared to measured TLS snow depths (markers in Fig. 5) on the dates of TLS campaigns. At the shaded N7 location, the measured and modelled snow depths fit well in early winter (December/January 2012-2013 and 2013-2014), while modelled snow depths are underestimated by 0.55 m in early summer (2012-2013). In the S facing slope differences between measured and modelled snow depths are modest in early winter (0.04 and 0.5 m in December), while during the course of the winter and ablation period modelled snow depths were underestimated by up to 0.9 m. Although absolute snow depth differences are up to 0.9 m in the S slope, the snow cover durations (Table 1) were satisfactory reproduced by the model. The accurately modelled timing of snow cover onset and disappearance was confirmed by NSRT data in the grid cells corresponding to N7 and S9 (see Figs. 7b, c; Section 4.3), as well as at all other NSRT locations (not shown).

## 4.2 Modelled surface energy balance at selected points

The modulating influence of the snow cover on the rock thermal regime close to the surface (0.1 m depth) can be assessed by comparing the modelled surface energy balance of the snow-free to that of the snow-covered scenario. This was done at the locations of one sun-exposed (S9) and one shaded (N7) NSRT logger. In Fig. 6, modelled monthly means of each individual energy flux are shown. The terms of the energy balance were defined in Section 3.1.1.

### 4.2.1 Snow-free scenario

In the absence of a snow cover, the modelled surface energy balance was strongly influenced by local topographic effects (e.g. steep rock, aspect). At the steep, shaded point N7 (Fig. 6b) almost no solar radiation was received and energy was lost by longwave radiation emission from October to February. The resulting net radiation flux $Q_{net}$ was therefore negative. Furthermore, the latent heat flux $Q_{latent}$ was negative during the entire 2-year period. To compensate the negative fluxes, energy was transferred towards the surface by convection of sensible heat $Q_{sensible}$ from the warmer air to the colder rock surface along with the ground heat release in fall and winter. The net flux resulted in effective ground heat loss during the months with low solar elevation (November-February). $Q_{rain}$ was negligibly small compared with other fluxes and will not be discussed further here. $Q_{net}$ increased uniformly from negative values in winter to positive values in summer. Between March and September/October more radiation was absorbed than reflected and emitted, causing a positive $Q_{net}$, which was

mainly compensated by $Q_{sensible}$. $Q_{ground}$ was positive (i.e. directed into the rock) during spring and summer resulting in effective ground warming.

The evolution of the energy transfer terms of S9 (Fig. 6d) were similar to N7. Only $Q_{net}$ was positive throughout the whole year in the sunny slope, displaying a sinusoidal cycle with minimum values in winter und maxima in summer. The strong $Q_{net}$ input in winter is caused by stronger direct solar radiation input on steep S facing slopes due to the low solar elevation.

### 4.2.2 Snow-covered scenario

The accumulation of a thick, long lasting snow cover modulated the dominant driving factors of the surface energy balance considerably. Here too, the monthly evolution of the energy fluxes in the sun-exposed location S9 (Fig. 6c) were similar to those in the shaded location N7 (Fig. 6a), although variations in the magnitude of the fluxes were observed. The energy loss by $Q_{net}$ was mainly compensated by the sensible heat flux from the warmer air towards the colder snow surface during the months with low solar elevation (November-January). All other energy transfer terms were small compared to the snow-free scenario. The small $Q_{ground}$ is caused by the insulating effect of the snowpack, which prevented an effective heat emission in winter. Between March/April and September more radiation was absorbed than reflected and emitted, causing a positive $Q_{net}$. In contrast to the snow-free scenario, in which all energy was used to warm the ground, under snow-covered conditions any energy surplus $Q_{snow}$ was used for snow melt between March/April and July. The energy surplus first resulted in a heating of the snowpack to 0 °C followed by melt, which corresponded to the zero curtain period of measured and modelled NSRTs (Figs. 7b, c). Thus, the snow cover prevented ground warming between March and July with NSRTs remaining around 0 °C below the snowpack. $Q_{ground}$ was negligible during the snowmelt period and just increased after the snow ablation in July/August and September.

### 4.3 NSRT variability at selected locations

The measured and simulated NSRT evolution at 0.1 m depth in the four selected NSRT logger locations with differing snow conditions (no snow, snow; Table 1) in both the N and the S facing rock walls are illustrated in Fig. 7. Point verification ($r^2$, MBE, MAE) was performed between measured and modelled NSRT for each individual location. First the NSRT evolution at snow-free locations is described, and then the modulating effect of the snow cover on NSRT is emphasized.

### 4.3.1 NSRT variability at snow-free locations

At NSRT locations lacking snow, measured NSRTs closely followed air temperature in the shaded N face (N3, Fig. 7d) while pronounced daily NSRT amplitudes of up to 10 °C could be observed in the sun-exposed rock wall (R2, Fig. 7e) during the whole investigation period.

At N3 and R2 the modelled NSRT evolution was in good accordance with measured NSRT with $r^2 = 0.82$-0.94 (Figs. 8c, f). Although NSRT evolution was successfully reproduced by Alpine3D, the MAE between measured and modelled NSRT were 2.2/2.3 °C at N3 and 2.6/2.8 °C at R2 (Table 1). The MBE was -2.1/-1.7 °C for N3 and -2.5/-2.1 °C for R2, indicating persistently colder modelled NSRT conditions, which is also illustrated by dT in Figs. 7e and d.

### 4.3.2 NSRT variability at snow-covered locations

At locations favouring the accumulation of a thick snowpack the NSRT evolution was strongly controlled by snow for around 7.5 to 9 months of the year in both the N and the S facing rock walls. After the onset of the continuous snow cover in October/November the rock surface was partly decoupled from atmospheric influences. In the N facing slope (N7, Fig. 7b) measured NSRT oscillations were damped, but continuously decreased down to -4 °C, thus clearly showing the occurrence of permafrost at this location, while in the S facing

slope (S9, Fig. 7c) measured NSRT remained close to 0 °C. The timing of snow cover onset and disappearance were similar in both the N and the S facing slope. In 2013-2014 the snow cover onset was similar compared to 2012-2013, while the snow disappearance up to 4 weeks earlier (mid-June). The latter caused 0.3 °C (S9) respectively 0.4 °C (N7) warmer MANSRTs in 2013-2014 than in the previous year due to snow-free conditions during the weeks with most intense solar radiation (mid-June to mid-July).

At the locations accumulating a thick snow cover the temporal evolution of modelled NSRTs are in good accordance with the measured ones in both the shaded (N7) and the sun-exposed (S9) slopes with $r^2 = 0.80$-$0.94$ (Figs. 8a, d). When comparing measured and modelled NSRT evolution, the modelled timing of the snow cover onset, of the zero curtain period and of snow disappearance was similar (Figs. 7b, c and dT in these). This underlines that satisfactory modelled snow cover duration is the most important factor influencing modelled

NSRT evolution, rather than accurately modelled absolute snow depths. Variations between measured and modelled NSRT are small at the S facing S9 with a MAE of 0.6/0.6 °C and a MBE of -0.3/-0.4 °C, indicating too low modelled NSRTs in summer. At N7 measured and modelled NSRT fit well together during the snow-free period, while measured NSRTs are colder than modelled ones during the snow-covered period resulting in a MBE of 0.8/0.8 °C and a MAE of 1.1/1.0 °C (Table 1).


### 4.3.3   Thermal effect of snow

The previously discussed modulating influence of the snow cover on the surface energy balance and its effects on the ground thermal regime can be emphasized by comparing NSRTs at the snow-covered N7 and S9 to the modelled snow-free scenario at these locations (blue lines in Figs. 7b, c). Using the snow-free scenario, modelled

NSRT oscillations of N7 and S9 were pronounced during the whole study period, indicating a permanent energy exchange between the atmosphere and the rock. MANSRTs were -2.8/-1.9 °C at the shaded N7 and 0.4/1.3 °C at the sun-exposed S9. This contradicts the NSRT measurements at these locations (Section 4.3.2). Measurements reveal a permanent insulation of the rock by a continuous snowpack between October/November and June/July. Neither cold atmospheric conditions in winter, nor strong insolation and warm air temperatures between May

and July (all energy available used for snow melt, Figs. 6a, c) affected the rock thermal regime below the snowpack. Thus the potential thermal effect of a thick, long lasting snowpack accumulating in steep rock can locally be quantified: at locations accumulating a long lasting, insulating snow cover the measured MANSRTs were 2.9/2.4 °C higher in the shaded and 2.0/1.4 °C higher in the sun-exposed rock wall, while comparing to modelled MANSRT of the snow-free scenario (Table 1). The negligence of snow in steep rock resulted in

deviations between measured and modelled (snow-free) NSRT causing the $r^2$ to decrease by 0.26/0.21 at N7 and 0.57/0.51 at S9 (Figs. 8b, e) and the MAE to increase by 4.1/2.7 °C at N7 and 4.7/3.4 °C at S9.

### 4.4   MANSRT variability in the entire rock walls

A comprehensive analysis of all 22 NSRT locations was used to evaluate the spatial performance of Alpine3D in

modelling the potential effect of snow on NSRTs. Both the measured and modelled NSRT data of all 11 N

facing locations and of all 11 S facing ones were used to calculate means of MANSRT, MBE and MAE over the individual N and S facing rock walls (Table 3).

### 4.4.1 Snow-covered scenario

The topography driven difference of the measured mean MANSRT between the entire N and the entire S facing rock wall were 3.6/3.2 °C. Such a small deviation is reasonable when taking into account that the rock walls are facing rather NW and SE than N and S (Fig. 1a, Appendix Table 1A), as well as considering the accumulation of a thick snow cover at 7 of 11 locations in both the N and S slopes.

At the corresponding 22 grid cells, the modelled mean MANSRT difference for the snow-covered scenario

across the entire N and S facing slope is 2.6/2.3 °C and thus around 1.0 °C lower than the measured values (Table 3). This is mainly caused by too low modelled NSRTs and thus MANSRTs, especially in the sun-exposed rock wall during snow-free periods (Fig. 9) and at locations without snow (N and S slopes) resulting in a MBE of -1.3/-1.0 °C. These results are supported by the model verification at the single locations in Section 4.3.2, but clearly show that model uncertainties increase on the rock wall-scale due to the pronounced spatial variability.

Uncertainties while applying Alpine3D to simulate NSRT in steep rough rock implies a MAE of 1.6/1.7 °C for both the entire shaded and sunny rock wall.

The measured and modelled small-scale variability of MANSRT at all 22 NSRT locations and corresponding grid cells separated for the individual N and S facing rock walls are illustrated in Fig. 9, as well as the modelled MANSRT variations for the entire model domain, depending on whether the grid cells are N or S facing. For all

cases, the MANSRT variability within the individual N and S slopes was higher in 2012-2013, which is the result of two effects. In 2012-2013 the mean annual air temperature was 0.8 °C lower than in 2013-2014, causing MANSRTs at snow-free locations to decrease by around 0.6 °C. In contrast, MANSRTs at snow-covered locations in the N slope increased by up to 0.4 °C due to an early onset of a long lasting, insulating snow cover. In early winter 2013-2014 the absence of a sufficiently thick, insulating snow cover resulted in effective ground

heat loss at these locations (Haberkorn et al., 2015a).

### 4.4.2 Snow-free scenario

In the absence of a snow cover, the modelled MANSRT variability was much lower within the individual rock walls (Fig. 9). Assuming the modelled snow-free scenario in the entire rock walls, resulted in mean MANSRT of

-3.3/-2.3 °C within the N and of 0.1/0.8 °C within the S facing slopes (Table 3). In correspondence to the single NSRT locations (Section 4.3.3) the mean MANSRT of snow-free simulations confirmed too low modelled MANSRT when compared with both observations and snow-covered simulations (Fig. 9).

### 4.4.3 Modelled spatial distribution of MANSRT variability

The influence of the snow cover on rock surface temperatures and the previously discussed rock temperature results are summarized in Fig. 10. Here modelled MANSRT for each grid cell of the entire model domain of the Gemsstock ridge (not just at selected NSRT locations) are shown for both the snow-free (Figs. 10a, b) and the snow-covered scenario (Figs. 10c, d) for the year 2012-2013, as well as their differences (Figs. 10e, f). Pronounced MANSRT deviations between both scenarios are obvious.

Under snow-free conditions the mean MANSRT averaged over the entire N slope are -2.9 °C in 2012-2013 and -1.9 °C in 2013-2014 and thus clearly indicate a possible occurrence of permafrost in the rock walls under snow-

free conditions. Mean MANSRTs averaged over the entire S facing slope are -0.3/0.7 °C and therefore correspond to conditions at the lower fringe of permafrost occurrence. The MANSRT variability within the slopes is more homogenous compared to the snow-covered scenario, since rock temperatures mainly depend on topography and thus solar insolation.

In contrast to the snow-free scenario, the accumulation of a heterogeneously distributed snow cover strongly changes the conditions at the rock surface and thus rock temperatures. In the snow-covered scenario, MANSRT variability is pronounced in steep rock walls depending on the accumulation of a continuous snow cover, on snow depth and snow cover duration. The snow depth distribution varies strongly due to the complex micro-topography in the rock walls with rock portions accumulating thick snow in close vicinity to rock portions lacking snow. MANSRTs were highest at the foot of both rock walls and gradually decreased from flat to steeper areas due to both snow depth decrease and low insolation in the N slope at locations without snow. MANSRT at locations shadowed by rock outcrops or in rock dihedrals were colder compared to their surrounding areas (arrows in Figs. 10c, d). The influence of the snow on rock surface temperatures is emphasized by 2.5/1.8 °C (N), respectively 2.3/1.3 °C (S) higher modelled MANSRTs averaged over the individual N and S facing slopes for snow-covered, than for snow-free conditions.

### 4.5    Influence of grid resolution

The Alpine3D model performance was tested at different spatial-scales (0.2 m, 1 m, 5 m) to analyse the loss of model accuracy for lower computational effort. At locations with a rough micro-topography the loss of information was important due to the aggregation of the initial DEM (0.2 m resolution) to 1 m and 5 m. Slope angles were only sampled at <70° (1 m resolution) and <60° (5 m resolution), whereas in reality the rock was nearly vertical. Aspects were displaced by up to 90° (Appendix Table 1A). This reduces the accuracy of the precipitation scaling and the modelled energy balance components (e.g. net radiation, turbulent fluxes). Shortwave incoming radiation was inadequately modelled at locations with strongly varying micro-topography when increasing grid cell size. However, on a monthly basis, errors in net radiation due to a coarser resolution were smoothed. In addition to smoothed slope angles, 2 or 3 NSRT locations are often merged together in a single grid cell at 5 m resolution. The strongly varying micro-topography and consequently also the snow depth distribution is thus inadequately represented at the 5 m scale. Considering NSRT simulations at each of the 22 logger locations separately revealed that NSRTs modelled at 0.2 and 1 m resolution are in good accordance with measurements, while at 5 m resolution NSRTs are at most locations poorly modelled due to too strong aggregation and thus the over- or underestimation of snow in both the N and the S facing slopes. In Table 4 the influence of different grid resolutions on measured and modelled (snow-covered scenario) MANSRTs averaged over the individual rock walls and their uncertainties are shown. In the N facing slope a resolution of 1m is sufficient to model rock temperatures. Comparing the modelled MANSRTs to measurements result only in up to 0.3 °C deviations for 0.2 and 1 m resolution, while these MANSRT deviations increased to 1.2 °C at 5 m resolution. The MBE and MAE are similar for all resolutions. In contrast in the more homogenous S facing slope the modelled MANSRT at 5 m resolution corresponds well to measurements, since micro-topography and snow depth distribution are smoother than in the N slope.

## 5  Discussion
### 5.1  Model uncertainties

Limitations in reproducing snow cover characteristics, energy balance components and rock temperatures in the simulations were introduced by uncertainties in the input data (see Section 3), as well as by the adequacy of the

process representation in the Alpine3D model. Some physical processes, such as lateral heat fluxes at the rock surface (in our grid model heat fluxes are calculated perpendicular to the rock surface, all other fluxes are lateral) or through the narrow ridge and the heterogeneous wind field in extremely steep terrain, are currently insufficiently represented by our model setup. Some model uncertainties and their consequences on the modelled rock thermal regime of steep rock walls are discussed below.

In this study discrepancies in modelling absolute snow depths in steep rock walls are evident (Figs. 4, 5). This is a consequence of the linear precipitation scaling algorithm used here. Snow settlement is calculated for snow depths at the AWS location and is then linearly scaled into the rock walls, but snow depths and the meteorological forcing obviously differ between the flat field AWS and the rock walls. This causes the snowpack to settle differently and in a non-linear manner. Differences in settling calculated at the AWS and for

the grid points in the Alpine3D model domain therefore cause absolute snow depth errors. However, on the basis of measured NSRTs (Figs. 7b, c) it is evident that the snow cover duration (Table 1) is well reproduced by the model. The realistically modelled snow cover duration over the winter was found to be more important for modelling the ground thermal regime than accurately modelled absolute snow depths at certain points in time. This agrees with the findings of Marmy et al. (2013) and Fiddes et al. (2015). Although measured and modelled

snow depth differences were >1.0 m (Figs. 4, 5), these snow depth differences do not affect the rock thermal regime since steep, bare rock is already decoupled from atmospheric influences at snow depths >0.2 m (Haberkorn et al., 2015a). Amongst others, Luetschg et al. (2008) and Zhang (2005) stated that the influence of snow depth variations on ground temperatures in the presence of a thick snow cover are small, whereas snow depth variations only have strong effects on the ground thermal regime for snow thinner than 0.2 m.

As a consequence of the strong snow depth variability in the rock walls snow depth comparisons at specific points are difficult. Although, verification of snow depth over the entire rock walls suggest an overestimation of snow depth (Table 2, Figs. 4g-i), snow depths were underestimated locally by Alpine3D, e.g. at NSRT locations (Fig. 5). The efficiently modelled snow cover duration at NSRT locations thus implies an underestimation of snow melt in the model. This agrees with an underestimation of surface heat fluxes (e.g. shortwave incoming

radiation), reflected in too low modelled NSRTs (dT in Figs. 7d, e) and consequently MANSRTs (MBE in Table 1) at locations lacking snow and during the snow-free period. A likely explanation is that both air temperature and wind speeds, measured at the flat field AWS may be poorly representative for the prevailing conditions in the rock walls and therefore turbulent flux simulations are biased. In addition, the underestimation of snow melt may also be partly explained by the 1d snow module which does not account for lateral heat flow between

adjacent snow-free and snow-covered rock portions, as well as micro-meteorological processes due to unevenly distributed heating during the ablation period which in reality accelerates snow melt. Nevertheless, the model verification showed that the overall performance of Alpine3D modelling snow depths and consequently rock temperatures in steep slopes in the current setup provides useful improvements compared to the common assumption of a lack of snow in thermal modelling of idealized rock walls exceeding 50° (e.g. Fiddes et al.,

2015; Gruber et al., 2004a; Noetzli and Gruber, 2009; Noetzli et al., 2007).

Further, we found that the insulation by snow was too strong in the simulations. Modelled NSRT and consequently MANSRT were therefore positively biased during the snow-covered period in the steep, rough N facing slope and thus measured negative NSRTs could not be reproduced (Fig. 7b). This has two possible

explanations: (i) The snow thermal conductivity is too low in the model and/or (ii) the existence of lateral heat fluxes due to the strong thermal interaction of micro-topography and micro-climate between snow-covered and snow-free rock portions, which lead to stronger cooling below snow pixels than simulated with the 1d model. While assuming predominately 1d vertical heat conduction in the snow and ground, a part of the energy balance and thus the complex lateral heat flow occurring at the rock surface, as well as in steep, narrow ridges is poorly described or missing (Noetzli et al., 2007). Effective ground heat loss in autumn 2013-2014 was observed and modelled at exposed locations due to an initially thin snow cover, but a heat exchange between adjacent locations covered with thick snow was not reproducible by the model, although it was measured (Haberkorn et al., 2015a). In contrast modelled and measured NSRTs in the homogenous S facing slope supported the validity of the 1d heat conduction assumption at snow-covered locations since here a continuous, smooth snowpack was an effective barrier to heat loss from the ground to the air (Fig. 7c). Finally, difficulties in partitioning the measured incoming shortwave radiation in a direct and diffuse component, particularly for low sun angles, may explain the stronger modelled net radiation for snow-free conditions in the shaded (Fig. 6b) than in the sun-exposed slope (Fig. 6d), which is amplified by differences in slope and aspect between the model domain and reality (Appendix Table 1A).

## 5.2 Impacts of snow in rock walls

Meteorological conditions and topographic properties like slope angle, aspect, surface roughness (Gruber et al., 2004b; Noetzli et al., 2007) and local shading effects (Mott et al., 2011) control the surface energy balance and their annual variations in rock wall sectors lacking snow. Changes of local conditions at the rock surface due to the accumulation of a snow cover modify the importance of influencing factors on the ground energy balance (Hoelzle et al., 2001). This study emphasizes the need to account for the strongly varying snow cover in thermal modelling of steep, fractured, complex rock walls.

Alpine3D was used to simulate rock surface temperatures for both a snow-covered (precipitation scaling) and a snow-free scenario (zero precipitation input), in order to estimate the error introduced by neglecting snow in steep bedrock thermal modelling. The results are summarized in Fig. 10, where the comparison of snow-free and snow-covered simulations show a prominent warming effect of the snowpack on MANSRT over the entire N and S facing rock walls. These model results are supported by measured NSRT data and model predictions at both the point- (Table 1) and rock wall-scale (Table 3), as well as by previous observations reported by Haberkorn et al. (2015a). Modelled MANSRT differences between snow-covered and snow-free conditions were due to the insulation of the rock by a continuous snowpack, despite the strong solar insolation in spring and early summer (Fig. 6). Under snow-free conditions the excessive radiation input in early summer cannot compensate the effective ground heat loss in winter. The modelled MANSRT increase of 1.3 – 2.5 °C found for both snow-covered N and S facing steep rock walls compared to snow-free simulations (Figs. 10e, f) is in the same order of magnitude than the cooling or warming effect of snow on mean-annual ground surface temperatures modelled by Pogliotti (2011). However, Pogliotti (2011) suggested that a warming effect of mean-annual ground surface temperatures can only occur on gentle slopes, while cooling can occur everywhere and also in conditions of a nearly perennial thin snow cover. The latter is doubted, since our observations show that thin snow melts fast at elevations around 3000 m a.s.l. especially on steep S faces with strong insolation. In shaded slopes the increased MANSRT caused by thick snow confirms the findings of Magnin et al. (2015). In contrast, in sunny rock walls both measurements and model results at the point- and spatial scale (Tables 1,3) challenge the hypotheses

presented by Magnin et al. (2015) and Hasler et al. (2011), who supposed a cooling effect of a snow cover due to the shielding of the rock surface from radiation influences during the months with most intense insolation. Discrepancies with our observations may have three reasons: (i) These authors estimated snow depths qualitatively rather than quantitatively. (ii) They adopt the widespread theory of an insulating snow cover with depths exceeding 0.6 m for blocky terrain (Hanson and Hoelzle, 2004, Keller and Gubler, 1993, Luetschg et al., 2008), while Haberkorn et al. (2015a) found the insulation effect on NSRT at smooth rock surfaces already present for snow depths exceeding 0.2 m. (iii) Their observations are a few point measurements, whereas we complemented multiple point measurements with simulations of the entire rock walls. At Gemsstock a thick snow cover accumulates in most parts of the rock walls between October and June/July. Considering snow in sunny, steep rock for shorter periods or only for the months with strongest insolation (March to June) most likely has a cooling effect on rock surface temperatures.

In this study it has been proven that both net radiation and the snow cover are the key factors driving ground temperatures and determine whether permafrost is present or not in steep, rough rock walls, which was already proposed for moderately inclined terrain by Hoelzle et al. (2001). In steep S facing mountain ridges up to 3000 m a.s.l., permafrost is most likely absent independent of the evolution of a thick snow cover, as shown in Figs. 10b and d. In contrast in steep rugged N facing rock walls the accumulation of a thick snow cover prevents a continuous permafrost distribution (Fig. 10c), while permafrost would most likely be present in areas without or with only thin snow (Fig. 10a). These results confirm recent two-dimensional numerical simulations made for east/north-east facing Scandinavian rock walls by Myhra et al. (2015), who found that the size of snow-free rock portions are crucial for warming or cooling a rock wall. In addition, these authors show that the existence of permafrost in steep bedrock varies strongly depending on thickness and extension of an insulating snow cover, which can lead to permafrost temperature increase and taliks in steep slopes. We therefore suggest that in recent permafrost distribution assessments in the European Alps based on energy balance (Fiddes et al., 2015) or statistical modelling (Boeckli et al., 2012a,b) mean annual rock surface temperatures were possibly modelled too low by around 2 °C in steep bedrock as a result of neglecting snow.

Mismatches of scale issues in distributed permafrost modelling arise often while validating the model results based on grids of tens to hundreds of metres to point measurements (e.g. Gubler et al., 2011; Gupta et al., 2005; Schlögl et al., 2016). Here, a point- and spatial model validation of NSRTs and snow depths were performed at different grid cell sizes (0.2 m, 1 m, 5 m; Table 4). In both the N and the S facing rock walls, the point- and spatial validation with data at 1 m resolution is reasonable, to accurately model the snow cover and ground surface temperatures in steep rugged rock faces. The decrease in computational time by reducing the grid resolution from 0.2 to 1 m, is significant (25 times lower). Additionally, a DEM resolution of 1 m is considered to be precise enough to detect ledges within the rock face, which are essential for snow accumulation in steep rock (Haberkorn et al., 2015a; Sommer et al., 2015). At a resolution of 5 m the loss of topographic, as well as accurate snow depth information results in an inadequately modelled rock thermal regime. Model runs at coarser spatial-scales are thus assumed to be unsuitable for modelling temperatures in complex steep rock walls, such as the Gemsstock ridge. Variations of surface processes due to micro-topographic inhomogeneity occur at small-scales, providing the motivation for high-resolution numerical modelling in complex topography in order to establish a basis of proper validation of grid-based model results.

## 6   Conclusions

The potential to model the strongly heterogeneous snow cover and its influence on the rock thermal regime on two rugged, steep mountain rock walls has been studied at the Gemsstock ridge (central Swiss Alps) over a two year period. The results were obtained using the spatially distributed physics-based model Alpine3D in combination with a precipitation scaling approach.

In the rough rock walls, the heterogeneously distributed snow cover was moderately well reproduced by Alpine3D with absolute snow depth differences varying between +1.5 and -1.0 m and a MAE between 0.47 and 0.77 m averaged over the entire rock walls. However, the snow cover duration was well reproduced by the model and proved to be most important for realistically NSRT modelling.

Rock temperatures are convincingly modelled, although modelled NSRTs and thus MANSRTs are somewhat too low during snow-free periods and at locations without snow, as indicated by a MBE varying between -0.2 and -1.3 °C in the rock walls. Model verification suggests an MAE of 1.6/1.7 °C in both the entire shaded and sunny rock walls.

Remaining errors in snow depth and consequently rock temperature simulations are explained by inadequate snow settlement modelling, due to linear precipitation scaling, missing lateral heat fluxes in the rock and by errors due to shortwave radiation, air temperature and wind interpolation, which are complex in such terrain.

The influence of the snow cover on rock surface temperatures was investigated by comparing a snow-covered model scenario (precipitation input provided by precipitation scaling) with a snow-free (zero precipitation input) one. A strong increase in MANSRTs in both the shaded and sun-exposed steep rock walls induced by a thick long lasting snow cover were both measured and modelled. MANSRT were by 2.5/1.8 °C higher in the shaded and 2.3/1.3 °C higher in the sun-exposed rock walls when comparing the modelled snow-covered scenario to the snow-free one. As snow reduces ground heat loss in winter, it has an overall warming effect on both N and S facing rock walls despite the fact that it provides protection from solar radiation in early summer.

The model performance was tested at different scales ranging from 0.2 m to 5 m. A DEM resolution of 1 m was found to be detailed enough to detect the strongly variable micro-topography in steep, rugged rock walls and hence a grid resolution of 1 m is adequate to accurately model the snow cover and rock surface temperatures. Coarser resolutions are not appropriate at the Gemsstock site.

The correction of winter precipitation input using a precipitation scaling method based on TLS improved snow cover and thus also rock temperature simulations in the complex rock walls. The results of this study help to quantify the potential errors in ground temperature modelling when neglecting the evolution of a snow cover in steep rock exceeding 50°, as has often been done for idealized rock walls.

## 7  Outlook

The observations and model results discussed here are from an individual site with specific characteristics. In future studies, additional rock faces with diverse characteristics and climates should be investigated to assess the general validity of our results. The precipitation scaling method presented is currently only valid at the site-scale, but can potentially also rely on satellite imagery or airborne laser scan data to enable snow depth scaling for larger areas. Correcting for different snow settlement rates due to different snow depths will be a feasible improvement for snow depth simulations. Further improvements can be expected by considering wind fields in steep terrain and lateral heat fluxes with the Alpine3D model. While the generation of wind fields over steep slopes is an unsolved and challenging issue, the implementation of 3d advective heat fluxes in steep ridges influencing both the rock surface and ground temperatures at depth can be addressed by coupling the modelled

surface energy balance to a ground model representing 3d heat flow in the rock. This will likely allow to model a more accurate evolution of ground temperatures especially when considering only thin snow and potential disposition for slope instability. However, the need for modelled lateral heat fluxes is questionable when the model accuracy has a MAE of 1.6/1.7 °C (Table 3) and the significantly higher computational costs must be taken into account. Although ground temperature modelling over larger areas, such as the entire Alps, is not feasible at such high resolutions, our site specific approach has demonstrated the potential to reveal temperature variations for different snow cover conditions and to discuss limitations of permafrost models running at coarse-scales. Climate change impact studies critically depend on the small-scale variability at the atmosphere-surface interface. This physics-based approach can be used to study the long-term effect of a changing climate on rock temperatures and permafrost distribution.

## 8 Data availability

The data is available on request from WSL Institute for Snow and Avalanche Research SLF. The model used is available online at http://models.slf.ch/.

## 9 Author contribution

A. Haberkorn and M. Phillips designed the measurement set-up, carried out the measurements and analysed the data. R. Kenner was responsible for terrestrial laser scanning and analysis of this data. M. Lehning, N. Wever and M. Bavay developed the Alpine3D model code. In addition, N. Wever developed the precipitation scaling approach. A. Haberkorn performed the model simulations and prepared the manuscript with contributions from all co-authors.

## 10 Acknowledgements

This project was funded by the Swiss National Science Foundation (DACH Project no. 200021E-135531). We thank our project partners M. Krautblatter, D. Dräbing and S. Gruber, as well as H. Rhyner who was responsible for safety in the field. We acknowledge preparation and support of fieldwork by C. Danioth and Team at Gemsstock, the SLF Electronics, IT and Mechanics team, M. Keller, M.O. Schmid, L. Dreier, M. Sättele and F. Stucki. W. Steinkogler, J. Caduff-Fiddes and J. Noetzli are thanked for their constructive input. We thank the editor K. Isaksen and two anonymous reviewers for their valuable comments on the manuscript.

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

**Table 1. Topographic characteristics of selected NSRT logger locations with different snow conditions and the distance to the nearest ledge below (DLB). In addition analysis of observed (O) and predicted (P) snow cover duration, as well as observed MANSRT and predicted MANSRT for both snow-covered (PS) and snow-free (PSF) scenarios at selected NSRT locations and for the years 2012-2013 (12-13) and 2013-2014 (13-14). The MBE and MAE were calculated between observations and model predictions of the snow-covered respectively snow-free scenarios.**

| Logger (slope/aspect) | DLB (m) | Year | Snow cover duration | | MANSRT [°C] | | | MBE [°C] | | MAE [°C] | |
|---|---|---|---|---|---|---|---|---|---|---|---|
| | | | O | P | P | PS | PSF | PS | PSF | PS | PSF |
| N7 | 0.1 | 12-13 | 10 Oct-8 Jul | 13 Oct-4 Jul | 0.1 | 0.9 | -2.8 | 0.8 | -3.0 | 1.1 | 5.2 |
| (90°/289°) | | 13-14 | 7 Oct-11 Jun | 11 Oct-13 Jun | 0.5 | 1.2 | -1.9 | 0.8 | -2.4 | 1.0 | 3.7 |
| N3 | 10 | 12-13 | - | - | -1.4 | -3.6 | -3.6 | -2.1 | -2.1 | 2.3 | 2.3 |
| (90°/284°) | | 13-14 | - | - | -0.8 | -2.5 | -2.5 | -1.7 | -1.7 | 2.2 | 2.2 |
| S9 | 0 | 12-13 | 28 Oct-6 Jul | 31 Oct-12 Jul | 2.4 | 2.1 | 0.4 | -0.3 | -2.1 | 0.6 | 5.3 |
| (72°/165°) | | 13-14 | 4 Nov-11 Jun | 9 Nov-20 Jun | 2.7 | 2.3 | 1.3 | -0.4 | -1.4 | 0.6 | 4.0 |
| R2 | 15 | 12-13 | - | - | 2.2 | -0.3 | -0.3 | -2.5 | -2.5 | 2.8 | 2.8 |
| (58°/164°) | | 13-14 | - | - | 2.7 | 0.6 | 0.6 | -2.1 | -2.1 | 2.6 | 2.6 |

**Table 2. Snow depth validation (MBE, MAE) between measured and modelled snow depths averaged over the entire N and S facing rock walls at the dates of the independent TLS campaigns. The MBE and MAE are in [m].**

| TLS campaign | Rock wall | MBE | MAE |
|---|---|---|---|
| 7 June 2013 | N | 0.25 | 0.81 |
| | S | 0.52 | 0.74 |
| 11 December 2013 | N | 0.73 | 0.75 |
| | S | 0.47 | 0.48 |
| 28 January 2014 | N | 0.42 | 0.59 |
| | S | 0.17 | 0.31 |

**Table 3. MANSRT, MAE and MBE [all in °C] calculated within the individual N and S facing rock walls at NSRT locations. The MAE and MBE were calculated between measurements (O) and model predictions of both the snow-covered (PS) and the snow-free scenarios (PSF) at NSRT locations. Additionally mean annual air temperature (MAAT) for the years 2012-2013 and 2013-2014 is shown.**

| | | 2012- 2013 | | | 2013- 2014 | | |
|---|---|---|---|---|---|---|---|
| Scenario | Rock wall | MANSRT | MAE | MBE | MANSRT | MAE | MBE |
| O | N | -0.7 | | | -0.5 | | |
| PS | N | -1.0 | 1.6 | -0.4 | -0.6 | 1.7 | -0.2 |
| PSF | N | -3.3 | 3.9 | -2.6 | -2.3 | 2.7 | -1.8 |
| O | S | 2.9 | | | 2.7 | - | |
| PS | S | 1.6 | 1.6 | -1.3 | 1.7 | 1.7 | -1.0 |
| PSF | S | 0.1 | 4.7 | -2.8 | 0.8 | 3.4 | -1.9 |
| MAAT | | -3.2 | | | -2.4 | | |


**Table 4. Differences in grid resolution: MANSRT, MAE and MBE [all in °C] calculated within the individual N and S facing rock walls at NSRT locations. The MAE and MBE were calculated between measurements (O) and model results of the snow-covered scenario (PS) at NSRT locations for 0.2 m, 1 m and 5 m grid resolution.**


| Scenario | Rock wall | Resolution [m] | 2012- 2013 | | | 2013- 2014 | | |
|---|---|---|---|---|---|---|---|---|
| | | | MANSRT | MAE | MBE | MANSRT | MAE | MBE |
| O | N | | -0.7 | | | -0.5 | | |
| PS | N | 0.2 | -1.0 | 1.6 | -0.4 | -0.6 | 1.7 | -0.2 |
| PS | N | 1 | -0.6 | 1.7 | 0.0 | -0.2 | 1.5 | 0.3 |
| PS | N | 5 | 0.5 | 1.8 | 1.1 | 0.7 | 1.9 | 0.9 |
| O | S | | 2.9 | | | 2.7 | | |
| PS | S | 0.2 | 1.6 | 1.6 | -1.3 | 1.7 | 1.7 | -1.0 |
| PS | S | 1 | 1.7 | 1.7 | -1.2 | 2.1 | 1.8 | -0.6 |
| PS | S | 5 | 2.4 | 1.8 | -0.4 | 2.4 | 1.7 | -0.3 |

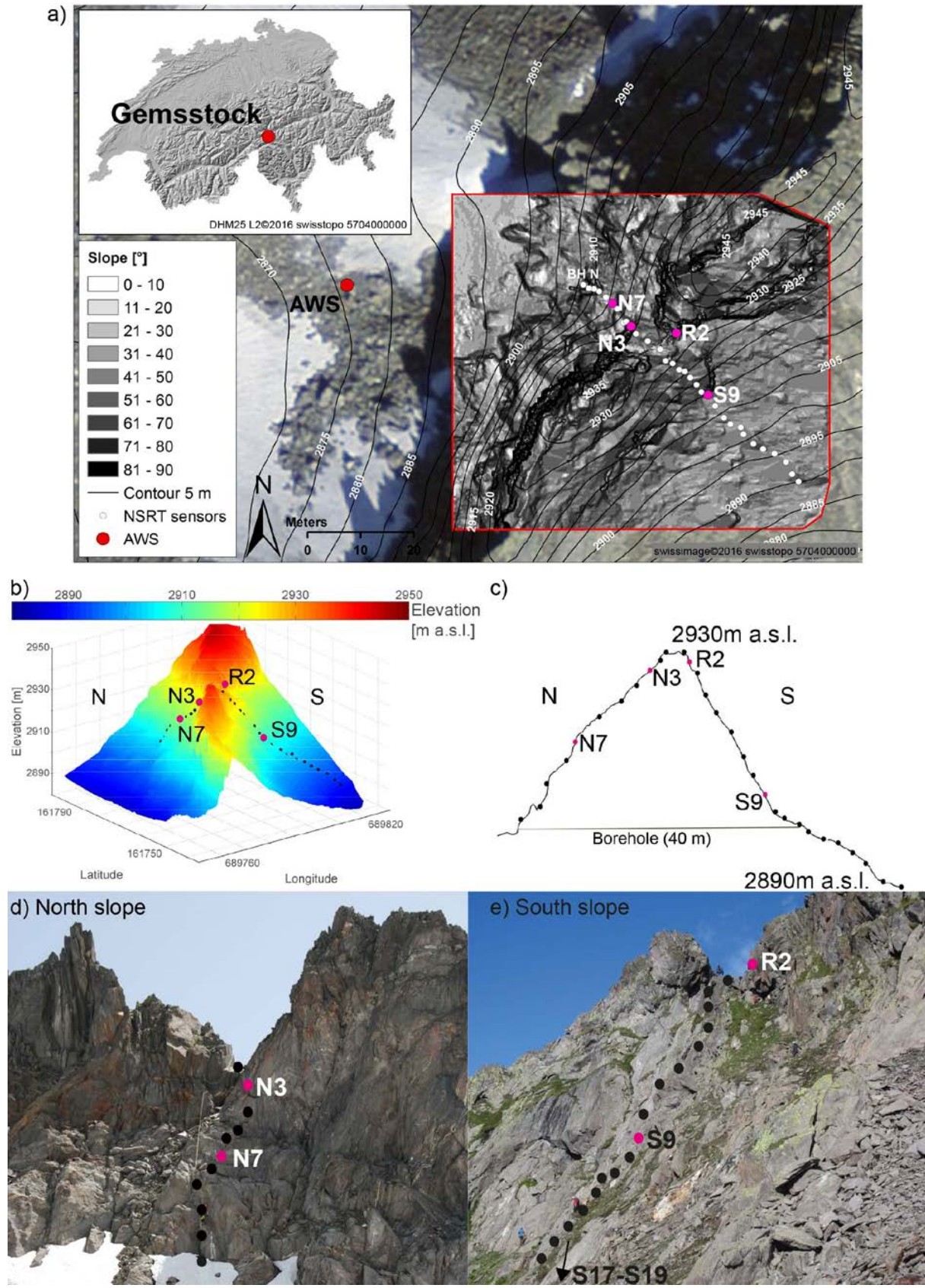

 **Figure 1. The Gemsstock study site: (a) The Alpine3D model domain with slope angles (red rectangle) based on TLS data, as well as the locations of the AWS and the NSRT devices. The location of Gemsstock in the Swiss Alps is shown in the top left inset. (b) 3d view of the DEM of Gemsstock, as well as (c) the cross-section of the Gemsstock ridge with all 30 NSRT locations. Photographs showing the (d) N and (e) S**

rock faces and the measurement set-up. (b-e) Black dots indicate the locations of the 30 NSRT locations and selected ones, discussed in further detail are highlighted in pink and labelled.


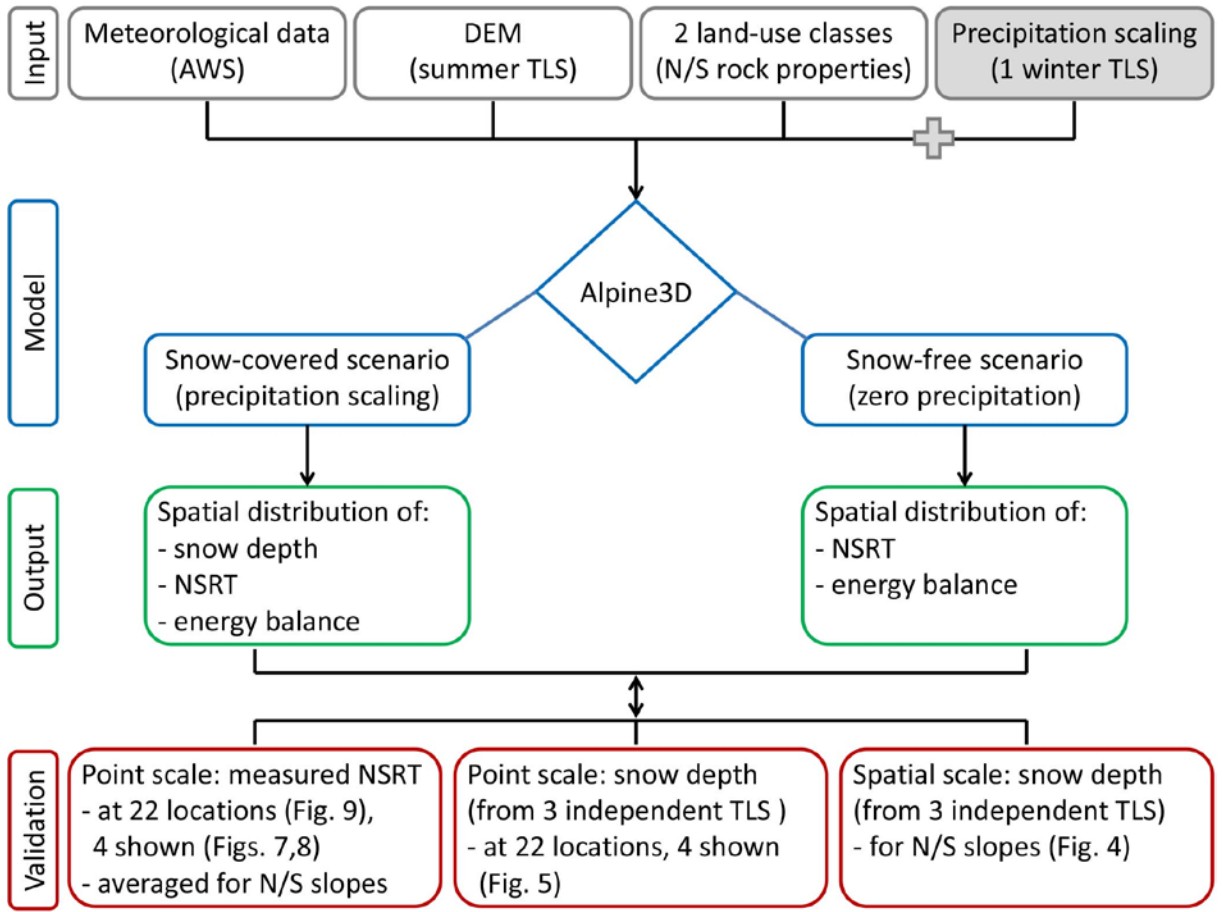

Figure 2. Flow chart of the methods applied in order to run the numerical model Alpine3D and to validate the model output at both the point- and the spatial-scale.



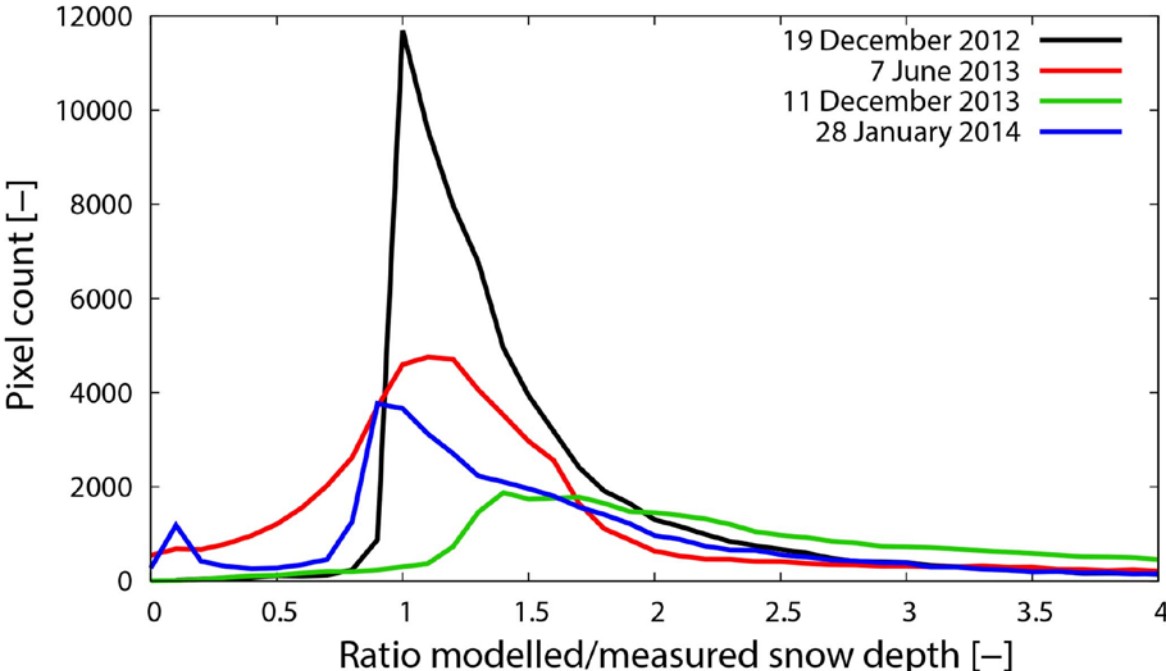

**Figure 3. Histogram of measured and modelled snow depth data. Solid lines denote the distribution of the ratio modelled over measured snow depth for the 4 TLS available. The TLS of 19 December 2012, 7 June 2013 and 28 January 2014 are centred by 1. The TLS of 19 December 2012 was used for precipitation scaling and shows the best agreement between modelled and measured snow depths.**


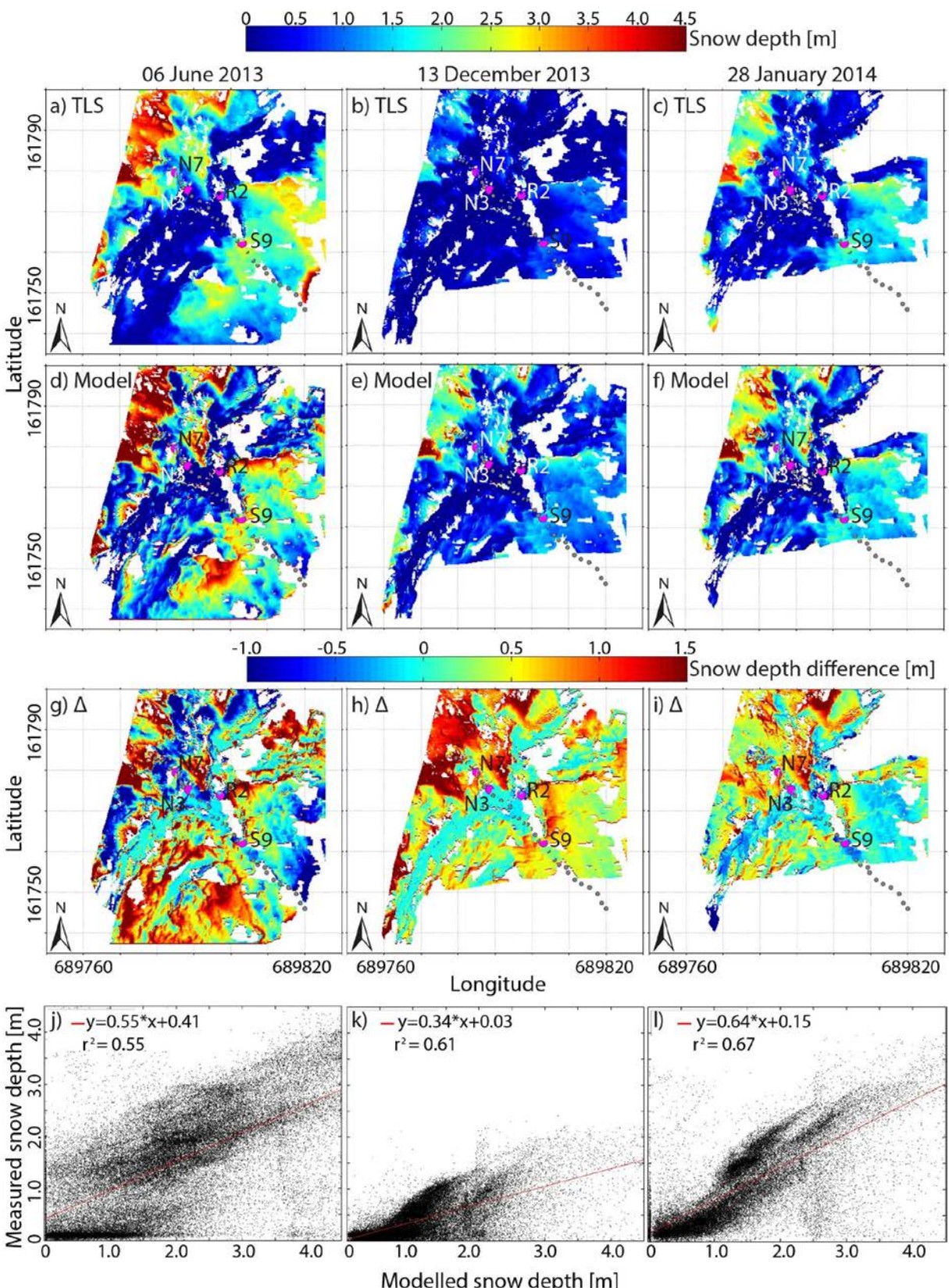

**Figure 4. Snow depth distribution: (a-c) measured based on TLS, (d-f) modelled at the same dates as the**
**TLS campaigns, (g-i) differences Δ between modelled and measured snow depth and (j-l) measured snow**
**depths as function of modelled snow depths. Grey dots indicate the locations of NSRT loggers and selected**
**ones are highlighted in pink and labelled.**

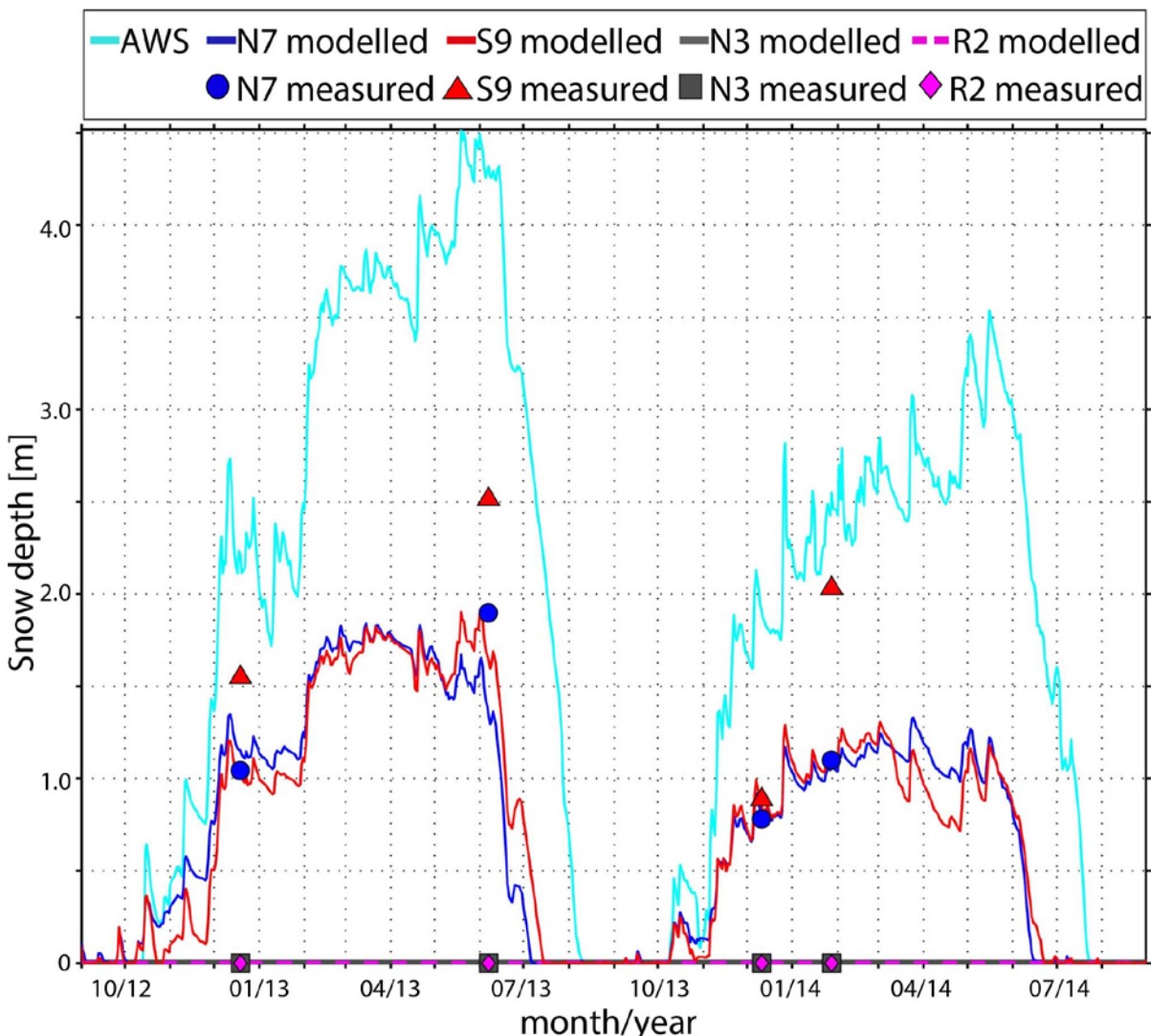

**Figure 5.** Snow depth evolution (lines) measured at the flat field AWS Gemsstock (AWS), as well as
modelled at the NSRT locations discussed in detail (N7, S9, N3, R2). Snow depths at the NSRT locations
obtained by TLS are shown as blue, red, grey dots and pink markers. The locations of N3 and R2 lack
snow for the entire investigation period. Data of the TLS campaign on 19 December 2012 is also shown
here, although the measured snow depth was used for precipitation scaling.


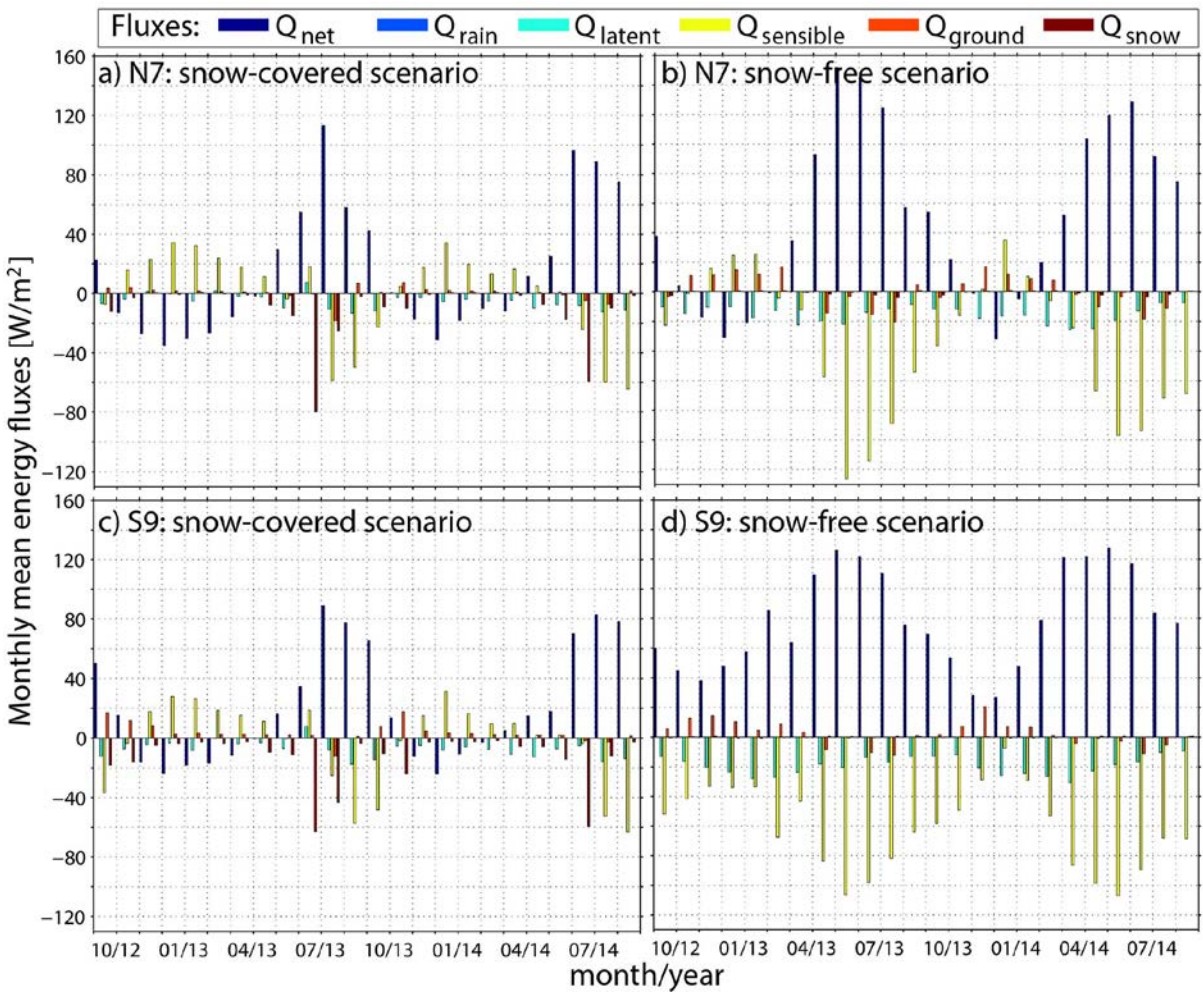

Figure 6. Modelled monthly means of all energy balance components for two selected NSRT locations. N7 (top) faces north-west and S9 (bottom) south-east. To illustrate the influence of the snow cover on the surface energy balance, the energy fluxes are shown for the snow-covered (left) and the snow-free scenarios (right). Energy fluxes are considered positive when directed towards the snowpack surface. $Q_{snow}$ is the energy available to melt the isothermal snowpack and is thus illustrated here as an energy sink.

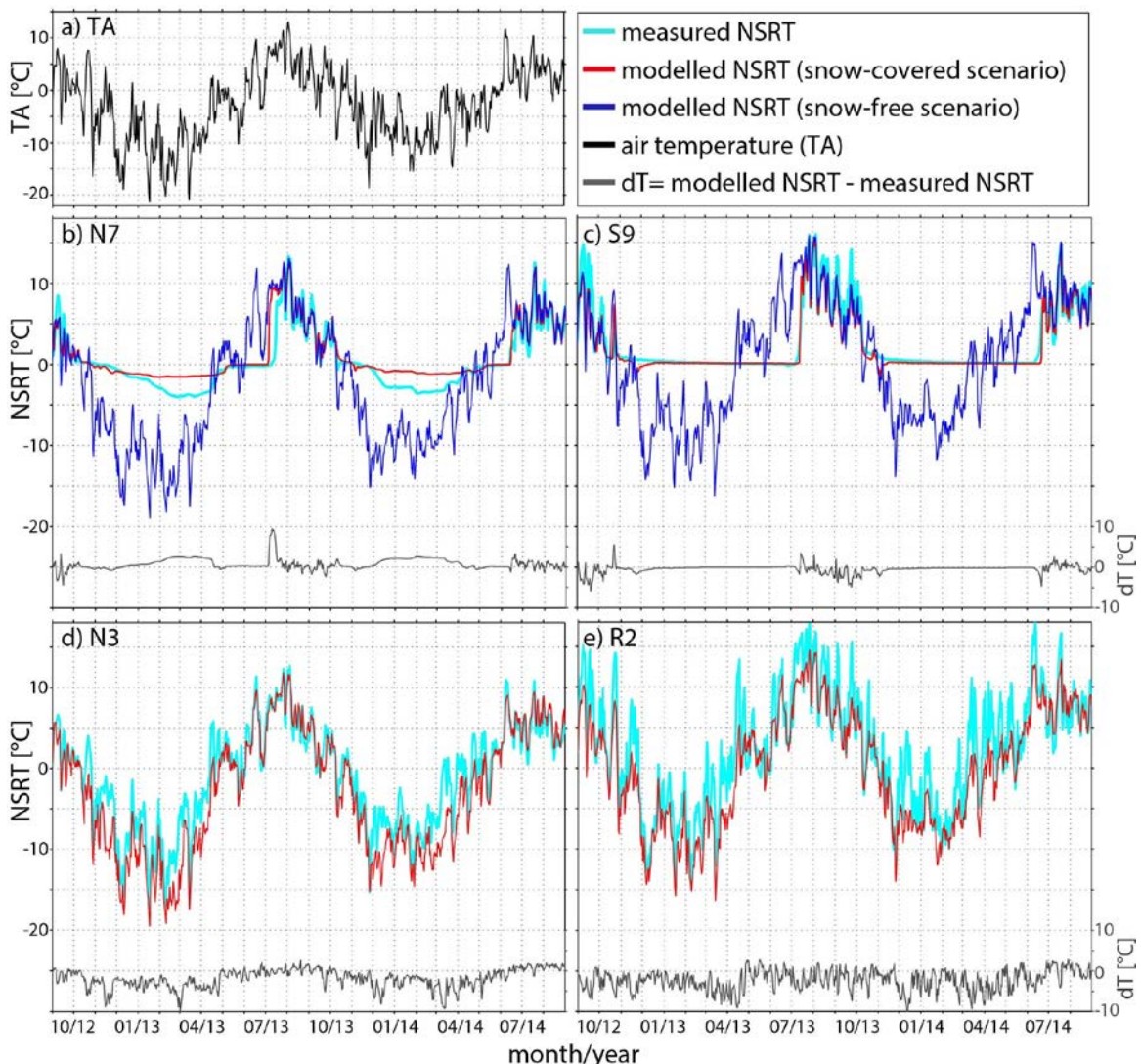

**Figure 7. (a) Daily mean air temperature at the AWS Gemsstock. (b-e) Measured and modelled daily mean NSRT are shown for four selected locations in the N and the S facing rock walls representing typical snow conditions (snow, no snow). At locations accumulating snow (N7, S9) modelled NSRTs are shown for both the snow-covered and the snow-free scenarios, while the NSRT differences (dT) were only shown between measured and modelled snow-covered conditions. At locations without snow (N3, R2) measured and modelled NSRT differences (dT) are also shown.**


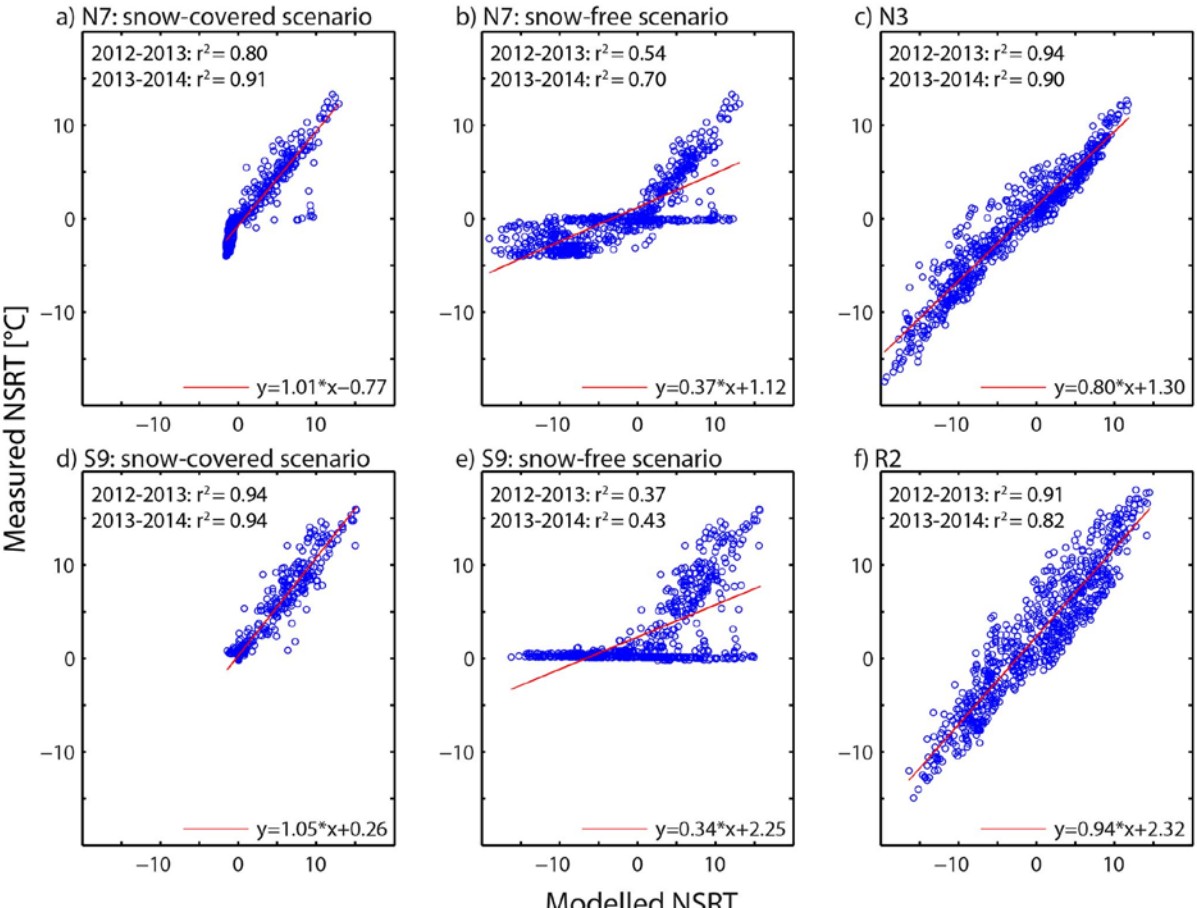

Figure 8. Two year data showing the relation between measured and modelled NSRT data for both (a,d) snow-covered and (b,e) forced snow-free scenarios, as well as for (c,f) generally snow-free NSRT locations. The mean annual $r^2$, as well as the linear relation between measured and modelled NSRT data are shown.

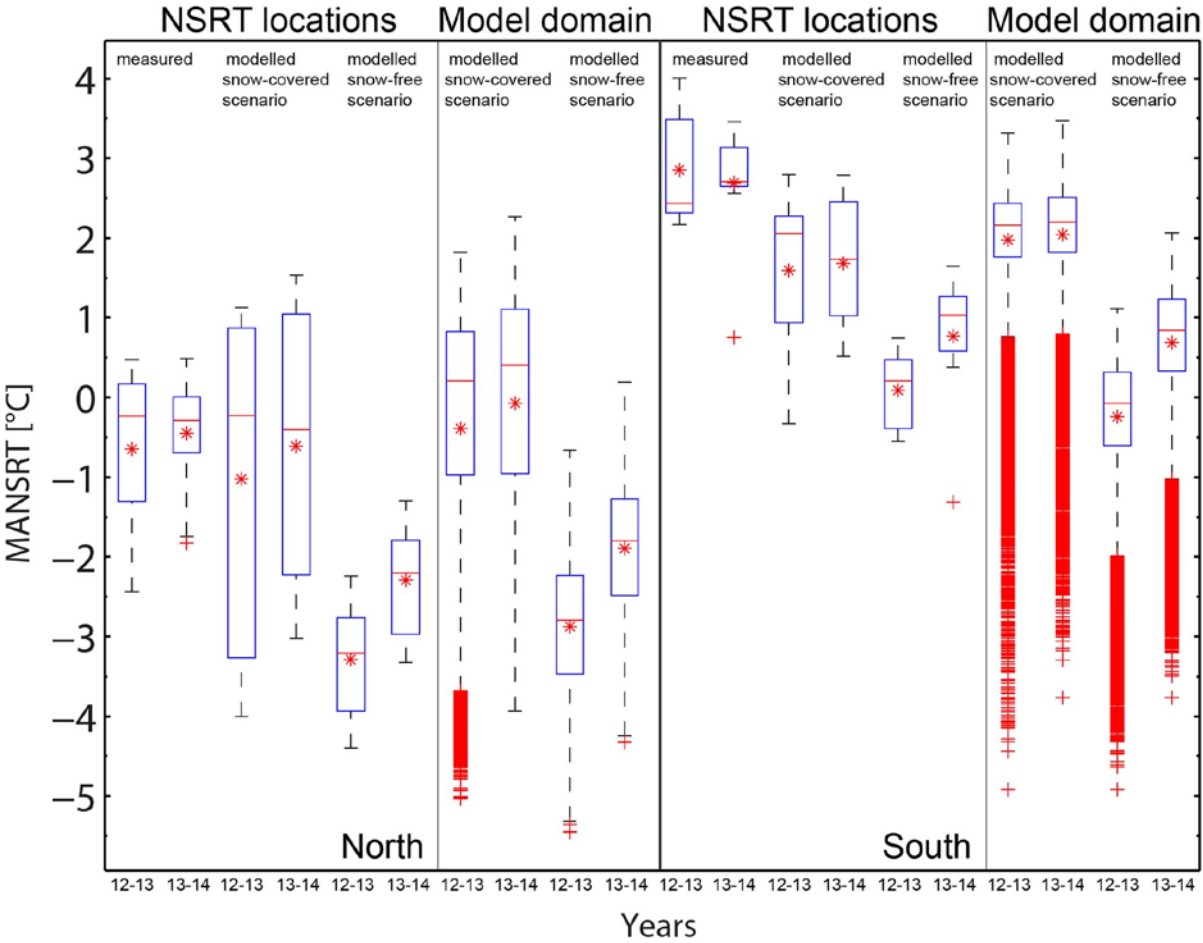

**Figure 9. MANSRT variability within the individual N (left) and S (right) facing rock walls for the years 2012-2013 (12-13) and 2013-2014 (13-14). The MANSRT variability in the rock walls were based on 22 measured NSRTs, 11 facing N and 11 facing S. Measured MANSRT variabilities are compared to modelled MANSRT differences calculated at the grid cells of NSRT locations, shown for both the snow-covered and the snow-free scenarios. In addition to the MANSRT differences calculated at all 22 NSRT locations, the modelled MANSRT variability of each grid cell of the entire model domain is shown, depending on whether the grid cell is N or S facing. The median is marked with a red horizontal line in each box, the mean is additionally plotted as a red asterix, the box edges are the 25th and the 75th percentiles, the whiskers extend to the 2.5 % and 97.5 % quantiles and outliers are plotted as individual crosses.**

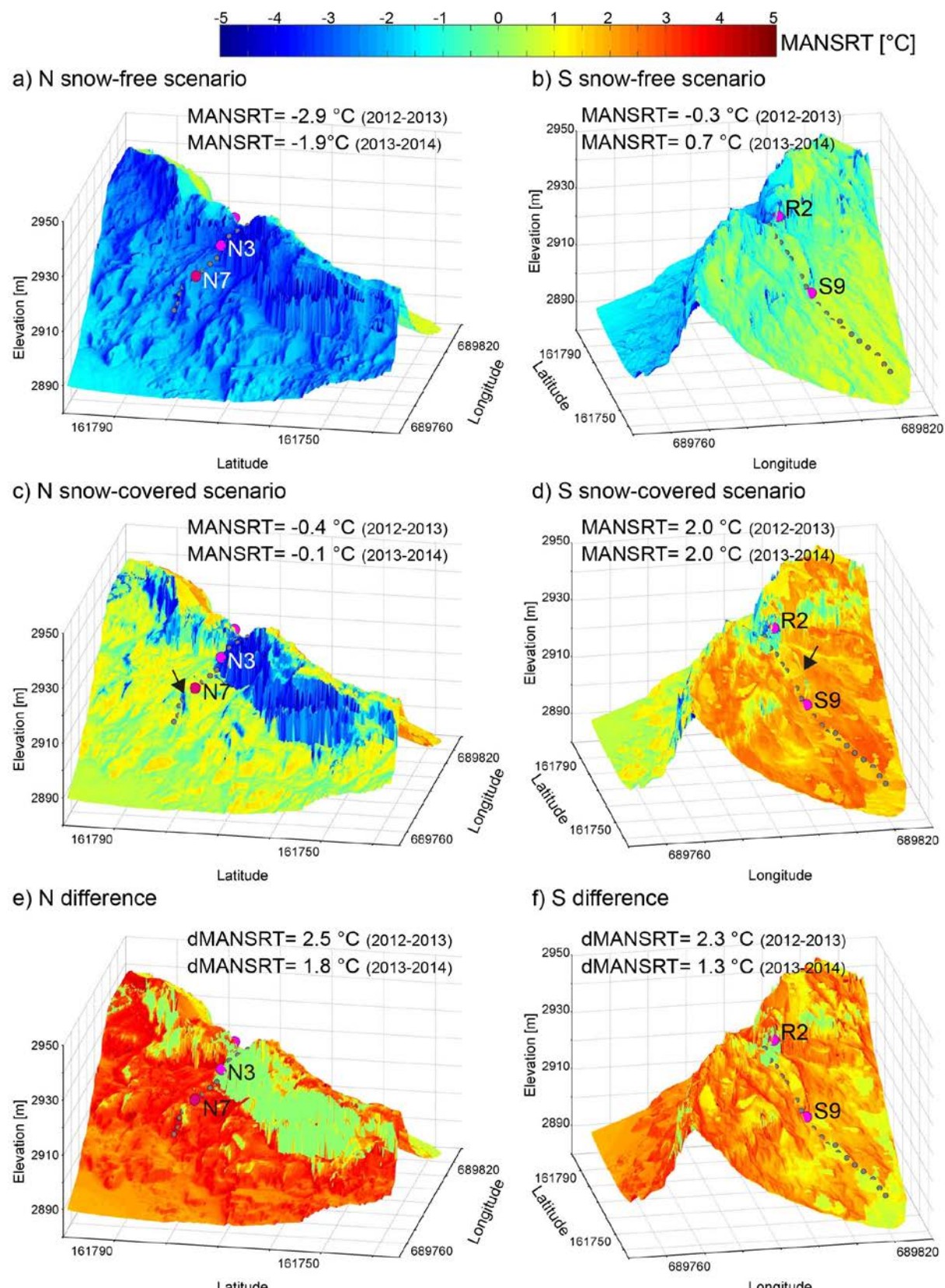

**Figure 10. Modelled MANSRT distribution in the N (left) and the S (right) facing slopes for the snow-free scenario (top) and the snow-covered one (middle), as well as their differences (bottom; snow-covered – snow-free). Arrows indicate rock outcrops and rock dihedrals partly shadowing the NSRT locations, which are marked by grey dots (selected locations in pink and labelled). The model results are only shown**

for the year 2012-2013, but MANSRT averaged over the individual N respectively S facing rock walls are given for both study years, as well as the difference between the MANSRTs of the snow-covered and snow-free scenarios (dMANSRT).

**Appendix: Table 1A. Slope angle (slope) and aspect [both in °] measured at the 22 NSRT locations, as well as their topography in the model domain with varying grid cell size.**

| | Measured | | Cell size 0.2 m | | Cell size 1 m | | Cell size 5 m | |
|---|---|---|---|---|---|---|---|---|
| Location | slope | aspect | slope | aspect | slope | aspect | slope | aspect |
| N1 | 34 | 4 | 53 | 9 | 52 | 8 | 28 | 288 |
| N2 | 47 | 23 | 53 | 6 | 66 | 341 | 50 | 309 |
| N3 | 90 | 284 | 83 | 281 | 70 | 288 | | |
| N4 | 84 | 296 | 36 | 264 | 62 | 282 | | |
| N5 | 72 | 226 | 55 | 250 | 57 | 284 | | |
| N6 | 68 | 324 | 75 | 288 | 61 | 266 | 52 | 289 |
| N7 | 90 | 289 | 69 | 267 | 59 | 268 | | |
| N8 | 74 | 204 | 56 | 228 | 44 | 282 | 56 | 292 |
| N9 | 80 | 340 | 77 | 313 | 68 | 303 | | |
| N10 | 81 | 289 | 80 | 280 | 69 | 286 | 53 | 282 |
| N11 | 89 | 349 | 75 | 323 | 69 | 289 | | |
| S1 | 40 | 132 | 42 | 138 | 5 | 189 | 11 | 124 |
| S2 | 67 | 173 | 67 | 167 | 59 | 160 | | |
| S3 | 79 | 147 | 65 | 142 | 62 | 138 | 41 | 140 |
| S4 | 60 | 122 | 55 | 125 | 58 | 124 | 57 | 143 |
| S5 | 50 | 125 | 62 | 127 | 59 | 130 | | |
| S8 | 57 | 132 | 64 | 143 | 64 | 146 | 55 | 146 |
| S9 | 72 | 165 | 50 | 161 | 61 | 158 | | |
| S10 | 39 | 128 | 38 | 143 | 41 | 161 | 52 | 146 |
| S11 | 42 | 139 | 38 | 146 | 42 | 157 | 48 | 154 |
| S15 | 53 | 184 | 51 | 184 | 64 | 162 | 43 | 158 |
| R2 | 58 | 164 | 64 | 153 | 70 | 151 | 18 | 186 |