# Peer review of "Distributed snow and rock temperature modelling in steep rock walls using Alpine3D"

_The Cryosphere, 2016_

## Referee Comment (RC1) · Anonymous Referee #1 · 27 May 2016

The paper Distributed snow and rock temperature modelling in steep rock walls using Alpine3D from A. Haberkorn and her team presents a 3D modelling method to describe the small-scale patterns of snow accumulation over a steep rock ridge in the Swiss Alps and its effect on the rock wall surface temperature. The study focuses at the measurement point scale and the ridge scale, using a high resolution data set and model domain. The topic is of high interest for the scientific community investigating the distribution and changes of steep slope permafrost in alpine environment. The investigation of the snow control on steep slope permafrost, especially in 3D, is one of the current research challenges. This study therefore tackles a very challenging topic, which is well aligned with the current research directions and previous work from the team.

The methodological approach is sound and achieved. Analysis of the small-scale snow

accumulation patterns and surface temperature variability in steep slope requires high spatial resolution. The presented study uses a high spatial resolution dataset with a 0.2 m DEM built upon TLS data, snow height data based on TLS surveys and rock wall temperature measurements at the surface. It proposes a scaling method to approximate snow accumulation from snow height records at a close weather station, and a 3D energy balance model. The method combination is relevant and field data are substantial, but after several readings, it is still difficult for the reader to have a clear overview. Some suggestions are given to improve the clarity of the method imbrication.

The paper is structured into seven parts further divided into subchapters. The structure is clear and appropriate to present the study. The overall paper is well written but some sections remain hard to follow because many details are given, and the way that the results are partitioned sometimes lead to confusions. As a result, it is very difficult to draw the outlines and retrace the main results at the end. The results are of high interest for the research field, but it is difficult for the reader to point out the most important findings and the main outreaches of the paper. The abstract is relatively poor given the high interest of the results; the main outcomes are not highlighted as much as they would deserve. Some major findings could be better emphasized in the result sections, but also in the abstract and conclusion. The conclusion does not seem to report the main outlines of the paper. In the same way, it is difficult to understand the key-objectives of the study: the validation of Alpine3D? The characterisation of the snow accumulation patterns over steep rock ridges? The characterisation of the small-scale snow control? Possibly this paper tackles all of these objectives, but they should be better outlined and better addressed. Detailing the specific research questions addressed in the paper would possibly help. The broad scientific context is not clear either: is this paper having outreaches in the climate change researches? In the cryosphere distribution investigations? For geomorphological studies? Being clearer with the global goals and results of the paper will ensure a greater impact of the results and will give the paper more visibility. Some suggestions are also given to help in improving the clarity of the paper. Due to the wealth of data and methods, also due

to the possible outreaches and significance of the results for the scientific community, this paper definitely deserves publication in TC. However, it is important that the reader gain an overview of the main findings both for the global scientific context and for the research questions specifically addressed in the study, as well as a clear outline of the methods, related results and limits. It thus appear necessary to improve the text, at least in a first step since it is difficult to retrace the main interests and limits of the paper at the current stage. I recommend this paper for publication in TC after major revisions.

MAJOR REVISIONS

1. I suggest writing the abstract again: first, state what is the global scientific context and specific objectives of the paper. Improve the highlights of your main findings in the results paragraph. Try to be more quantitative if possible The results provided at lines 20-21 seem to contradict previous statements from former studies (e.g. Hasler et al. 2011 found a cooling effect of snow in the sun-exposed rock walls such as mentioned line 81), which is of high interested for the scientific community. This contradiction with current theory might deserve more developments, at least a few words in the abstract and some more in the results/discussion/conclusion to explain why the here presented findings differ from the previous ones (matter of snow height? Snow timing?).

2. Improve the presentation of the methods with (1) a specific figure and (2) an introductory section to explain in a very short paragraph how the methodological approaches, the various spatial resolutions from the different approaches, the characteristics of the input and output data are imbricated. The methodological approach is very complete and involves many steps with various sources of data and computing steps at various time and space scale. After several readings, it is still difficult to gain an overview of the imbrication of data and processing. To improve the visibility and understanding of the method outlines, I suggest preparing a specific figure to sum up the imbrication of the input/output data and processing (e.g. similar and maybe slightly more detailed than the Figures 2 from Noetzli et al. 2007; Figures 1 from Noetzli and Gruber 2009 or from

Fiddes et al., 2015). Also, it would be very helpful to have one or a few introductory sentences for chapter 3 to sum up how the methods and data are imbricated.

3. Improve the presentation of the results and their discussion. In a similar way than the methods, an introductory paragraph to sum up your approach and clearly explain the outlines of the result chapter would be really helpful given the high number of steps in the result presentation. Some general suggestions and comments are given here below, but more specific comments are given in the appropriate section. When simulating at high spatial resolution, the sources of uncertainty are many, and this result in sometimes important bias. Those sources of bias are not always well discussed (e.g. why the model is more performant on the N than on the S face?), whereas some discussion points seem disconnected from the study (why to mention the effects of water percolation along fractures? Where is the link with your study?). Also, to help in understanding the sources of errors, it seems important to compare topographical characteristics of your measurement points used for model evaluation in the "real-world" and those in the "numerical environment" (DEM). Do the sensors have same aspect in both situations? Same slope angle? This information could be added in Table 1 for instance. In case of substantial discrepancies between both environments, this could explain a part of the bias. Results are sometimes hard to follow due to the numerous back-and-forth between figures and the text. You sometimes refer to several figures for a same thing, and not all references are relevant (e.g line 336: you refer to Fig. 3b to show the difference between measured and modelled MANRST, whereas Figure 3 shows daily variation). Figure 3a is not referred in the text. The data on which the MBE and $R^2$ are calculated not always clearly indicated. Many confusions are arising and being more precise would help the reader to go straight to the point. Section 4.1.3. must be written more clearly, at that stage, it is hard to follow. It contains lots of essential information but some details are missing to well understand how the model evaluation is performed, how the misfit between measured and modelled value are taken into account to go further in the study (see detailed comments). Finally, the study seems to contradict previous findings. So far, it was suspected that snow on South faces cools

the surface temperature (e.g. Hasler et al., 2011). In this study, the opposite is stated, and the contradiction is not well discussed, nor well emphasized. What would be the possible factors/processes explaining that your findings are in contradiction with previous findings? Also, Figure 6 which shows an important part of the results is poorly discussed. By looking at this figure it clearly appear that vertical faces without snow induce colder conditions than snow covered slopes. This is well aligned with recent findings in Norway (Myhra et al., 2015) and should be better emphasized and discuss.

GENERAL COMMENTS

1. The references to the existing literature are not always consistent with the text. Some examples of inconsistencies between the text and the references are given in the specific comments, but not all of them. Please, consider this comment and verify your references all along the text.

2. Introduction: 1st paragraph is poorly written. First two sentences focus on rock wall permafrost (with a strange way to use references) whereas the two other sentences, apparently aligned with the first two sentence mention the need to model permafrost with example from very different alpine permafrost terrains, that are not relevant to address the questions related to "rock wall permafrost". This must be improved to be more consistent and to better settle your study in its global research field.

3. The study site is made of a NW and SSE faces (according to Table 1) named N and S face. Whilst naming N and S face is not a problem, it seems that these slopes are considered as real N and S facing slopes in the study (e.g. the apparently unexpected low difference in surface temperature, which is maybe not as low as suggested given the real aspect of the slopes). During revision, this should be taken into consideration to avoid scientific imprecision and straightforward conclusions.   SPECIFIC AND TECHNICAL COMMENTS:

- Lines 20-21: is this sentence written in proper English? It seems confusing.

- Line 29: what does "large" mean? Some rock falls affected "narrow" rock faces, pinnacles, ridges... Is this word really appropriate?

- Line 31: Davies et al. 2001 didn't not investigate the stability of permafrost in high Alpine regions but proposed a laboratory study under very specific conditions. Gruber et al. 2004a didn't not investigate rock wall stability, but only permafrost distribution. Are these references really appropriate? - Line 35: the reference to Gruber, 2012 doesn't seem appropriate since the sentence focus on permafrost modelling in the European Alps and Gruber's work focused on global models.

- Line 36: there are better examples than Fiddes et al. 2015 as numerical modelling of mountain permafrost (especially in rock walls).

- Line 41: Harris et al. 2009 paper does not focus on modelling transient changes in rock wall permafrost. Here again, better examples could be provided (e.g. only keep Noetzli et al. 2007 and move it at the end of the sentence, other examples could be added: e.g. Noetzli and Gruber 2009).

- Line 46: "However this approach cannot capture..." Is it really because of the modelling approach that the small scale variability cannot be captured or because of the spatial resolution?

- Line 59: Gruber et al., 2004b and Gruber and Haeberli 2007 didn't really study the snow control. The last reference, proposed some theories and hypotheses about the snow control, but not a study dedicated to its effect.

- Line 63: Pogliotti, 2011 focused on the snow control in steep rock faces similarly to the here presented study, but in 1D. He only proposed a review of the existing literature stating ablation processes in steep alpine rock faces, but did not study the gravitational processes directly such as suggested by this reference.

- Line 65: Gruber et al. 2004a study considered ideal rock walls, not the kind of "natural" rock walls described in the text before the reference.

[Figure]

- Line 82: is "However," really the right term? It connects the starting sentence with the former sentence in the sense of "Nevertheless", and opposes the new sentence to previous statement. But the smoothed temperature difference between N and S face results of the warming/cooling effect of snow, it is a consequence. Could you consider this and revise your sentence accordingly to avoid confusion? Maybe there is an opposition between two sentences but it is not clear when reading.

- Line 82-84: References are not consistent: do you mean that thick snow smoothes the variability of MAGST compared to snow free bedrock (Gruber et al., 2004b; Noetzli et al., 2007) or compared to bedrock with thin and intermittent snow (Hasler et al., 2011)? The sentence has to be more precise and the references better used.

- Line 89: What is the difference between NRST and the "rock thermal regime"? Do you mean the thermal regime at depth?

- Lines 118-119: could you explain why did you choose this reference period? Data availability?

- Lines 123-125 could you at least tell when the iButtons were installed in order that the reader doesn't have to look for essential information into the referred paper.

- Line 127: here also I don't understand the meaning of "However,".

- Lines 131-132 and 135-136: could you be more precise with the features that you describe? What is the difference in temperature amplitude between N7 and N3? How do you see the snow influence on S9?

- Lines 183-184: it is difficult to understand the end of the sentence: "hence a constant upward ground heat flux is applied as the lower boundary condition". Please, could you reformulate and be more precise?

- Lines 190: could you give an indication of the gap proportion in the meteorological data and of the bias induced by the gap filling procedure (even if information also exists in Haberkorn et al., 2015b)? Does the gap filling procedure induce a part of the bias in

model results?

- Lines 200-201: the reason for which the thermal parameters, especially those at depth (such as 100% solid content which is unusual in modelling rock wall thermal regime) have been chosen is not clear. The utility of these parameters to simulate rock wall surface temperature is not clear either. Could you be more precise about this points?

- Line 233: Wouldn't be "one" instead of "an" in "an Alpine3D run"?

- Lines 234-235: could you provide a concise overview of the results for the three other TLS. What "coincided best with validation data" means quantitatively?

- Lines 278-280: it is not really easy to report the mentioned results to the figure. On which data are the $R^2$ and MBE calculated? You report to figure 2b and c to compare snow heights measured with TLS and modelled with Alpine3D, but those figures only show the measured snow depth. Also, a scatter plot would help the reader to better see the comparison between modelled and measured values.

- Line 284: "for each NRST logger": it is only 4 loggers, right? Why other NRST loggers were not used (except that those used are enough to represent snow cover variability according to lines 127-128)? One could easily think that using more loggers could provide more robustness to the MBE and $R^2$ analysis.

- Lines 288-292: this paragraph is not clear either. "four independent TLS", but one of them was used to scale the snow accumulation (11.12.2013), right? So, is it true to say "independent". "$R^2$=0.95": which data were used for this calculation: modelled versus measured snow height for each grid cells and for each TLS survey? Why to show results from 11.12.13 if those data are used for scaling (and are therefore not independent)? Could you show a scatter plot or at least better illustrate the model output by e.g. replacing one of the 3D or 2D view in Figure 2?

- Line 296: Is the term "validation" really appropriate?

- Lines 299-301: here you give a reason for misfit between modelled and measured

values. The same explanation could be expected lines 290-292: where the under/over-estimations are coming from? Modelling of ablation? If it is given in the discussion, the same should be done for these lines 299-301. If you make the choice to directly discuss your results, an explanation could be expected lines 290-292.

- Lines 300-301: Here the modelled snow depth for the S measured point does not fit the measured values. A 1 m difference may have huge implications for the NRST simulations. How is that taken into account?

- Lines 342-344: why such a big bias (-2°C)? What is its implication in the overall results?

- Lines 362-364: the difference is calculated using the 30 NRST time series?

- Lines 363-364: not as high as expected for "real" North and South walls, which is not the case here, with rather NW and SE faces. This must be taken into consideration!

- Lines 392-399: figure 6 deserves much more description and precision. Could you be more precise in the text with the 1.9°C? For what? Entire model domain? North face.? South face? Measurement point controls?

- Lines 425ff: here the energy balance of snow free N7 is presented like absolutely different from snow covered N7 ("In contrast to", "differed strongly") . However, when looking at Figure 7, the pattern of Qnet seems quite similar, only the magnitude differs, Qsensible differs in a certain degree. Are the terms really appropriate?

- Line 367: "effects: In" is either "effects. In" or "effects: in"

- Line 447: why not modelling heat transfers in fractures is a limit of your model? Are you also modelling the interior of the ridge? If not, please remove, there are already enough details to discuss. If yes, it should appear clearly all along the text that you do not only model surface temperature and substantial results on the model temperature at depth must be provided!

[Figure]

- Line 450: the consideration of snow cover at the ground surface is especially important to model small scale temperature variability. Some studies have shown that equilibrium temperature fields and long-term changes can mainly consider air temperature and solar radiation in steep slope. Please, rework the sentence accordingly.

- Lines 456-457: the statement is interesting (it appear more important to correctly model snow timing to better represent snow effect) but could you at least provide one example in order to help the reader to connect this discussion to the results?

- Line 461: again, the references are not adapted to the text: Gruber et al., 2004 and Noetzli et al., 2007 do not propose a "traditional snow modelling technique".

- Lines 485-486: this belongs to the results, so move in another section and connect it with the presented results. On which source of data is this calculated? What is the difference with other presented MBE (e.g. MBE of -2°C line 342)?

- Lines 504-505: Isn't the difference of snow free/snow covered faces between N/S aspects in the range of model uncertainty?

- Lines 515-520: these appear as important results that would confirm recent findings in Scandinavian rock walls (Myhra et al., 2015): rock walls favour the presence of permafrost (here in the Alps, that would be especially true for North slopes?). This must be better emphasized.

- Lines 541-542: reaching that stage of the paper, the use of 30 NRST logger is still not clear: where the validation is shown? In figures, only 4 loggers are used and discussed. Same question as previously: is "validation" really appropriate?

- Line 553: "50°", how this threshold has been defined? It appears for the first time in the conclusion.

- Line 554: is "accurately" really appropriate when significant bias have been displayed?

- Lines 569-571: this is an interesting result but it has only be mentioned in the discussion. No quantitative information nor graphical results are provided for such statement. Either remove from the conclusion and remain as close as possible of your major findings, or develop the results related to grid-scale sensitivity analysis.

FIGURE AND TABLE

- Figure 1c: what are the peaks between 2930-2950 and 161750-161780 on the y and x axis respectively? They look like artefacts in the DEM. How did you clean up the points cloud before generating the DEM? Furthermore, I the figure could be improved by including a hillshade below the elevation colour scale to improve the visibility of micro-topography.

- Figure 2a: it is very difficult to read the legend, could you make it bigger?

- Figure 3: This figure must be improved. I propose the following modification for better clarity and readability. The legend: measured NRST and the measured-modelled NRST have the same line colours. Make different colour. Some lines are dashed or dotted but this does not appear in the legendOf course, the reader can then easily find out which line in the legend corresponds to which line in the graph, but it is confusing at first glance and does not support rapid overview of the Figure: make the legend consistent with the line style. The measured-modelled NRST is not shown at an appropriate scale. Why not displaying these differences in independent plots below the model output?

---

## Referee Comment (RC2) · Anonymous Referee #2 · 10 Jun 2016

**GENERAL COMMENTS**

This paper presents the spatially distributed application of the physical based model Alpine3D and its snow module SNOWPACK to the small-scale simulation of snow cover patterns and rock surface temperatures in two rugged, steep rock walls on the Gemsstock ridge in the central Swiss Alps. The topic is of high-interest for the scientific community studying mountain permafrost. In fact the distribution, persistence and consequently the thermal effect of snow cover in steep rock walls is poor known and its modeling is challenging due to the scarcity of field observations and the incapacity of the existing models to reproduce wind and gravitational transport of snow in steep topography. The field dataset used in this study has, in my opinion a very high potential. It consists of 30 rock surface temperatures aligned over a ridge cross-section, a 0.2m DEM derived from terrestrial laser scanner (TLS), 4 snow-depth maps derived from

TLS winter campaigns, meteo and snow-depth data from a near automatic weather station (AWS). The study spans two years with 2 complete winter seasons.

Despite these excellent premises, the objectives of the work are not well defined and it is difficult to understand what are the main results. The exposition is very fragmented, each system component (eg. Measured snow, modeled snow, measured temperature, modeled temperature,...) is treated separately and is very difficult to gain an overall view and draw more general findings and conclusions. In the discussion there is scarce attention in the references to plots and tables and some errors have been reported. Some speculative sentences have been reported in the conclusions. From the technical point of view I believe that, if one of the goal of this work is to reproduce carefully the thickness of snow on the rock wall (to assess the effect on surface temperatures), then the adopted precipitation scaling is not appropriate since it generates errors exceeding 0.5m which have huge effects on modeled temperatures. Finally, there are serious deficiencies in the use of references as well as in the choice of the statistics (R2 and MBE) used to evaluate the model performance. All these topics are further explained in the Specific Comments section below.

In conclusion I believe that this work has the potential for providing very important results for the scientific community but a big work of revision and reprocessing must be done before publication on TC. Major revisions.

**SPECIFIC COMMENTS (MAJOR REVISIONS)**

1. The approximate use of technical terms as well as of references (often totally wrong!) denotes the scarce attention paid by writing the introduction chapter. I suggest to the authors to deeply review this chapter by checking carefully the references (all along the paper!).

2. Sections 3.3.1 and 3.3.2 can be merged and shortened (mainly 3.3.1) by providing less detail about Alpine3D and SNOWPACK that are well known and documented models. 3. The precipitation scaling is a very promising idea but it does not seem to work very well as it is. It would be very interesting to understand why in 3 of the TLS campaign does not work providing quantitative analysis of these discrepancies (see technical comment). Moreover, looking at figure 2 is evident that it works quite well on the validation point N7 but is scarce at point S9. In my opinion a simple ratio between AWS snow-depth and TLS snow-depth is a too simplistic approach and represents the main limitation of the present study. I suggest the authors to put together all the TLS campaign data and AWS snow-depth data and try a more complex statistical approach which includes at least the topographical characteristics (ele, slp, asp) and doy (day of year) of the cells as scaling predictors. A first attempt could be to build a linear model with all the predictors, run a stepAIC on it for selecting the significant ones and use the resulting regressive model to scale the precipitation.

4. In my opinion the sections 3.3.4 and 4.4 are totally disconnected from other chapters, not in terms of concepts (energy balance is fundamental) but in terms of contents and argumentations. There are no links or references to what observed or discussed in the other sections, there is no think over possible source of modeling uncertainty, is just a chronicle on the course of each component along the seasons. I suggest the authors to remove these chapters, due to the already high number of data and elements to discuss. As it is, the energy balance discussion looks a digression that distracts the reader from the main subject of the paper. Alternatively the section 4.4 must be deeply reworked in order to provide precise evidence of what is discussed in the section 5.1 (Lines 471-484).

5. Section 4.1.1. The description of the measured snow cover variability by TLS is interesting but useless for the purpose of the paper and has scarce relevance for the scientific community because is too detailed and site-specific. It lacks effort to outline most general patterns of snow accumulation in steep rock walls. It would be very interesting to explore if in your dataset exists a relationship between snow-depth-TLS and steepness of the grid-cell. This analysis might be, I guess for the first time, a real mea-

СЗ

sure of snow-depth in steep rock walls and provide the community some indications on the snow-depth thresholds to use for modeling experiments in steep rock walls. At first, this analysis (i) could exclude the cells above ledges and (ii) could analyze NW and SE faces separately.

6. Section 4.1.2. The statistics provided (R2 and MBE) are not sufficient. R2 indicates the fraction of variability (variance) in the observation that is explained by the model. Used alone it says little about model performance in strict sense because e.g. in case of temperature you can have an R2=0.99 with 10°C of bias. The modeling efficiency (ME) must be used also. MBE describes the direction of the error bias. Its value is related to the magnitude of values under investigation. A negative MBE occurs when predictions are greater in value than observations, positive MBE occurs when predictions are greater in value than observations. In case of snow-detph has no sense to provide a mean value of MBE (-0.002 m!!) over the entire model domain because over- and under- estimations vanish each other. Mean absolute error (MAE) or root mean square error (RMSE) must be used instead. Also error bars in Fig.2 look strange, see technical comments. I suggest this paper for further detail: Mayer, D., and D. Butler (1993), Statistical validation, Ecological modelling, 68(1-2), 21–32.

7. Section 4.1.3. If one of the objective of this paper is (accurately) simulate the influence of snow cover on NSRT in steep rock walls I guess that differences in the order of 0.5 - 1m between observed and modeled snow depth is too much for obvious reasons. To reduce this uncertainty, as said in specific comments n.3, the precipitation scaling must be totally revised.

8. Section 4.2.2. This section would be a validation of NSRT but is very poor under this point of view. The absence of statistical metrics to evaluate model performance is evident here (see general comments). The description of discrepancies between obs. and mod. is only qualitative, comments are limited to temperature without any reference to the modeled snow which is the main constraining factor. In particular, observing together Fig.2a and Fig.3 results that temperature modeling has better performance where snow modeling has worst performance (point S9). Nothing is said about that. This section, that potentially could be the core of the paper, must be strongly improved.

9. Section 4.2.3. The idea of a run with forced snow-free condition is good but results are not exploited at all. This run could be used as reference to quantify the potential thermal effect of snow cover at different slope and aspect (see Pogliotti, 2011). This is a way to generalize the results and valorize the dry run. Of course, the precipitation scaling must be improved before (specific comments 2).

**TECHNICAL CORRECTIONS**

- Line 29: the term "rock avalanche" refers to big falls of earth material (of up to millions of metric tons) able to reach velocities of more than 50 meters per second and leave a long trail of destruction. In the Alps such phenomena are not "numerous" (e.g. Val Pola 1987, Tschierva 1988, Brenva 1997, Thurwieser 2005) and even less those where permafrost can be directly listed among the trigger factors. The right term is "rock falls".

- Line 30: strange references, Gruber & Haeberli 2007 is better and more comprehensive than Gruber 2004b, e.g. Fisher 2012 (Nat. Hazards Earth Syst. Sci) is missing.

- Line 31: Davies et. al 2001 is wrong! Gruber et. Al 2004a is wrong! Fisher 2012 (Nat. Hazards Earth Syst. Sci) is more appropriate than Fisher 2006, Gruber & Haeberli 2007 is missing, Allen & Huggel 2013 (Glob. and Planetary Change) is missing, Saas 2012 (Nat. Hazards Earth Syst. Sci) is missing, Deline et al. 2015 (Snow and Ice-Related Hazards, Risks, and Disasters, chapter 15) is missing... and many more.

- Line 35: Gruber 2012 is wrong! e.g. Guglielmin 2003 (Geomorphology) is missing

- Line 36: if you cite only Fiddes et al. 2015 add "e.g." because exist more

- Line 37: kilometers

- Line 41: transient changes... Harris et al. 2009 alone has no sense because is a big

state-of-the-art of mostly all fields of research around mountain permafrost... Noetzli & Gruber 2009??

- Line 46-49: ...cannot capture... the ground thermal regime. I'm not sure of that. The Fiddes 2015 approach has not been yet validated against field measures.

- Line 56: remove "However"

- Line 56-58: this statement is too strong and do not consider that the temperature of a point in depth integrates the contribution of a certain area at surface. This area is wider as deeper is the point so the effect you are talking about is probably limited to few meters. Thus, in my opinion, to investigate the 3D subsurface heat flow is not necessary to reproduce surface temperatures with so-high spatial resolution. Please, reformulate this sentence considering also these aspect.

- Line 59-60: Gruber 2004 is wrong!, Gruber & Haberli 2007 is a kind of review and snow control only is mentioned, remove it. Pogliotti 2011 is probably the first work that systematically investigate the thermal effect of snow cover (moreover with high affinity with the present work) even in steep rock walls and is missing. Magnin 2015, Haberkorn 2015a & 2015b are missing too!

- Line 63: Pogliotti 2011 is wrong!
- Line 65: Gruber et al. 2004A is wrong!
- Lines 82-85: this sentence is not clear, explain better.
- Line 106: elevation range must be explicit in the site description.
- Line 127: Remove However. In this study, only data from...

- Lines 130-136: what you describe here is not evident neither from figure 1 nor from table 1 but just in figure 3. If you don't show a plot you have to describe better the differences you observe in the temperature fluctuations in order to justify your choices.

- Lines 191-194: the initialization is important. Provide here, synthetically, more details about initialization without reference to another paper. Is not clear as it is.

- Line 205: remove high resolution

- Line 211: Uncertainties in modeling...

- Line 213: R2 is the coefficient of determination! MBE is not the right statistic in this case, look at specific comments.

- Lines 209-213: move this paragraph as preamble of chapter 4.

- Lines 216-218: remove.

- Lines 222-224: what is the "snow depth driving mode" of snowpack? Something that convert snowfall in liquid precipitation? By which snow density value? This is a key step of your precipitation scaling, please explicit all the detail, synthetically, without references to other papers.

- Lines 225: "integrated" seems a mathematical term, please use a synonym.

- Line 228: replace "onto the DEM" with "in each grid cell".

- Lines 228-232: replace this sentence with "cells where TLS data were non available have been excluded from the analysis".

- Line 233: TLS campaign.

- Lines 233-241: explain better why you choose only the TLS of December 2013 for driving the precipitation scaling and provide quantitative proofs for this choice (model performance on modeled vs. observed NSRT). Look also specific comments.

- Line 247: see specific comments 4.

- Line 262: see specific comments 5.

- Line 277: see specific comments 6.

- Line 279: MBE = -0.002 m has no sense. MBE is the wrong statistic in this case (see specific comments).

- Lines 282-283: explain the method used for calculating the error bars and exactly what they represent. Is not clear. How can I have an error bar of  $\pm$ 0.3m and a difference obs./mod. (red dot, red line) of about 1 m?

- Lines 300-301: explain/explore better the reasons of such a huge difference in S9.

- Line 287: see specific comments 7.

- Line 334: what does it means "auspicious accordance"? please try to be more adjective

- Line 335: MBE is the wrong statistic in this case (see specific comments).

- Line 330: see specific comments 8.

- Line 346: see specific comments 9.

- Lines 363-364: this sentence is ambiguous, what does it means "not pronounced as expected"? Expected for N/S differences (?) this is not the real case. Expected for snow-free, steep, conditions(?) this is not the real case. If you average all the measures of a mountain side like the yours, the value you got is exactly what I expected.

- Line 366: remove "compensating"

- Line 367: remove "In 2013-2014"

- Lines 367-370: respect the colon, merge these two sentences in one

- Lines 374-376: the higher SD of modeled temperatures derives essentially by the scarce ability in reproduce real (in terms of thickness) snow cover conditions on both sides.

- Line 378: how can you say that underestimation is mainly in summer? (fig. 3?). Explicit.

- Lines 379-380: remove "therefore", this sentence is not a direct consequence of what you said before, or only partially. This is a comparison with the 3.6°C stated at line 363. Contextualize better this sentence.

- Line 384: compared to what? Modeled or real snow covered conditions? It is very difficult to follow your reasoning looking at Table 3 because the number in the text are often means of values in different columns of the table and moreover rounded! If you need these numbers add columns in the table!

- Lines 383-390: rework this section in accordance with the previous comment. Consider also the specific comments n.9

- Lines 392-399: very poor description. Provide more details or remove this section, figure 6 and the "grid" lines in table 3.

- Line 401: see specific comments 4.

- Line 447: modeling of water flow within fractures is not relevant for reproducing surface rock temperatures. Also the influence of surface water flow is negligible in comparison to a correct simulation of snow cover thickness.

- Line 451: check the references (see specific comments 1)

- Lines 452: please explicit the value of snow density used (see also technical comment Lines 222-224)

- Line 453: remove "However"

- Lines 454-455: the first half of the sentence (from However to AWS) is obvious thus can be removed, the second half is not clear, explain better this concept of non-linear settling. Include also the sentence after.

- Lines 457-458: this is not evident from your data. Look the table attached (Fig.1) and justify your sentence.

- Line 459-461: is not evident to me. Check the references (see specific comments 1)

- Line 462: what is the "apparent insulation"?

- Lines 465-466: heat flux at the bottom (20m below) cannot be seen in surface in so short simulations!

- Lines 468: remove "While"

- Lines 471-484: this is interesting but is very difficult to see the evidence of what you are saying in the plot 7 as well as find references in the text of section 4.4. See specific comment 4.

- Lines 485-486: move this in the results providing evidence of the source data. Keep in mind specific comments about the use of MBE.

- Lines 489-499: in my opinion this belong to section 5.1. Check the references (see specific comments 1) all along this paragraph.

- Line 500: replace "possibly made" with "introduced"

- Lines 504-505: looking at table 3 the warming effect on MANSRT is up to 3.7°C at N7 (2012-2013) and up to 1.5°C at S9 (2012-2013). Please keep attention and precision in reference to plot and table contents!

- Lines 508-511: this obviously depends on the amount of snow. A persistent thin snow cover has always cooling effect both at N and S faces, while a thick snow cover has warming effect. Thus the reason you observe on average a warming effect of snow cover is because you allow the accumulation of thick snow. If you have a look a other cells with thin snow I'm sure you can observe cooling effect between dry and snow simulation. So change this sentence keeping in mind also these aspect.

- Lines 515-520: this sentence is very interesting but not well introduced nor supported by findings of this paper. Provide more detail, evidence and argumentations in order to support this suggestion.

- Line 524: this section is very interesting and useful for the modeling community, but is poor of numerical evidences. Please, provide a synthetic table (or plot) where the influence of grid resolution on the model performance becomes evident (see also specific comments for assessing model performance in the correct way).

- Lines 551-553: I would say, "the results of the present work help to quantify the potential error..."

- Line 554-556: "Alpine3D simulates near-surface rock temperatures and snow depth in the heterogeneous terrain accurately." in general this is true but is not the case of this work. The reason is that the precipitation scaling procedure is weak and provide unreliable precipitation input to the model. In my opinion this conclusion does not reflect the real result of this work.

- Line 556-558: lateral heat-flux is negligible in comparison to the effect of a bad precipitation input.

- Line 559-561: this is true, the potential of the dataset is very high but the choice of exploring just 2 cells on the N face and 2 cells in the S face strongly constrain this potential. See also general comments.

- Line 562: this sentence on the lateral heat flux is speculative. Nothing in the results provides the basis to verify this statement.

- Lines 569-571: also in that case no numerical evidence about model performance are provided in the results hence this sentence is speculative too.

**FIGURES AND CAPTIONS**

Table 3.

- Caption (Line 812), replace "data" with "cells". How do you identified snow-free cells? Figure 1.

- The boreholes are not considered at all in this work then I suggest to remove it from the figure and caption to avoid confusion.

- I suggest to replace the three colorful elevation plot by a "classic" but more readable cross-section along the logger line which easily can gives the information about elevation and steepness at one-shot. Figure 2.

- Just figures a) and f) are relevant for the interpretation and discussion of the precipitation scaling. Remove figures b) c) d) e) that are not relevant and enlarge figure a).

- The range in figure f) has been constrained at  $\pm 0.5m$  for graphical reasons, but a frequency distribution plot (barplot) of differences on the model domain should be inserted as compendium to provide a comprehensive overview of modeled snow depth uncertainties.

Figure 3.

- Caption: dT are present also in the plots d) and e) not only in b) and c) as stated.

Figure 5.

- The boxplot shows the meadian but in the text and table 3 the references are always to the mean. Please modify the boxplot in accordance with the text.

| sensor | 2012-2013     |         | 2013-2014    |                |
|--------|---------------|---------|--------------|----------------|
|        |  Adays | ΔMANSRT | Adays | AMANSRT |
| N7     | -15           | +1.1    | 6            | -0.1           |
| S9     | -1            | +1.2    | -15          | -0.2           |

Fig. 1.

---

## Author Comment (AC1) · 31 Aug 2016

**Author final responses to Reviewers (Ref. No.: tc-2016-73)**

The authors thank referee #1 for the useful remarks and suggestions. All the referee' comments (left) and our responses to them (right) are listed below.

General comments:

To evaluate the performance of the model applied, model results of snow depth and rock temperatures in 10 cm depth (NSRT) were compared to detailed measured validation data (snow depth of 3 independent TLS, 30 NSRT measurement locations). An error analysis of snow depth and NSRT (MBE, $r^2$) was performed at each of the 30 NSRT locations. While analysing the influence of snow on rock temperatures for the entire N and S facing rock walls, means of all NSRT loggers were calculated. The same is true for the error analysis.

Since it would be too much to show model results and their comparison to measured data, only 4 of the NSRT logger locations were chosen to show in detail. These 4 temperature loggers were chosen in order to represent typical NSRT evolutions depending on whether the location is snow-covered or not. For these 4 locations single error analyses are presented, but the error analysis for the 30 other NSRT loggers are also presented. According to both referees, that was not clear from the manuscript. We therefore will clearly state this.

**Referee 1:**

| Referee comment | Author answer |
|---|---|
| 1. I suggest writing the abstract again: first, state what is the global scientific context and specific objectives of the paper. Improve the highlights of your main findings in the results paragraph. Try to be more quantitative if possible The results provided at lines 20-21 seem to contradict previous statements from former studies (e.g. Hasler et al. 2011 found a cooling effect of snow in the sun-exposed rock walls such as mentioned line 81), which is of high interested for the scientific community. This contradiction with current theory might deserve more developments, at least a few words in the abstract and some more in the results/discussion/conclusion to explain why the here presented findings differ from the previous ones (matter of snow height? Snow timing?). | We will rewrite the abstract after revising the whole manuscript. The research questions and objectives of this study will be clarified. In addition the main results will be clearly emphasized.
The contradiction of the presented results to previous studies (e.g. Hasler et al. 2011; Magnin et al. 2015) will be discussed in more detail in the discussion section, but will not be provided in the abstract. There are still open research questions, which our study does not answer (e.g. thin snow > cooling yes or no). |
| 2. Improve the presentation of the methods | An additional figure in form of a flow chart |

| | |
|---|---|
| with (1) a specific figure and (2) an introductory section to explain in a very short paragraph how the methodological approaches, the various spatial resolutions from the different approaches, the characteristics of the input and output data are imbricated. The methodological approach is very complete and involves many steps with various sources of data and computing steps at various time and space scale. After several readings, it is still difficult to gain an overview of the imbrication of data and processing. To improve the visibility and understanding of the method outlines, I suggest preparing a specific figure to sum up the imbrication of the input/output data and processing (e.g. similar and maybe slightly more detailed than the Figures 2 from Noetzli et al. 2007; Figures 1 from Noetzli and Gruber 2009 or from Fiddes et al., 2015). Also, it would be very helpful to have one or a few introductory sentences for chapter 3 to sum up how the methods and data are imbricated. | will be provided in order to present an overview of the methods/ data used for both driving and validating the model. We will adapt the chronology of the methods section based on this flow chart. In addition to the flow chart we will provide a short introduction in the methods section. |
| 3. Improve the presentation of the results and their discussion. In a similar way than the methods, an introductory paragraph to sum up your approach and clearly explain the outlines of the result chapter would be really helpful given the high number of steps in the result presentation. Some general suggestions and comments are given here below, but more specific comments are given in the appropriate section. When simulating at high spatial resolution, the sources of uncertainty are many, and this result in sometimes important bias. Those sources of bias are not always well discussed (e.g. why the model is more performant on the N than on the S face?), whereas some discussion points seem disconnected from the study (why to mention the effects of water percolation along fractures? Where is the link with your study?). Also, to help in understanding the sources of errors, it seems important to compare topographical characteristics of your | The structure of the paper will be revised. At the end of the introduction we will clearly state the aims and objectives of our study for a better understanding of the whole manuscript. According to this the methods and results section will be reorganised. Some introductory sentences will be provided in the beginning of both the methods and results sections to better lead the reader through the manuscript. It will be clearly distinguished between model results and validation data. Concerning the results section: first modelled and measured snow cover data will be presented since the snow strongly influences the thermal regime of the rock walls. The snow cover section (4.1) will be revised to clarify various confusing points. Fig. 2 will correspondingly be adapted (please see specific comments). In a next step the NSRT data will be discussed. The usage of NSRT data of the 30 |

measurement points used for model evaluation in the "real-world" and those in the "numerical environment" (DEM). Do the sensors have same aspect in both situations? Same slope angle? This information could be added in Table 1 for instance. In case of substantial discrepancies between both environments, this could explain a part of the bias. Results are sometimes hard to follow due to the numerous back-and forth between figures and the text. You sometimes refer to several figures for a same thing, and not all references are relevant (e.g line 336: you refer to Fig. 3b to show the difference between measured and modelled MANRST, whereas Figure 3 shows daily variation). Figure 3a is not referred in the text. The data on which the MBE and R2 are calculated not always clearly indicated. Many confusions are arising and being more precise would help the reader to go straight to the point. Section 4.1.3. must be written more clearly, at that stage, it is hard to follow. It contains lots of essential information but some details are missing to well understand how the model evaluation is performed, how the misfit between measured and modelled value are taken into account to go further in the study (see detailed comments).

Finally, the study seems to contradict previous findings. So far, it was suspected that snow on South faces cools the surface temperature (e.g. Hasler et al., 2011). In this study, the opposite is stated, and the contradiction is not well discussed, nor well emphasized. What would be the possible factors/processes explaining that your findings are in contradiction with previous findings? Also, Figure 6 which shows an important part of the results is poorly discussed. By looking at this figure it clearly appear that vertical faces without snow induce colder conditions than

loggers for model validation will be better explained.

In order to understand differences between modelled and measured NSRT data better, the topographic differences between the validation locations and their location in the model domain will be provided in an additional Table in the appendix.

The references to figures in the results section will be shortened for better reading and only the most appropriate ones will be cited. In addition we will clearly distinguish between results and discussion.

The calculation of the statistics between model and validation data will be addressed in the methods section for a better understanding. Only results of the statistics will then be given in the results section.

The contradiction of results with previous studies (e.g. Hasler et al., 2011) will be discussed in more detail in the discussion section. We will provide an explanation of possible factors leading to this contradiction and will connect our findings to findings in literature of steep rock wall temperatures (e.g. Hasler et al., 2011; Magnin et al., 2015; Myhra et al., 2015). Associated to this we will also emphasize more Fig. 6, which is the core of our modelling study. To do so, we most likely merge section 4.3.4. with the other sections of 4.3.

| | |
|---|---|
| snow covered slopes. This is well aligned with recent findings in Norway (Myhra et al., 2015) and should be better emphasized and discuss. | |
| 4. The references to the existing literature are not always consistent with the text. Some examples of inconsistencies between the text and the references are given in the specific comments, but not all of them. Please, consider this comment and verify your references all along the text. | The references to the literature are often too generalised. Therefore we will revise the references in the introduction and will check references throughout the manuscript. |
| 5. Introduction: 1st paragraph is poorly written. First two sentences focus on rock wall permafrost (with a strange way to use references) whereas the two other sentences, apparently aligned with the first two sentence mention the need to model permafrost with example from very different alpine permafrost terrains, that are not relevant to address the questions related to "rock wall permafrost". This must be improved to be more consistent and to better settle your study in its global research field. | The first paragraph will be rewritten in order to emphasize the need to study (measure and model) rock wall permafrost. The application of modelling mountain permafrost occurrence correctly over large areas, such as the Alps will be introduced elsewhere in the introduction. |
| 6. The study site is made of a NW and SSE faces (according to Table 1) named N and S face. Whilst naming N and S face is not a problem, it seems that these slopes are considered as real N and S facing slopes in the study (e.g. the apparently unexpected low difference in surface temperature, which is maybe not as low as suggested given the real aspect of the slopes). During revision, this should be taken into consideration to avoid scientific imprecision and straightforward conclusions. | An additional table giving aspect and slope measured in 'reality' and in the model domain (based on the DEM) for all 30 NSRT logger locations will be inserted, most likely in the appendix. In addition the interpretation of measured and modelled data will be improved with a special focus on the real aspect (NW/SSE) of the rock walls. |
| 7. Lines 20-21: is this sentence written in proper English? It seems confusing. | The sentence will be rewritten. |
| 8. Line 29: what does "large" mean? Some rock falls affected "narrow" rock faces, pinnacles, ridges... Is this word really appropriate? | 'Large' will be deleted. |
| 9. Line 31: Davies et al. 2001 didn't not investigate the stability of permafrost in high Alpine regions but proposed a laboratory | Davies et al. (2001) and Gruber et al. (2004a) will be deleted. |

| | |
|---|---|
| study under very specific conditions. Gruber et al. 2004a didn't not investigate rock wall stability, but only permafrost distribution. Are these references really appropriate? | |
| 10. Line 35: the reference to Gruber, 2012 doesn't seem appropriate since the sentence focus on permafrost modelling in the European Alps and Gruber's work focused on global models. | Gruber (2012) will be deleted. |
| 11. Line 36: there are better examples than Fiddes et al. 2015 as numerical modelling of mountain permafrost (especially in rock walls). | The reference Fiddes et al. (2015) refers to physics-based modelling of permafrost distribution in the European Alps. We think this reference is appropriate in this context, but of course just one of many examples. Since the first paragraph will be rewritten in order to point out the need to study rock wall permafrost, we will provide other references. |
| 12. Line 41: Harris et al. 2009 paper does not focus on modelling transient changes in rock wall permafrost. Here again, better examples could be provided (e.g. only keep Noetzli et al. 2007 and move it at the end of the sentence, other examples could be added: e.g. Noetzli and Gruber 2009). | Harris et al. (2009) will be deleted. Noetzli et al. (2007) and Noetzli and Gruber (2009) will be moved at the end of the sentence. |
| 13. Line 46: "However this approach cannot capture..." Is it really because of the modelling approach that the small scale variability cannot be captured or because of the spatial resolution? | Fiddes et al. (2015) cannot capture the small scale variability because of too coarse spatial resolution of the approach used. The sentence will be rewritten for a better understanding. |
| 14. Line 59: Gruber et al., 2004b and Gruber and Haeberli 2007 didn't really study the Snow control. The last reference, proposed some theories and hypotheses about the snow control, but not a study dedicated to its effect. | The three references will be replaced with more appropriate references on snow control in steep rock walls, such as Haberkorn et al. (2015a,b), Magnin et al. (2015), Mhyra et. al. (2015) and Pogliotti (2011). |
| 15. Line 63: Pogliotti, 2011 focused on the snow control in steep rock faces similarly to the here presented study, but in 1D. He only proposed a review of the existing literature stating ablation processes in steep alpine rock faces, but did not study the gravitational | Pogliotti (2011) will be deleted. |

| | |
|---|---|
| processes directly such as suggested by this reference. | |
| 16. Line 65: Gruber et al. 2004a study considered ideal rock walls, not the kind of "natural" rock walls described in the text before the reference. | Correct. Gruber et al. (2004a) will be replaced with references to studies dealing with snow in steep rock walls, such as Haberkorn et al. (2015a), Sommer et al. (2015) or Wirz et al. (2011). |
| 17. Line 82: is "However," really the right term? It connects the starting sentence with the former sentence in the sense of "Nevertheless", and opposes the new sentence to previous statement. But the smoothed temperature difference between N and S face results of the warming/cooling effect of snow, it is a consequence. Could you consider this and revise your sentence accordingly to avoid confusion? Maybe there is an opposition between two sentences but it is not clear when reading. | The sentence will be deleted. |
| 18. Line 82-84: References are not consistent: do you mean that thick snow smoothes the variability of MAGST compared to snow free bedrock (Gruber et al., 2004b; Noetzli et al., 2007) or compared to bedrock with thin and intermittent snow (Hasler et al., 2011)? The sentence has to be more precise and the references better used. | Please see answer 17. |
| 19.  Line 89: What is the difference between NRST and the "rock thermal regime"? Do you mean the thermal regime at depth? | The rock thermal regime close to the surface and at depth is meant. We will rewrite the sentence accordingly. |
| 20. Lines 118-119: could you explain why did you choose this reference period? Data availability? | The study period from 1 September 2012 to 31 August 2014 was chosen in order to present 2 years of complete meteorological input, as well as validation data (TLS, NSRT). |
| 21. Lines 123-125 could you at least tell when the iButtons were installed in order that the reader doesn't have to look for essential information into the referred paper. | The iButtons were installed on 9 July 2012. In this study we focus on the investigation period from 1 September 2012 to 31 August 2014, which is stated in section 2. The date of installation of NSRT loggers will not be provided in the text, since it is not relevant for the reader. |
| 22. Line 127: here also I don't understand the | 'However' will be deleted and sentence |

| | |
|---|---|
| meaning of "However,". | rewritten for better understanding. |
| 23. Lines 131-132 and 135-136: could you be more precise with the features that you describe? What is the difference in temperature amplitude between N7 and N3? How do you see the snow influence on S9? | Lines 127-136 will be rewritten in order to describe better the snow/ no snow influence on NSRT. In addition it will be referred to Fig. 3 to better show the features described. |
| 24. Lines 183-184: it is difficult to understand the end of the sentence: "hence a constant upward ground heat flux is applied as the lower boundary condition". Please, could you reformulate and be more precise? | The whole section 3.3 will be shortened (please see also answer 2 to referee 2). Therefore this sentence will be rewritten and the depth, as well as the magnitude of the geothermal heat flux at the lower boundary will be provided. |
| 25. Lines 190: could you give an indication of the gap proportion in the meteorological data and of the bias induced by the gap filling procedure (even if information also exists in Haberkorn et al., 2015b)? Does the gap filling procedure induce a part of the bias in model results? | Data from the AWS Gütsch, maintained by MeteoSwiss, were used for gap filling for all parameters of the meteorological data series of Gemsstock from 22 March to 15 April 2013, as well as for correcting the erroneous ISWR measured at Gemsstock between 1 September 2012 and 15 April 2013. For the corrected ISWR in 2012 the mean absolute error was 14.4 W $m^{-2}$, the mean bias error was 8 W $m^{-2}$ and the root mean squared error was 30.2 W $m^{-2}$. Calculated errors are reasonable, since the radiation sensor accuracy is ±20 W $m^{-2}$.

In order to parameterize ILWR between 1 September 2012 and 15 April 2013, a combination of a clear-sky algorithm developed by Dilley and O'Brien (1998) and a cloud correction algorithm from Unsworth and Monteith (1975) is applied. For the all-sky ILWR the mean absolute error was 26.8 W $m^{-2}$, the mean bias error was −6.3 W $m^{-2}$ and the root mean squared error was 31.9 W $m^{-2}$. Hence, parameterization errors are reasonable compared with the error range suggested e.g. by Flerchinger et al. (2009) for the combination of the Unsworth cloud correction and the Dilley clear-sky algorithm (root mean squared error of 27.1 W $m^{-2}$). Gaps in snow depth data were filled based |

| | on similarity with data from adjacent stations using geostatistical interpolation tools. Snow depth from 10 surrounding IMIS AWS within a distance of 20 km and from AWS Gütsch served as correction data for the snow depth of Gemsstock. Stations are located in flat terrain and cover all directions to consider different air flows at each station. Detrended weighting procedures were applied to account for elevation differences between Gemsstock and the neighbouring stations. |
| | The presented correction methods may are inappropriate to determine ISWR and ILWR exactly at one certain point in time, but are considered to be an acceptable solution for the input of an energy balance model running on a multi-annual timescale where the conservation of natural variability of the model input variables is much more important than the projection of single time steps. We think that gap filling only induces a minor part of the bias in model results and a meteorological error analysis is not within the scope of this study. All this information will therefore not be provided in the manuscript. The correction and error analysis are well documented in Haberkorn et al. (2015b). |
| 26. Lines 200-201: the reason for which the thermal parameters, especially those at depth (such as 100% solid content which is unusual in modelling rock wall thermal regime) have been chosen is not clear. The utility of these parameters to simulate rock wall surface temperature is not clear either. Could you be more precise about this points? | Down to 0.5 m depth 99% solid and 1% pore space containing ice or water was assumed to account for near-surface fracture space. Between 0.5 and 20 m depth 100% un-fractured, solid rock was assumed. Further, it is not the scope of this study to model the influence of fractures on rock temperatures, which we addressed (but did not model) in Phillips et al. (2016). Although the geothermal heat flux is most likely negligible in the narrow and steep Gemsstock ridge, a geothermal heat flux (here: 0.001 W m$^{-2}$) had to be applied as |

| | lower boundary condition of the model. To ensure a marginal impact of this boundary condition on the analysed rock thermal regime close to the surface, it was important to model deep into the rock. We chose to model down to 20 m depth, since detailed rock temperatures for initializing the model were available from an on-site borehole. Model uncertainties resulting from the use of the geothermal heat flux as the lower boundary condition were evaluated in 1d SNOWPACK test simulations at the borehole location at Gemsstock. Here, modelled rock temperatures accord well with borehole rock temperatures measured at various depths down to 15 m for both NE- and SW-facing locations ($r^2$ = 0.6–0.88). Correlation decreases with increasing depth, since modelled temperatures are biased by the geothermal heat flux. Consequently, simulated rock temperatures could be considered to depths of approx. 10 m at Gemsstock. This data however, will be presented elsewhere. The physical properties of the granodiorite bedrock used for ground modelling are discussed in Haberkorn et al. (2015b). For consistency the same bedrock properties are applied. |
|---|---|
| 27. Line 233: Wouldn't be "one" instead of "an" in "an Alpine3D run"? | Will be changed. |
| 28. Lines 234-235: could you provide a concise overview of the results for the three other TLS. What "coincided best with validation data" means quantitatively? | We will provide an additional figure (histogram) to justify the choice of one TLS as precipitation scaling input. Please see also answer 3 to referee 2. |
| 29. Lines 278-280: it is not really easy to report the mentioned results to the figure. On which data are the R2 and MBE calculated? You report to figure 2b and c to compare snow heights measured with TLS and modelled with Alpine3D, but those figures only show the measured snow depth. Also, a scatter plot | The results section 4.1 will be reorganized and rewritten, since it is confusing. Subsections 4.1.1 and 4.1.2 will be merged. Thus the sentences in line 278-280 will be deleted.
In addition Fig. 2 will be reworked in order to provide more meaningful snow depth |

| | |
|---|---|
| would help the reader to better see the comparison between modelled and measured values. | information. Subfigures 2b, c, d, e will be replaced and instead three subfigures each will be shown on: independent TLS data, differences between the independent TLS data and model results at date of the TLS campaigns, as well as scatter plots of measured and modelled snow depth.

For your clarification: the $r^2$ and the MBE were calculated between the measured snow depth (TLS) at 11 December 2013 and the scaled snow depth (precipitation scaling) at the same date. |
| 30. Line 284: "for each NRST logger": it is only 4 loggers, right? Why other NRST loggers were not used (except that those used are enough to represent snow cover variability according to lines 127-128)? One could easily think that using more loggers could provide more robustness to the MBE and R2 analysis. | It is correct that the more validation data the better in order to provide robustness to the statistics applied (MBE, $r^2$). The MBE and $r^2$ error analysis between measured and modelled NSRT data was performed for each of the 30 NSRT loggers (section 4.3, Fig. 5, Table 3), but only NSRT and snow depth data of 4 loggers (representative for typical snow conditions in the rock walls) were presented in detail (section 4.2, Figs. 3, 4, Table 2), since providing data from all loggers would be too much. This will be clearly stated in the text and the appropriate sections (last section in introduction, methods, results) will be reworked accordingly, since it seems that the use of all 30 NSRT loggers was not clear for the reader. For instance in Table 3 data of all 30 NSRT loggers are used. |
| 31. Lines 288-292: this paragraph is not clear either. "four independent TLS", but one of them was used to scale the snow accumulation (11.12.2013), right? So, is it true to say "independent". "R2=0.95": which data were used for this calculation: modelled versus measured snow height for each grid cells and for each TLS survey?
Why to show results from 11.12.13 if those data are used for scaling (and are therefore not independent)? Could you show a scatter | Correct. The TLS of 11 December 2013 was used to scale the precipitation for model input. Only three independent TLS are therefore available for model validation. This will be changed in the text.

The $r^2$ = 0.95 is a mean calculated between modelled and measured snow depth for each grid cell and each of the TLS. MBE, $r^2$ and MAE (please see answer 6 to referee 2) between the modelled and the independent measured snow depth data of |

| | |
|---|---|
| plot or at least better illustrate the model output by e.g. replacing one of the 3D or 2D view in Figure 2? | each of the three independent TLS campaigns will be provided in the revised manuscript. In addition Fig. 2 will be revised (please see answer 29). |
| 32. Line 296: Is the term "validation" really appropriate? | 'Validation' is correct, but only for the three independent TLS campaigns. This will be rewritten in the text and figure caption. |
| 33. Lines 299-301: here you give a reason for misfit between modelled and measured values. The same explanation could be expected lines 290-292: where the under/overestimations are coming from? Modelling of ablation? If it is given in the discussion, the same should be done for these lines 299-301. If you make the choice to directly discuss your results, an explanation could be expected lines 290-292. | The misfit between modelled and measured snow depths (lines 299-301) is also valid for lines 290-292. Possible explanations for model uncertainties are again presented in the discussion section. Hence, the results will be presented without any assessment or interpretation of results. Lines 299-301 will therefore be deleted. |
| 34. Lines 300-301: Here the modelled snow depth for the S measured point does not fit the measured values. A 1 m difference may have huge implications for the NRST simulations. How is that taken into account? | Although measured and modelled snow depth differences were > 0.5 m (especially on the S slope), these snow depth differences do not affect the rock thermal regime as long as snow depths are > 0.2 m. Steep, bare rock is decoupled from atmospheric influences for snow depths exceeding 0.2 m (Haberkorn et al. 2015a). Amongst others, Luetschg et al. (2008) and Zhang (2005) stated that the influence of snow depth variations on ground temperatures in the presence of a thick snow cover are small, whereas snow depth variations only have strong effects on the ground thermal regime for thin snow cover (in steep, bare rock we found the threshold to be 0.2 m). Of course such big snow depth differences might have an effect on the snow cover *duration*. However, both snow cover timing and duration were reproduced nicely by the model, which can be observed comparing measured and modelled NSRTs in Fig. 3b, c and Table 2 (snow cover duration). It has been shown repeatedly that realistically |

| | modelled snow cover duration over the winter is more important than accurately modelled snow depths at certain points in time (e.g. Fiddes et al. 2015; Marmy et al. 2013). |
|---|---|
| | While absolute snow depths were underestimated by Alpine3D, the well modelled snow cover duration implies an underestimation of snow melt in the model. This may be at least partly explained with the 1d snow module which does not account for 3d heat flow between adjacent snow-free and snow-covered rock portions, as well as micro-meteorological processes due to uneven heating during the ablation period which accelerate snow melt in reality. We will amend the manuscript regarding this issue. |
| 35. Lines 342-344: why such a big bias (-2_C)? What is its implication in the overall results? | Likely explanations of model uncertainties are given in the discussion. Especially at locations lacking snow the underestimation of modelled NSRT may result from both air temperature and wind speed differences between rock walls and the flat field AWS. Air temperature and wind speed measured at the AWS may be a poor surrogate for the prevailing conditions in steep rock. Hence the turbulent flux simulations are biased (provided in discussion lines 480-483). Further, also differences in slope and aspect between the model domain and reality can be a possible error source (please see answer 6). The effects of too cold modelled NSRT at locations lacking snow are shown in Fig. 5. Boxplots representing model results show a bigger scatter. |
| 36. Lines 362-364: the difference is calculated using the 30 NRST time series? | Correct. This is mentioned in lines 357-359. The text will be clarified, since it might not be clear. |
| 37. Lines 363-364: not as high as expected for "real" North and South walls, which is not the case here, with rather NW and SE faces. This must be taken into consideration! | We will clarify the text. Although NW and SE facing rock slopes are considered, the NSRT differences are still smoothed due to thick snow. |

| | |
|---|---|
| 38. Lines 392-399: figure 6 deserves much more description and precision. Could you be more precise in the text with the 1.9_C? For what? Entire model domain? North face.? South face? Measurement point controls? | To do so we will rewrite Section 4.3.4, also for better understanding.

 The value of 1.9 °C is the difference of MANSRT averaged over the whole model domain (taking into account each pixel regardless of aspect) between the modelled snow-covered and the modelled snow-free scenario. For the snow-free scenario the precipitation input of the model was forced to be zero (explanation in lines 242-245). |
| 39. Lines 425ff: here the energy balance of snow free N7 is presented like absolutely different from snow covered N7 ("In contrast to", "differed strongly") . However, when looking at Figure 7, the pattern of Qnet seems quite similar, only the magnitude differs, Qsensible differs in a certain degree. Are the terms really appropriate? | Comparing the energy balance of both snow-covered and snow-free (see explanation above) scenarios at location N7 reveals important differences especially in May, June and July (ablation period): for snow-covered conditions (Fig. 7a) all available energy is used to melt the snow, indicated by the snow melt term $Q_{melt}$. In contrast during the same period all available energy is used to directly warm the rock assuming snow-free conditions (Fig. 7b), which is then compensated by the sensible heat flux. These differences in energy fluxes between snow-covered and snow-free scenarios result in totally different NSRT evolution. In addition different albedo effects arise between both scenarios (snow versus bare rock). |
| 40. Line 367: "effects: In" is either "effects. In" or "effects: in" | Will be changed. |
| 41. Line 447: why not modelling heat transfers in fractures is a limit of your model? Are you also modelling the interior of the ridge? If not, please remove, there are already enough details to discuss. If yes, it should appear clearly all along the text that you do not only model surface temperature and substantial results on the model temperature at depth must be provided! | 'Water flow along fractures' will be removed. |
| 42. Line 450: the consideration of snow cover at the ground surface is especially important to model small scale temperature variability. | It is true, that in near-vertical, ideal, snow-free rock faces air temperature and solar radiation might be sufficient to model |

| | |
|---|---|
| Some studies have shown that equilibrium temperature fields and long-term changes can mainly consider air temperature and solar radiation in steep slope. Please, rework the sentence accordingly. | ground surface temperatures. However, in fractured, structured and variably inclined rock faces this is not the case and the snow has to be taken into account, as already stated e.g. in Haberkorn et al. (2015a,b) or Magnin et al. (2015). |
| 43. Lines 456-457: the statement is interesting (it appear more important to correctly model snow timing to better represent snow effect) but could you at least provide one example in order to help the reader to connect this discussion to the results? | We will cite Fig. 3b, c. Here it is shown that modelled and measured NSRT are in good agreement, although absolute snow depths vary by around 0.5 m. Please see also answer 34. |
| 44. Line 461: again, the references are not adapted to the text: Gruber et al., 2004 and Noetzli et al., 2007 do not propose a "traditional snow modelling technique". | We will rework this sentence. Gruber et al. (2004a) and Noetzli et al. (2007) do not account for snow in idealized slopes > 50°. |
| 45. Lines 485-486: this belongs to the results, so move in another section and connect it with the presented results. On which source of data is this calculated? What is the difference with other presented MBE (e.g. MBE of -2_C line 342)? | This sentence will be moved to the results section 4.3.3 and will be reworked for better understanding. In addition we will refer to Table 3. The MBE analysis was calculated between measured and modelled NSRT data for each of the 30 NSRT logger locations. The average MBE was than calculated for the entire N and S facing slopes while averaging all MBE of N facing locations and averaging all MBE of the S facing locations. Hence the average MBE error of -0.2 °C in the N slope and of -1 °C in the S facing slope include all N and S facing locations (30) and thus account for the various snow conditions in the rock walls. The MBE addressed in line 342 is only calculated between measured and modelled NSRT of one N (N3) and one S (R2) facing location lacking snow. |
| 46. Lines 504-505: Isn't the difference of snow free/snow covered faces between N/S aspects in the range of model uncertainty? | In the N facing slope the NSRT difference between snow-free and snow-covered scenarios is up to 2 °C, while the MBE is only -0.2 °C (please see lines 485-486 and comment above). In the S facing slope the NSRT difference |

| | between snow-free and snow-covered scenarios is up to 1.4 °C, while the MBE is up to -1 °C (please see lines 485-486 and comment above) and therefore close to the model uncertainty range. |
| --- | --- |
| | The differences given are calculated between modelled snow-free and modelled snow-covered scenarios (Fig. 5) and hence differences are relative. For both the point and spatial scale snow-covered scenarios are always warmer than snow-free scenarios. |
| 47. Lines 515-520: these appear as important results that would confirm recent findings in Scandinavian rock walls (Myhra et al., 2015): rock walls favour the presence of permafrost (here in the Alps, that would be especially true for North slopes?). This must be better emphasized. | This section will be better emphasized. Accordingly, the results will be improved and rewritten. |
| 48. Lines 541-542: reaching that stage of the paper, the use of 30 NRST logger is still not clear: where the validation is shown? In figures, only 4 loggers are used and discussed. Same question as previously: is "validation" really appropriate? | Please see answer 30.
 'Validation' is correct, since independent measured NSRT data is compared to modelled NSRT data. In addition the error analysis is also based on this data. |
| 49. Line 553: "50_", how this threshold has been defined? It appears for the first time in the conclusion. | This topic is addressed in the introduction (lines 60-63). It was in general assumed that wind and gravitational transport remove the snow from steep rock in slopes exceeding 50 to 60°. Please see also answer 44. The threshold of 50° will be stated in the last part of the introduction for better understanding. |
| 50. Line 554: is "accurately" really appropriate when significant bias have been displayed? | Sentence will be reworked. |
| 51. Lines 569-571: this is an interesting result but it has only be mentioned in the discussion. No quantitative information nor graphical results are provided for such statement. Either remove from the conclusion and remain as close as possible of your major findings, or develop the results related to grid-scale sensitivity analysis. | Another short chapter will be provided in the results (or at least a Table with simulation results of 0.2 m, 1 m and 5 m) in order to prove this statement. |
| 52. Figure 1c: what are the peaks between 2930-2950 and 161750-161780 on the y and | The peaks in Fig. 1c are artefacts in the DEM due to the projection of overhanging rocks. |

| | |
|---|---|
| x axis respectively? They look like artefacts in the DEM. How did you clean up the points cloud before generating the DEM? Furthermore, I the figure could be improved by including a hillshade below the elevation colour scale to improve the visibility of micro-topography. | Fig. 1c and 1d will be removed (comment 87 of referee 2) and replaced by a profile through the ridge for a better overview of the linear logger layout, elevations and slope angles.

 In Fig. 1a slope angle colours will be displayed in black and white to improve the visibility of the micro-topography. Fig. 1a then resembles a hillshade with an illumination angle of 90°. |
| 53. Figure 2a: it is very difficult to read the legend, could you make it bigger? | Will be changed. |
| 54. Figure 3: This figure must be improved. I propose the following modification for better clarity and readability. The legend: measured NRST and the measured-modelled NRST have the same line colours. Make different colour. Some lines are dashed or dotted but this does not appear in the legendOf course, the reader can then easily find out which line in the legend corresponds to which line in the graph, but it is confusing at first glance and does not support rapid overview of the Figure: make the legend consistent with the line style. The measured-modelled NRST is not shown at an appropriate scale. Why not displaying these differences in independent plots below the model output? | The line colours of the measured NSRT and the measured-modelled NSRT have different blue shades. They might be difficult to distinguish. Therefore we will change line colours. In addition we will modify the legend and will provide a legend which is consistent with the line style.

 The measured-modelled NSRT will not be moved to an independent plot, since these graphs shall only provide a quick overview on differences between measured and modelled results. |

[Figure]

Fig. 1 for revision: Histogram for TLS data: solid lines illustrate the distribution of the ratio modelled/measured snow depth for the 4 TLS available. The TLS of 11 December 2013 (20131211, pink line) is centred by 1 (since this TLS was used for precipitation scaling). Dashed lines show a comparison between each TLS. First each pixel is corrected with the mean value of the TLS. Thus the relative snow depth per scan is provided. Than the ratios of the relative snow depths of each TLS are compared to each other.

**Abbreviations**

AWS: automatic weather station

DEM: digital elevation model

ILWR: incoming longwave radiation

IMIS: Intercantonal Measurement and Information System

ISWR: incoming shortwave radiation

MAE: mean absolute error

 MANSRT: mean-annual near-surface rock temperature

MBE: mean bias error

NSRT: near-surface rock temperature

NW: north-west

$r^2$: coefficient of determination

SE: south-east

TLS: terrestrial laser scanning

**References used in the response to referees**

Davies, M.C.R., Hamza, O., and Harris, C.: The Effect of Rise in Mean Annual Temperature on the Stability of Rock Slopes Containing Ice-Filled Discontinuities, Permafr. Periglac. Process., 12, 137-144, doi:10.1002/ppp378, 2001.

Dilley, A.C., and O'Brien, D.M.: Estimating downward clear sky long-wave irradiance at the surface from screen temperature and precipitable water. Q. J. Roy. Meteor. Soc., 124, 1391 – 1401, doi: 10.1002/qj.49712454903, 1998.

Fiddes, J., Endrizzi, S., and Gruber, S.: Large-area land surface simulations in heterogeneous terrain driven by global data sets: application to mountain permafrost, Cryosphere, 9, 411-426, doi:10.5194/tc-9-411-2015, 2015.

Flerchinger, G.N., Xaio, W., Marks, D., Sauer, T.J., and Yu, Q.: Comparison of algorithms for incoming atmospheric long-wave radiation. Water Resour. Res., 45, W03423, doi: 10.1029/2008WR007394, 2009.

Gruber, S.: Derivation and analysis of a high-resolution estimate of global permafrost zonation, Cryosphere, 6, 221-233, doi:10.5194/tc-6-221-2012, 2012.

Gruber, S., Hoelzle, M., and Haeberli, W.: Rock-wall Temperatures in the Alps: Modelling their Topographic Distribution and Regional Differences, Permafr. Periglac. Process., 15, 299-307, doi:10.1002/ppp.501, 2004a.

Haberkorn, A., Hoelzle, M., Phillips, M., and Kenner, R.: Snow as driving factor of rock surface temperatures in steep rough rock walls, Cold Reg. Sci. Technol., 118, 64-75, doi:10.1016/j.coldregions.2015.06.013, 2015a.

Haberkorn, A., Phillips, M., Kenner, R., Rhyner, H., Bavay, M., Galos, S.P., and Hoelzle, M.: Thermal Regime of Rock and its Relation to Snow Cover in Steep Alpine Rock Walls: Gemsstock, Central Swiss Alps, Geogr. Ann.: Ser. A, 97, 579-597, doi:10.1111/geoa.12101, 2015b.

Harris, C., Arenson, L.U., Christiansen, H.H., Etzelmüller, B., Frauenfelder, R., Gruber, S., Haeberli, W., Hauck, C., Hölzle, M., Humlum, O., Isaksen, K., Kääb, A., Kern-Lütschg, M.A., Lehning, M., Matsuoka, M., Murton, J.B., Nötzli, J., Phillips, M., Ross, N., Seppälä, M., Springman, S.M., and Vonder Mühll, D.: Permafrost and climate in Europe: Monitoring and modelling thermal, geomorphological and geotechnical responses, Earth-Sci. Rev., 92, 117-171, doi:10.1016/j.earscirev.2008.12.002, 2009.

Hasler, A., Gruber, S., and Haeberli, W.: Temperature variability and offset in steep alpine rock and ice faces, Cryosphere, 5, 977-988, doi:10.5194/tc-5-977-2011, 2011.

Lehning, M., Bartelt, P., Brown, B., Russi, T., Stöckli, U., and Zimmerli, M.: SNOWPACK model calculations for avalanche warning based upon a new network of weather and snow stations, Cold Reg. Sci. Technol., 30, 145-157, doi:10.1016/S0165-232X(99)00022-1, 1999.

Lehning, M., Grünewald, T., and Schirmer, M.: Mountain snow distribution governed by an altitudinal gradient and terrain roughness, Geophys. Res. Lett., 38, L19504, doi:10.1029/2011GL048927, 2011.

Luetschg, M., Lehning, M., and Haeberli, W.: A sensitivity study of factors influencing warm/thin permafrost in the Swiss Alps, J. Glaciol., 54, 696-704, doi:103189/002214308786570881, 2008.

Magnin, F., Deline, P., Ravanel, L., Noetzli, J., and Pogliotti, P.: Thermal characteristics of permafrost in the steep alpine rock walls of the Aiguille du Midi (Mont Blanc Massif, 3842 m a.s.l.), Cryosphere, 9, 109-121, doi:10.5194/tc-9-109-2015, 2015.

Marmy, A., Salzmann, N., Scherler, M., and Hauck, C.: Permafrost model sensitivity to seasonal climatic changes and extreme events in mountainous regions, Environ. Res. Lett., 8, 035048 9pp, doi:10.1088/1748-9326/8/3/035048, 2013.

Myhra, k.S., Westermann, S., and Etzelmüller, B.: Modelled Distribution and Temporal Evolution of Permafrost in Steep Rock Walls Along a Latitudinal Transect in Norway by CryoGrid 2D, Permafr. Periglac. Process., doi: 10.1002/ppp.1884, 2015.

Noetzli, J., and Gruber, S.: Transient thermal effects in Alpine permafrost, Cryosphere, 3, 85-99, doi:10.5194/tc-3-85-2009, 2009.

Noetzli, J., Gruber, S., Kohl, T., Salzmann, N., and Haeberli, W.: Three-dimensional distribution and evolution of permafrost temperatures in idealized high-mountain topography, J. Geophys. Res., 112, F02S13, doi:10.1029/2006JF000545, 2007.

Phillips, M., Haberkorn, A., Draebing, D., Krautblatter, M., Rhyner, H., and Kenner, R.: Seasonally intermittent water flow through deep fractures in an Alpine Rock Ridge: Gemsstock, Central Swiss Alps, Cold Reg. Sci. Technol., 125, 117-127, doi:10.1016/j.coldregions.2016.02.010, 2016.

Pogliotti, P.: Influence of Snow Cover on MAGST over Complex Morphologies in Mountain Permafrost Regions, Ph.D. thesis, 79 pp., University of Torino, Torino, Italy, 2011.

Sommer, C.G., Lehning, M., and Mott, R.: Snow in a very steep rock face: accumulation and redistribution during and after a snowfall event, Front. Earth Sci., 3, Article 73, doi:10.3389/feart.2015.00073, 2015.

Unsworth, M.H., and Monteith, J.L.: Long-wave radiation at the ground I. Angular distribution of incoming radiation. Q. J. Roy. Meteor. Soc., 101, 13 – 24, doi: 10.1002/qj.49710142703, 1975.

Voegeli, C., Lehning, M., Wever, N., and Bavay, M.: Scaling precipitation input to distributed hydrological models by measured snow distribution, Front. Earth Sci. – Cryospheric Sciences (submitted).

Wever, N., Schmid, L., Heilig, A., Eisen, O., Fierz, C., and Lehning, M.: Verification of the multi-layer SNOWPACK model with different water transport schemes, Cryosphere, 9, 2271-2293, doi:10.5194/tc-9-2271-2015, 2015.

Wirz, V., Schirmer, M., Gruber, S., and Lehning, M.: Spatio-temporal measurements and analysis of snow depth in a rock face, Cryosphere, 5, 893-905, doi:10.5194/tc-5-893-2011, 2011.

Zhang, T.: Influence of the seasonal snow cover on the ground thermal regime: An overview, Rev. Geophys., 43, RG4002, doi:10.1029/2004RG0001, 2005.

---

## Author Comment (AC2) · 31 Aug 2016

**Author final responses to Reviewers (Ref. No.: tc-2016-73)**

The authors thank referee #2 for the useful remarks and suggestions. All referee' comments (left) and our responses to them (right) are listed below.

**Referee 2:**

| Referee comment | Author answer |
|---|---|
| 1. The approximate use of technical terms as well as of references (often totally wrong!) denotes the scarce attention paid by writing the introduction chapter. I suggest to the authors to deeply review this chapter by checking carefully the references (all along the paper!) | The introduction will be reworked. Please see also answer 4 to referee 1. |
| 2. Sections 3.3.1 and 3.3.2 can be merged and shortened (mainly 3.3.1) by providing less detail about Alpine3D and SNOWPACK that are well known and documented models. | We will shorten section 3.3.1 and 3.3.2 in order to avoid repetitions. In addition we will merge sections 3.3.1 and 3.3.4. The surface energy balance is a core element of Alpine3D and belongs in the description of the energy balance model. Apart from the changes mentioned, the model description is already concise. Subsections 3.3.1 and 3.3.2 will still be treated separately for a better overview. |
| 3. The precipitation scaling is a very promising idea but it does not seem to work very well as it is. It would be very interesting to understand why in 3 of the TLS campaign does not work providing quantitative analysis of these discrepancies (see technical comment). Moreover, looking at figure 2 is evident that it works quite well on the validation point N7 but is scarce at point S9. In my opinion a simple ratio between AWS snow-depth and TLS snow-depth is a too simplistic approach and represents the main limitation of the present study.

I suggest the authors to put together all the TLS campaign data and AWS snow-depth data and try a more complex statistical approach which includes at least the topographical characteristics (ele, slp, asp) and doy (day of year) of the cells as scaling predictors. A first attempt could be to build | We agree with the referee that the comparison of the modelled to the measured snow depth data clearly showed discrepancies in modelling absolute snow depths. However, snow depth distribution and especially snow cover duration are reproduced nicely by the model. Well reproduced snow cover duration was found to be most important for modelling the ground thermal regime (e.g. Fiddes et al. 2015; Marmy et al. 2013), which becomes obvious in Fig. 3b, c. Please see also answer 34 to referee 1.

Although a quantitative analysis of the precipitation scaling approach is currently being evaluated (Voegeli et al., submitted) and beyond the scope of this contribution, we performed some additional analysis. This has been done since both referees |

| a linear model with all the predictors, run a stepAIC on it for selecting the significant ones and use the resulting regressive model to scale the precipitation. | expressed concerns about the discrepancies resulting from precipitation scaling. An additional figure (histogram) will be provided in the results section in order to justify the choice of one TLS used for precipitation scaling. Please see this figure attached to this response letter (Fig. 1 for revision). Here solid lines illustrate the distribution of the ratio modelled/measured snow depth for the 4 TLS available. The TLS of 11 December 2013 (20131211, pink line) is centred by 1 (since this TLS was used for precipitation scaling). Snow depth is underestimated for the other 3 TLS campaigns, while using the TLS of 11 December 2013 for precipitation scaling. Based on the solid lines in the figure attached we think it might be better to use snow depths derived from the TLS 19 December 2012 for precipitation scaling. Dashed lines in the figure attached show an intercomparison between each TLS. First each pixel is corrected with the mean value of the TLS. Thus the relative snow depth per scan is calculated. Then the ratios of the relative snow depths of each TLS are compared to the other scans. For each pixel a ratio of 1 would imply that the ratio with the mean value is constant between TLS campaigns. Hence one can consider this to be the best possible result while building a statistical model. While comparing the envelope of the dashed and solid lines it becomes obvious that the scatter of the dashed lines is similar or larger than the precipitation scaling approach, especially for high-winter TLS. The scatter of the envelope is too wide to build a representative statistical model. We therefore come to the conclusion that the precipitation scaling is currently the best possible method to introduce varying snow depths into the rock |

| | walls. It is also clear that the method is not perfect, but we consider this future research to improve.

In addition it has been shown repeatedly (e.g. Lehning et al., 2011) that small-scale statistical modelling of snow depth based on terrain parameters does not work very well. This is why we decided to use the scaling approach based on the measured snow distribution. We will provide the figure attached (Fig. 1 for revision) and additional discussion regarding this point in the revised manuscript. |
|---|---|
| 4. In my opinion the sections 3.3.4 and 4.4 are totally disconnected from other chapters, not in terms of concepts (energy balance is fundamental) but in terms of contents and argumentations. There are no links or references to what observed or discussed in the other sections, there is no think over possible source of modeling uncertainty, is just a chronicle on the course of each component along the seasons. I suggest the authors to remove these chapters, due to the already high number of data and elements to discuss. As it is, the energy balance discussion looks a digression that distracts the reader from the main subject of the paper. Alternatively the section 4.4 must be deeply reworked in order to provide precise evidence of what is discussed in the section 5.1 (Lines 471-484). | Section 3.3.4 will be merged to section 3.3.1. Section 4.4 will be reworked. Please see also answer 2. |
| 5. Section 4.1.1. The description of the measured snow cover variability by TLS is interesting but useless for the purpose of the paper and has scarce relevance for the scientific community because is too detailed and site-specific. It lacks effort to outline most general patterns of snow accumulation in steep rock walls. It would be very interesting to explore if in your dataset exists a relationship between snow-depth-TLS and steepness of the grid-cell. This analysis might be, I guess for the first time, a real measure of snow-depth in steep rock walls and provide the community some | Section 4.1.1 will be reworked.
The relationship between measured snow depth and slope angle will not be provided, since already enough methods and results are presented. Further it is not within the scope of this study and such an analysis will be presented elsewhere. |

| | |
|---|---|
| indications on the snow-depth thresholds to use for modeling experiments in steep rock walls. At first, this analysis (i) could exclude the cells above ledges and (ii) could analyze NW and SE faces separately. | |
| 6. Section 4.1.2. The statistics provided (R2 and MBE) are not sufficient. R2 indicates the fraction of variability (variance) in the observation that is explained by the model. Used alone it says little about model performance in strict sense because e.g. in case of temperature you can have an R2=0.99 with 10_C of bias. The modeling efficiency (ME) must be used also. MBE describes the direction of the error bias. Its value is related to the magnitude of values under investigation. A negative MBE occurs when predictions are smaller in value than observations, positive MBE occurs when predictions are greater in value than observations. In case of snow-detph has no sense to provide a mean value of MBE (-0.002 m!!) over the entire model domain because over- and under- estimations vanish each other. Mean absolute error (MAE) or root mean square error (RMSE) must be used instead. Also error bars in Fig.2 look strange, see technical comments. I suggest this paper for further detail: Mayer, D., and D. Butler (1993), Statistical validation, Ecological modelling, 68(1-2), 21–32. | The subsections 4.1.1 and 4.1.2 will be combined and lines 278-280 will be deleted. Please see also answer 29 to referee 1. Regarding the statistics used in this manuscript: first, MBE is important in case of snow since the bias over a whole area has huge implications. Second, the modelling efficiency is approximated by the $r^2$, even if root mean squared error or MAE are more common in some communities. In general, there is no single error analysis that says it all and every one is a little different. The choice of the authors to use $r^2$ and MBE is not a bad one. However, as requested the MAE will additionally be provided. |
| 7. Section 4.1.3. If one of the objective of this paper is (accurately) simulate the influence of snow cover on NSRT in steep rock walls I guess that differences in the order of 0.5 – 1m between observed and modeled snow depth is too much for obvious reasons. To reduce this uncertainty, as said in specific comments n.3, the precipitation scaling must be totally revised. | Please see answer 3, as well as answer 34 to referee 1. |
| 8. Section 4.2.2. This section would be a validation of NSRT but is very poor under this point of view. The absence of statistical metrics to evaluate model performance is evident here (see general comments). The description of discrepancies between obs. and mod. is only qualitative, comments are limited to temperature without any | Section 4.2.2 will be rewritten. From our point of view differences between modelled and measured data are quantitatively. Please see answer 6. A link to snow cover conditions in the rock walls has been done in lines 336-338. More details will be given here. |

| | |
|---|---|
| reference to the modeled snow which is the main constraining factor. In particular, observing together Fig.2a and Fig.3 results that temperature modeling has better performance where snow modeling has worst performance (point S9). Nothing is said about that. This section, that potentially could be the core of the paper, must be strongly improved. | It is correct that the ground thermal regime depends on snow conditions, but mainly on snow cover duration, not on absolute snow depths. Please see answer 34 to referee 1, as well as answer 3. Not only snow cover duration, but also ground conditions are important for near-surface rock temperature modelling. In the S facing slope NSRT can be simulated well since permafrost is absent in the S and most NSRT are around 0 °C below a thick snowpack. In addition the S rock surface is more homogenous (dip slope) compared to the N face (scarp slope). Thus the interaction between adjacent rock portions sticking out of the snow and rock portions covered by thick snow is reduced on the S face. |
| 9. Section 4.2.3. The idea of a run with forced snow-free condition is good but results are not exploited at all. This run could be used as reference to quantify the potential thermal effect of snow cover at different slope and aspect (see Pogliotti, 2011). This is a way to generalize the results and valorize the dry run. Of course, the precipitation scaling must be improved before (specific comments 2). | A comparison between simulations of snow-covered and snow-free scenarios was done in order to quantify errors made while neglecting snow in steep rock wall thermal modelling. Please see answer 38 to referee 1, as well as lines 101-104, 242-245, section 4.2.3, 4.3.3, 4.3.4 with Fig.6 and parts of 4.4. The objective to run Alpine3D also with forced snow-free conditions might not have been clear. This will be clarified in the text. |
| 10. Line 29: the term "rock avalanche" refers to big falls of earth material (of up to millions of metric tons) able to reach velocities of more than 50 meters per second and leave a long trail of destruction. In the Alps such phenomena are not "numerous" (e.g. Val Pola 1987, Tschierva 1988, Brenva 1997, Thurwieser 2005) and even less those where permafrost can be directly listed among the trigger factors. The right term is "rock falls". | Will be changed to 'rock fall'. |
| 11. Line 30: strange references, Gruber & Haeberli 2007 is better and more comprehensive than Gruber 2004b, e.g. Fisher 2012 (Nat. Hazards Earth Syst. Sci) is missing. | Will be changed. |
| 12. Line 31: Davies et. al 2001 is wrong! | Davies et al. (2001) and Gruber et al. (2004a) |

| | |
|---|---|
| Gruber et. Al 2004a is wrong! Fisher 2012 (Nat. Hazards Earth Syst. Sci) is more appropriate than Fisher 2006, Gruber & Haeberli 2007 is missing, Allen & Huggel 2013 (Glob. and Planetary Change) is missing, Saas 2012 (Nat. Hazards Earth Syst. Sci) is missing, Deline et al. 2015 (Snow and Ice- Related Hazards, Risks, and Disasters, chapter 15) is missing: : : and many more. | will be deleted. Other references will be provided, also with respect to your suggestions. |
| 13. Line 35: Gruber 2012 is wrong! e.g. Guglielmin 2003 (Geomorphology) is missing | Gruber (2012) will be removed. |
| 14. Line 36: if you cite only Fiddes et al. 2015 add "e.g." because exist more | Will be changed. |
| 15. - - - Line 37: kilometers | We will change 'meters' to 'metres', since British English is used throughout the manuscript. |
| 16. Line 41: transient changes… Harris et al. 2009 alone has no sense because is a big state-of-the-art of mostly all fields of research around mountain permafrost… Noetzli & Gruber 2009?? | Harris et al. (2009) will be removed. Noetzli et al. (2007) and Noetzli and Gruber (2009) will be moved at the end of the sentence. |
| 17. Line 46-49: …cannot capture… the ground thermal regime. I'm not sure of that. The Fiddes 2015 approach has not been yet validated against field measures. | Please see answer 13 to referee 1. The model results of Fiddes et al. (2015) were validated in the same publication against a network of air temperature, ground surface temperature and snow depth measurements, as well as data loggers (PERMOS) to evaluate ground surface temperature in coarse debris and bedrock. |
| 18. Line 56: remove "However" | Will be removed. |
| 19. Line 56-58: this statement is too strong and do not consider that the temperature of a point in depth integrates the contribution of a certain area at surface. This area is wider as deeper is the point so the effect you are talking about is probably limited to few meters. Thus, in my opinion, to investigate the 3D subsurface heat flow is not necessary to reproduce surface temperatures with so-high spatial resolution. Please, reformulate this sentence considering also these aspect. | The sentence will be reworked. |
| 20. Line 59-60: Gruber 2004 is wrong!, Gruber & Haberli 2007 is a kind of review and snow control only is mentioned, remove it. Pogliotti 2011 is probably the first work that systematically investigate the thermal | Please see answer 14 to referee 1. |

| | |
|---|---|
| effect of snow cover (moreover with high affinity with the present work) even in steep rock walls and is missing. Magnin 2015, Haberkorn 2015a & 2015b are missing too! | |
| 21. Line 63: Pogliotti 2011 is wrong! | Pogliotti (2011) will be removed. |
| 22. Line 65: Gruber et al. 2004A is wrong! | Please see answer 16 to referee 1. |
| 23. Lines 82-85: this sentence is not clear, explain better. | The whole sentence will be deleted. |
| 24. Line 106: elevation range must be explicit in the site description. | Will be given. |
| 25. Line 127: Remove However. In this study, only data from… | 'However' will be deleted and sentence rewritten for better understanding. |
| 26. Lines 130-136: what you describe here is not evident neither from figure 1 nor from table 1 but just in figure 3. If you don't show a plot you have to describe better the differences you observe in the temperature fluctuations in order to justify your choices. | Please see answer 23 to referee 1. |
| 27. Lines 191-194: the initialization is important. Provide here, synthetically, more details about initialization without reference to another paper. Is not clear as it is. | The sentences will be rewritten. However, all information regarding the initialization is given. |
| 28. Line 205: remove high resolution | Will be removed. |
| 29. Line 211: Uncertainties in modeling... | Will be changed. |
| 30. Line 213: R2 is the coefficient of determination! MBE is not the right statistic in this case, look at specific comments. | Will be changed. Please see also answer 6. |
| 31. Lines 209-213: move this paragraph as preamble of chapter 4. | The methods and results section will be reworked. This paragraph will possibly remain in the methods section. |
| 32. - Lines 216-218: remove. | Will be removed. |
| 33. Lines 222-224: what is the "snow depth driving mode" of snowpack? Something that convert snowfall in liquid precipitation? By which snow density value? This is a key step of your precipitation scaling, please explicit all the detail, synthetically, without references to other papers. | The 'snow depth driving mode' means that SNOWPACK was driven with measured snow depth as model input (not liquid precipitation). SNOWPACK converts fresh snow falls in precipitation under consideration of snow settlement, as well as fresh snow density which are both calculated based on a statistical model. Although this is not a key step in our precipitation scaling, but rather a common approach to calculate liquid precipitation if only snow depth is available, we will provide additional explanation on this topic. Detailed |

| | information, however, is given in Lehning et al. (1999) and Wever et al. (2015). We think providing these references in the manuscript is sufficient. As you mentioned, SNOWPACK and Alpine3D are well known and documented models. |
|---|---|
| 34. lines 225: "integrated" seems a mathematical term, please use a synonym. | Will be changed. |
| 35. Line 228: replace "onto the DEM" with "in each grid cell". | Will be changed. |
| 36. Lines 228-232: replace this sentence with "cells where TLS data were non available have been excluded from the analysis". | Will be changed. |
| 37. Line 233: TLS campaign. | Not changed. |
| 38. Lines 233-241: explain better why you choose only the TLS of December 2013 for driving the precipitation scaling and provide quantitative proofs for this choice (model performance on modeled vs. observed NSRT). Look also specific comments. | Please see answer 3. |
| 39. Line 247: see specific comments 4. | Please see answer 4. |
| 40. Line 262: see specific comments 5. | Please see answer 5. |
| 41. Line 277: see specific comments 6. | Please see answer 6. |
| 42. Line 279: MBE = -0.002 m has no sense. MBE is the wrong statistic in this case (see specific comments). | Please see answer 6. |
| 43. Lines 282-283: explain the method used for calculating the error bars and exactly what they represent. Is not clear. How can I have an error bar of _0.3m and a difference obs./mod. (red dot, red line) of about 1 m? | The error bars in Fig. 2a represent the errors only of the validation data itself. An error bar of ±0.3m is composed of both an error of ±0.08 m due to errors of the TLS method itself and an error of ±0.22 m inherited in the precipitation input data due to precipitation scaling. The highest inaccuracies of validation data occurred in areas with a strongly heterogeneous surface (N face). The error bars do not indicate differences between measured and modelled snow depth. The error bars in Fig. 2a might be omitted. |
| 44. Lines 300-301: explain/explore better the reasons of such a huge difference in S9. | Differences up to 1 m between measured and modelled snow depths in the S facing slope are mainly due to inadequate |

| | description of snow settlement. This is explained in the discussion section 5.1 (lines 451-456).

Lines 299-301 will be removed, since the results will be presented without any assessment or interpretation of the data. Possible explanations for model un-certainties are presented in the discussion. |
|---|---|
| 45. Line 287: see specific comments 7. | Please see answer 3, as well as answer 34 to referee 1. |
| 46. Line 334: what does it means "auspicious accordance"? please try to be more adjective | Will be changed. |
| 47. Line 335: MBE is the wrong statistic in this case (see specific comments). | Please see answer 6. |
| 48. Line 330: see specific comments 8. | Please see answer 8. |
| 49. Line 346: see specific comments 9. | Please see answer 9. |
| 50. Lines 363-364: this sentence is ambiguous, what does it means "not pronounced as expected"? Expected for N/S differences (?) this is not the real case. Expected for snow-free, steep, conditions(?) this is not the real case. If you average all the measures of a mountain side like the yours, the value you got is exactly what I expected. | MANSRT differences between the NW and SE faces are smoothed due to thick snow. MANSRT differences between both faces would have been bigger if the slopes would have been snow-free, as it is often assumed in literature for steep rock faces. The text will be clarified. |
| 51. Line 366: remove "compensating" | Will be removed. |
| 52. Line 367: remove "In 2013-2014" | Will be removed. |
| 53. Lines 367-370: respect the colon, merge these two sentences in one | Will be changed. |
| 54. Lines 374-376: the higher SD of modeled temperatures derives essentially by the scarce ability in reproduce real (in terms of thickness) snow cover conditions on both sides. | Please see answer 3, as well as answer 34 to referee 1. |
| 55. Line 378: how can you say that underestimation is mainly in summer? (fig. 3?). Explicit. | The sentence will be deleted. |
| 56. Lines 379-380: remove "therefore", this sentence is not a direct consequence of what you said before, or only partially. This is a comparison with the 3.6_C stated at line 363. Contextualize better this sentence. | The sentence will be reworked. |
| 57. Line 384: compared to what? Modeled or real snow covered conditions? It is very difficult to follow your reasoning looking at | Modelled MANSRT of snow-free simulations were around 2 ° C colder to both measured |

| | |
|---|---|
| Table 3 because the number in the text are often means of values in different columns of the table and moreover rounded! If you need these numbers add columns in the table! | MANSRT and modelled MANSRT assuming snow-covered conditions. This will be stated in the text. In this section only the 2 °C value (line 384) was rounded. This will be clarified in the text. Other values can be calculated from Table 3. |
| 58. Lines 383-390: rework this section in accordance with the previous comment. Consider also the specific comments n.9 | Please see answer 57 above. The difference in line 88 is calculated for modelled snow-free conditions between the N and the S facing slopes. Please see answer 9. |
| 59. Lines 392-399: very poor description. Provide more details or remove this section, figure 6 and the "grid" lines in table 3. | Please see answer 38 to referee 1. |
| 60. Line 401: see specific comments 4. | Please see answer 4. |
| 61. Line 447: modeling of water flow within fractures is not relevant for reproducing surface rock temperatures. Also the influence of surface water flow is negligible in comparison to a correct simulation of snow cover thickness. | 'Water flow in fractures' will be removed. |
| 62. Line 451: check the references (see specific comments 1) | The references will be checked. |
| 63. Lines 452: please explicit the value of snow density used (see also technical comment Lines 222-224) | SNOPWACK calculates fresh snow density for each time step by a statistical model. Please see answer 33. Lines 451-455 will be reworked for a better understanding. |
| 64. Line 453: remove "However" | Will be removed. |
| 65. Lines 454-455: the first half of the sentence (from However to AWS) is obvious thus can be removed, the second half is not clear, explain better this concept of non-linear settling. Include also the sentence after. | Please see answer 63. |
| 66. Lines 457-458: this is not evident from your data. Look the table attached (Fig.1) and justify your sentence. | Fig. 3b, c will be cited. In Fig. 3b, c it is shown that modelled and measured NSRT are in good agreement, although absolute snow depths vary by around 0.5 m. Please see also answer 34 to referee 1. In addition, snow cover duration for the loggers shown in Fig. 3b, c is given in Table 2, which will be also referred to. |
| 67. Line 459-461: is not evident to me. Check the references (see specific comments 1) | Please see answer 44 to referee 1. |

| | |
|---|---|
| 68. Line 462: what is the "apparent insulation"? | 'Apparent' will be removed. |
| 69. Lines 465-466: heat flux at the bottom (20m below) cannot be seen in surface in so short simulations! | Will be removed. |
| 70. Lines 468: remove "While" | Will be removed. |
| 71. Lines 471-484: this is interesting but is very difficult to see the evidence of what you are saying in the plot 7 as well as find references in the text of section 4.4. See specific comment 4. | References to Fig. 7 only belong to lines 477-480. Please see also answer 4. |
| 72. Lines 485-486: move this in the results providing evidence of the source data. Keep in mind specific comments about the use of MBE. | Please see answer 45 to referee 1. |
| 73. Lines 489-499: in my opinion this belong to section 5.1. Check the references (see specific comments 1) all along this paragraph. | This paragraph will not be moved to section 5.1, since model uncertainties are not discussed in this paragraph. |
| 74. Line 500: replace "possibly made" with "introduced" | Will be changed. |
| 75. Lines 504-505: looking at table 3 the warming effect on MANSRT is up to 3.7_C at N7 (2012-2013) and up to 1.5_C at S9 (2012-2013). Please keep attention and precision in reference to plot and table contents! | Lines 504-505 refer to the entire rock wall (Table 3) not to single locations (Table 2). We will cite Table 3 and clarify the text. |
| 76. Lines 508-511: this obviously depends on the amount of snow. A persistent thin snow cover has always cooling effect both at N and S faces, while a thick snow cover has warming effect. Thus the reason you observe on average a warming effect of snow cover is because you allow the accumulation of thick snow. If you have a look a other cells with thin snow I'm sure you can observe cooling effect between dry and snow simulation. So change this sentence keeping in mind also these aspect. | The influence of snow on mean annual rock temperatures close to the surface of course depends on snow depth and especially on snow cover duration. In this study snow accumulates for around 9 months a year and has a warming effect on bot NW and SE faces. The effect of thin snow on rock surface temperatures, especially on mean annual temperatures is still poorly studied. Whether thin snow has a cooling or warming effect on mean annual rock temperatures on both N and S faces strongly depends on snow cover duration. Thin snow < 0.2 m will not persist on S faces for several months, especially not during the months with most intense radiation and its effect on mean annual rock temperatures is still not clear and should be better investigated in future. |

| | The contradiction of the presented results to previous studies (e.g. Hasler et al. 2011; Magnin et al. 2015) will be discussed more differentiated and the sentence will be reworked. |
|---|---|
| 77. Lines 515-520: this sentence is very interesting but not well introduced nor supported by findings of this paper. Provide more detail, evidence and argumentations in order to support this suggestion. | Please see answer 47 to referee 1. |
| 78. Line 524: this section is very interesting and useful for the modeling community, but is poor of numerical evidences. Please, provide a synthetic table (or plot) where the influence of grid resolution on the model performance becomes evident (see also specific comments for assessing model performance in the correct way). | Please see answer 51 to referee 1. |
| 79. Lines 551-553: I would say, "the results of the present work help to quantify the potential error…" | Sentence will be reworked. |
| 80. Line 554-556: "Alpine3D simulates near-surface rock temperatures and snow depth in the heterogeneous terrain accurately." in general this is true but is not the case of this work. The reason is that the precipitation scaling procedure is weak and provide unreliable precipitation input to the model. In my opinion this conclusion does not reflect the real result of this work. | Sentence will be reworked. |
| 81. Line 556-558: lateral heat-flux is negligible in comparison to the effect of a bad precipitation input. | Please see answer 3, as well as answer 34 to referee 1. Paragraph will be reworked slightly (lines 554-558). |
| 82. Line 559-561: this is true, the potential of the dataset is very high but the choice of exploring just 2 cells on the N face and 2 cells in the S face strongly constrain this potential. See also general comments. | Please see answer 30 to referee 1. |
| 83. Line 562: this sentence on the lateral heat flux is speculative. Nothing in the results provides the basis to verify this statement. | Sentence will be removed. |
| 84. Lines 569-571: also in that case no | Please see answer 51 to referee 1. |

| | |
|---|---|
| numerical evidence about model performance are provided in the results hence this sentence is speculative too. | |
| 85. Table 3: Caption (Line 812), replace "data" with "cells". How do you identified snow-free cells? | The sentence in lines 811-812 refers to the model run considering snow (in Table 3: modelled N grid snow & modelled S grid snow) and to the model run lacking snow (in Table 3: modelled N grid snow-free & modelled S grid snow-free), in the latter the precipitation input was forced to be zero. Modelled results given in the respective lines of Table 3 were averaged over the entire N and S facing model domain. Thus a comparison between the run considering snow and the run without snow has been done.
We might replace 'data' with 'runs'. In this case 'cells' are wrong. We will rework the table for better understanding. |
| 86. Figure 1: The boreholes are not considered at all in this work then I suggest to remove it from the figure and caption to avoid confusion. | The 30 shallow NSRT logger locations were used to validate model results. Please see answer 30 to referee 1. The horizontal borehole (points BH N and BH S in Fig 1a, e, f), which was drilled through the whole ridge, provided rock temperature data in various depths, which were used to initialize our model (Section 3.3.2). We will therefore not remove the boreholes. |
| 87. I suggest to replace the three colorful elevation plot by a "classic" but more readable cross-section along the logger line which easily can gives the information about elevation and steepness at one-shot. | Will be changed. Please see also answer 52 to referee 1. |
| 88. Figure 2: Just figures a) and f) are relevant for the interpretation and discussion of the precipitation scaling. Remove figures b) c) d) e) that are not relevant and enlarge figure a). | We will revise Fig. 2. Please see answer 29 to referee 1. |
| 89. The range in figure f) has been constrained at _0.5m for graphical reasons, but a frequency distribution plot (barplot) of differences on the model domain should be | We will either provide a scatter or a bar plot to show differences between measured and modelled snow depth. Please see also answer 29 to referee 1. |

| | |
|---|---|
| inserted as compendium to provide a comprehensive overview of modeled snow depth uncertainties. | |
| 90. Figure 3: Caption: dT are present also in the plots d) and e) not only in b) and c) as stated. | The caption was ambiguous. We meant that dT was calculated in Fig. 3b, c between measured and modelled snow-covered conditions, although snow-free conditions were also shown. In Fig. 3d, f dT is calculated between measured and modelled snow-free conditions. Will be reworked. |
| 91. Figure 5: The boxplot shows the meadian but in the text and table 3 the references are always to the mean. Please modify the boxplot in accordance with the text. | In the boxplots the mean will also be provided. |
| 91. Figure 5: The boxplot shows the meadian but in the text and table 3 the references are always to the mean. Please modify the boxplot in accordance with the text. | In the boxplots the mean will also be provided. |

[Figure]

Fig. 1 for revision: Histogram for TLS data: solid lines illustrate the distribution of the ratio modelled/measured snow depth for the 4 TLS available. The TLS of 11 December 2013 (20131211, pink line) is centred by 1 (since this TLS was used for precipitation scaling). Dashed lines show a comparison between each TLS. First each pixel is corrected with the mean value of the TLS. Thus the relative snow depth per scan is provided. Than the ratios of the relative snow depths of each TLS are compared to each other.

**Abbreviations**

AWS: automatic weather station

DEM: digital elevation model

ILWR: incoming longwave radiation

IMIS: Intercantonal Measurement and Information System

ISWR: incoming shortwave radiation

MAE: mean absolute error

MANSRT: mean-annual near-surface rock temperature

MBE: mean bias error

NSRT: near-surface rock temperature

NW: north-west

$r^2$: coefficient of determination

SE: south-east

TLS: terrestrial laser scanning

**References used in the response to referees**

Davies, M.C.R., Hamza, O., and Harris, C.: The Effect of Rise in Mean Annual Temperature on the Stability of Rock Slopes Containing Ice-Filled Discontinuities, Permafr. Periglac. Process., 12, 137-144, doi:10.1002/ppp378, 2001.

Dilley, A.C., and O'Brien, D.M.: Estimating downward clear sky long-wave irradiance at the surface from screen temperature and precipitable water. Q. J. Roy. Meteor. Soc., 124, 1391 – 1401, doi: 10.1002/qj.49712454903, 1998.

Fiddes, J., Endrizzi, S., and Gruber, S.: Large-area land surface simulations in heterogeneous terrain driven by global data sets: application to mountain permafrost, Cryosphere, 9, 411-426, doi:10.5194/tc-9-411-2015, 2015.

Flerchinger, G.N., Xaio, W., Marks, D., Sauer, T.J., and Yu, Q.: Comparison of algorithms for incoming atmospheric long-wave radiation. Water Resour. Res., 45, W03423, doi: 10.1029/2008WR007394, 2009.

Gruber, S.: Derivation and analysis of a high-resolution estimate of global permafrost zonation, Cryosphere, 6, 221-233, doi:10.5194/tc-6-221-2012, 2012.

Gruber, S., Hoelzle, M., and Haeberli, W.: Rock-wall Temperatures in the Alps: Modelling their Topographic Distribution and Regional Differences, Permafr. Periglac. Process., 15, 299-307, doi:10.1002/ppp.501, 2004a.

Haberkorn, A., Hoelzle, M., Phillips, M., and Kenner, R.: Snow as driving factor of rock surface temperatures in steep rough rock walls, Cold Reg. Sci. Technol., 118, 64-75, doi:10.1016/j.coldregions.2015.06.013, 2015a.

Haberkorn, A., Phillips, M., Kenner, R., Rhyner, H., Bavay, M., Galos, S.P., and Hoelzle, M.: Thermal Regime of Rock and its Relation to Snow Cover in Steep Alpine Rock Walls: Gemsstock, Central Swiss Alps, Geogr. Ann.: Ser. A, 97, 579-597, doi:10.1111/geoa.12101, 2015b.

Harris, C., Arenson, L.U., Christiansen, H.H., Etzelmüller, B., Frauenfelder, R., Gruber, S., Haeberli, W., Hauck, C., Hölzle, M., Humlum, O., Isaksen, K., Kääb, A., Kern-Lütschg, M.A., Lehning, M., Matsuoka, M., Murton, J.B., Nötzli, J., Phillips, M., Ross, N., Seppälä, M., Springman, S.M., and Vonder Mühll, D.: Permafrost and climate in Europe: Monitoring and modelling thermal, geomorphological and geotechnical responses, Earth-Sci. Rev., 92, 117-171, doi:10.1016/j.earscirev.2008.12.002, 2009.

Hasler, A., Gruber, S., and Haeberli, W.: Temperature variability and offset in steep alpine rock and ice faces, Cryosphere, 5, 977-988, doi:10.5194/tc-5-977-2011, 2011.

Lehning, M., Bartelt, P., Brown, B., Russi, T., Stöckli, U., and Zimmerli, M.: SNOWPACK model calculations for avalanche warning based upon a new network of weather and snow stations, Cold Reg. Sci. Technol., 30, 145-157, doi:10.1016/S0165-232X(99)00022-1, 1999.

Lehning, M., Grünewald, T., and Schirmer, M.: Mountain snow distribution governed by an altitudinal gradient and terrain roughness, Geophys. Res. Lett., 38, L19504, doi:10.1029/2011GL048927, 2011.

Luetschg, M., Lehning, M., and Haeberli, W.: A sensitivity study of factors influencing warm/thin permafrost in the Swiss Alps, J. Glaciol., 54, 696-704, doi:103189/002214308786570881, 2008.

Magnin, F., Deline, P., Ravanel, L., Noetzli, J., and Pogliotti, P.: Thermal characteristics of permafrost in the steep alpine rock walls of the Aiguille du Midi (Mont Blanc Massif, 3842 m a.s.l.), Cryosphere, 9, 109-121, doi:10.5194/tc-9-109-2015, 2015.

Marmy, A., Salzmann, N., Scherler, M., and Hauck, C.: Permafrost model sensitivity to seasonal climatic changes and extreme events in mountainous regions, Environ. Res. Lett., 8, 035048 9pp, doi:10.1088/1748-9326/8/3/035048, 2013.

Myhra, k.S., Westermann, S., and Etzelmüller, B.: Modelled Distribution and Temporal Evolution of Permafrost in Steep Rock Walls Along a Latitudinal Transect in Norway by CryoGrid 2D, Permafr. Periglac. Process., doi: 10.1002/ppp.1884, 2015.

Noetzli, J., and Gruber, S.: Transient thermal effects in Alpine permafrost, Cryosphere, 3, 85-99, doi:10.5194/tc-3-85-2009, 2009.

Noetzli, J., Gruber, S., Kohl, T., Salzmann, N., and Haeberli, W.: Three-dimensional distribution and evolution of permafrost temperatures in idealized high-mountain topography, J. Geophys. Res., 112, F02S13, doi:10.1029/2006JF000545, 2007.

Phillips, M., Haberkorn, A., Draebing, D., Krautblatter, M., Rhyner, H., and Kenner, R.: Seasonally intermittent water flow through deep fractures in an Alpine Rock Ridge: Gemsstock, Central Swiss Alps, Cold Reg. Sci. Technol., 125, 117-127, doi:10.1016/j.coldregions.2016.02.010, 2016.

Pogliotti, P.: Influence of Snow Cover on MAGST over Complex Morphologies in Mountain Permafrost Regions, Ph.D. thesis, 79 pp., University of Torino, Torino, Italy, 2011.

Sommer, C.G., Lehning, M., and Mott, R.: Snow in a very steep rock face: accumulation and redistribution during and after a snowfall event, Front. Earth Sci., 3, Article 73, doi:10.3389/feart.2015.00073, 2015.

Unsworth, M.H., and Monteith, J.L.: Long-wave radiation at the ground I. Angular distribution of incoming radiation. Q. J. Roy. Meteor. Soc., 101, 13 – 24, doi: 10.1002/qj.49710142703, 1975.

Voegeli, C., Lehning, M., Wever, N., and Bavay, M.: Scaling precipitation input to distributed hydrological models by measured snow distribution, Front. Earth Sci. – Cryospheric Sciences (submitted).

Wever, N., Schmid, L., Heilig, A., Eisen, O., Fierz, C., and Lehning, M.: Verification of the multi-layer SNOWPACK model with different water transport schemes, Cryosphere, 9, 2271-2293, doi:10.5194/tc-9-2271-2015, 2015.

Wirz, V., Schirmer, M., Gruber, S., and Lehning, M.: Spatio-temporal measurements and analysis of snow depth in a rock face, Cryosphere, 5, 893-905, doi:10.5194/tc-5-893-2011, 2011.

Zhang, T.: Influence of the seasonal snow cover on the ground thermal regime: An overview, Rev. Geophys., 43, RG4002, doi:10.1029/2004RG0001, 2005.

---

## Author Response (AR1)

**Author final responses to Reviewers (Ref. No.: tc-2016-73)**

The authors thank referee #1 and #2 for the useful remarks and suggestions. We have made significant changes to the entire manuscript (including Figures and Tables), as required by the referees' and because we carried out all simulations again. All the **referee' comments**

5 **are in bold**, our responses to them are in grey and without formatting and *changes to the initial manuscript are in italics*.

General comments:

To evaluate the performance of the model applied, model results of snow depth and rock
10 temperatures in 10 cm depth (NSRT) were compared to detailed measured validation data (snow depth of 3 independent TLS, 22 of 30 NSRT measurement locations). An error analysis of snow depth and NSRT (MBE, $r^2$) was performed at each of the 22 NSRT locations. While analysing the influence of snow on rock temperatures for the entire N and S facing rock walls, means of all NSRT loggers were calculated. The same is true for the error analysis.

15 Since it would be too much to show model results and their comparison to measured data, only 4 of the NSRT logger locations were chosen to show in detail. These 4 temperature loggers were chosen in order to represent typical NSRT evolutions depending on whether the location is snow-covered or not. For these 4 locations single error analyses are presented, but the error analysis for the 22 other NSRT loggers are also presented.
20 According to both referees, that was not clear from the manuscript. We therefore will clearly state this.

**Answers to Referee 1:**

**1. I suggest writing the abstract again: first, state what is the global scientific context and specific objectives of the paper. Improve the highlights of your main findings in the results paragraph. Try to be more quantitative if possible The results provided at lines 20-21 seem to contradict previous statements from former studies (e.g. Hasler et al. 2011 found a cooling effect of snow in the sun-exposed rock walls such as mentioned line 81), which is of high interested for the scientific community. This contradiction with current theory might deserve more developments, at least a few words in the abstract and some more in the results/discussion/conclusion to explain why the here presented findings differ from the previous ones (matter of snow height? Snow timing?).**

We will rewrite the abstract after revising the whole manuscript. The research questions and objectives of this study will be clarified. In addition the main results will be clearly emphasized.

The contradiction of the presented results to previous studies (e.g. Hasler et al. 2011; Magnin et al. 2015) will be discussed in more detail in the discussion section, but will not be provided in the abstract. There are still open research questions, which our study does not answer (e.g. thin snow > cooling yes or no).

*Abstract, page 1, new lines 8-27:*

***Abstract**. In this study we modelled the influence of the spatially and temporally heterogeneous snow cover on the surface energy balance and thus on rock temperatures in two rugged, steep rock walls on the Gemsstock ridge, central Swiss Alps. The heterogeneous snow depth distribution in the rock walls was introduced to the distributed, process based energy balance model Alpine3D with a precipitation scaling method based on snow depth data measured by terrestrial laser scanning. The influence of the snow cover on rock temperatures was investigated by comparing a snow-covered model scenario (precipitation input provided by precipitation scaling) with a snow-free (zero precipitation input) one. Model uncertainties are discussed and evaluated at both the point- and spatial-scale against 22 near-surface rock temperature measurements and high-resolution snow depth data from winter terrestrial laser scans.*

*In the rough rock walls, the heterogeneously distributed snow cover was moderately well reproduced by Alpine3D with mean absolute errors ranging between 0.47 and 0.77 m. However, snow cover duration was reproduced well and consequently near-surface rock temperatures were modelled convincingly. Uncertainties in rock temperature modelling were found to be around 1.6 °C. Errors in snow cover modelling and consequently in rock temperature simulations are explained by inadequate snow settlement due to linear precipitation scaling, missing lateral heat fluxes in the rock, as well as by errors caused by interpolation of shortwave radiation, wind and air temperature into the rock walls.*

*Mean annual near-surface rock temperature increases were both measured and modelled in the steep rock walls as a consequence of a thick, long lasting snow cover. Rock temperatures*
65 *were 1.3-2.5 °C higher in the shaded and sunny rock walls, while comparing snow-covered to the snow-free simulations. This helps to assess the potential error made in ground temperature modelling when neglecting snow in steep bedrock.*

**2. Improve the presentation of the methods with (1) a specific figure and (2) an**
70 **introductory section to explain in a very short paragraph how the methodological approaches, the various spatial resolutions from the different approaches, the characteristics of the input and output data are imbricated. The methodological approach is very complete and involves many steps with various sources of data and computing steps at various time and space scale. After several readings, it is still difficult to gain an**
75 **overview of the imbrication of data and processing. To improve the visibility and understanding of the method outlines, I suggest preparing a specific figure to sum up the imbrication of the input/output data and processing (e.g. similar and maybe slightly more detailed than the Figures 2 from Noetzli et al. 2007; Figures 1 from Noetzli and Gruber 2009 or from Fiddes et al., 2015). Also, it would be very helpful to have one or a few**
80 **introductory sentences for chapter 3 to sum up how the methods and data are imbricated.**

An additional figure in form of a flow chart will be provided in order to present an overview of the methods/ data used for both driving and validating the model. We will adapt the chronology of the methods section based on this flow chart. In addition to the flow chart we
85 will provide a short introduction in the methods section.

*Please see new Figure 2.*

[Figure]

Figure 2. Flow chart of the methods applied in order to run the numerical model Alpine3D and to validate the model output at both the point- and the spatial-scale.

An introduction paragraph was added: Section 3, page 4, new lines 128-133 :

Applying the Alpine3D model chain for spatially distributed steep rock wall thermal modelling requires various input data and computing steps. In Fig. 2 a brief synopsis of the methods used in this study are shown. Based on Fig. 2 first the distributed numerical model used in this study is introduced. Then the data and model settings required to drive the model are specified, followed by a description of the computation of the precipitation input, which is essential in order to introduce varying snow depths to the extremely steep terrain. Finally the validation data-sets used to evaluate the model performance are introduced.

In addition the chronology of Section 3 was adopted according to new Figure 2:

*3. 1 Distributed energy balance modelling (page 4, new line 135)*
*3.1.1 The Alpine3D model (pages 4-5, new lines 136-169)*
*3.1.2 Model setup (page 5, new lines 171-196)*
*3.1.3 Precipitation input for Alpine3D (page 5, new line 198)*
*Terrestrial laser scanning (TLS) (pages 5-6, new lines 199-211)*
*Precipitation scaling (pages 6-7, new lines 213-247)*
*3.2 Sensitivity study (page 7, new lines 249-257)*

*3.3 Model validation (page 7, new lines 259-265 )*
  *Near-surface rock temperature (NSRT) data (pages 7-8, new lines 266-287)*
  *Snow depth data (page 8, new lines 289-296)*

115 **3. Improve the presentation of the results and their discussion. In a similar way than the methods, an introductory paragraph to sum up your approach and clearly explain the outlines of the result chapter would be really helpful given the high number of steps in the result presentation. Some general suggestions and comments are given here below, but more specific comments are given in the appropriate section. When simulating at high**
120 **spatial resolution, the sources of uncertainty are many, and this result in sometimes important bias. Those sources of bias are not always well discussed (e.g. why the model is more performant on the N than on the S face?), whereas some discussion points seem disconnected from the study (why to mention the effects of water percolation along fractures? Where is the link with your study?). Also, to help in understanding the sources**
125 **of errors, it seems important to compare topographical characteristics of your measurement points used for model evaluation in the "real-world" and those in the "numerical environment" (DEM). Do the sensors have same aspect in both situations? Same slope angle? This information could be added in Table 1 for instance. In case of substantial discrepancies between both environments, this could explain a part of the**
130 **bias. Results are sometimes hard to follow due to the numerous back-and forth between figures and the text. You sometimes refer to several figures for a same thing, and not all references are relevant (e.g line 336: you refer to Fig. 3b to show the difference between measured and modelled MANRST, whereas Figure 3 shows daily variation). Figure 3a is not referred in the text. The data on which the MBE and R2 are calculated not always clearly**
135 **indicated. Many confusions are arising and being more precise would help the reader to go straight to the point. Section 4.1.3. must be written more clearly, at that stage, it is hard to follow. It contains lots of essential information but some details are missing to well understand how the model evaluation is performed, how the misfit between measured and modelled value are taken into account to go further in the study (see detailed**
140 **comments).**
**Finally, the study seems to contradict previous findings. So far, it was suspected that snow on South faces cools the surface temperature (e.g. Hasler et al., 2011). In this study, the opposite is stated, and the contradiction is not well discussed, nor well emphasized. What would be the possible factors/processes explaining that your findings are in contradiction**
145 **with previous findings? Also, Figure 6 which shows an important part of the results is poorly discussed. By looking at this figure it clearly appear that vertical faces without snow induce colder conditions than snow covered slopes. This is well aligned with recent findings in Norway (Myhra et al., 2015) and should be better emphasized and discuss.**

150 The structure of the paper will be revised. At the end of the introduction we will clearly state the aims and objectives of our study for a better understanding of the whole manuscript.

*Section 1, page 3, new lines 91-110:*

*We therefore present a spatially distributed model study of the influence of the snow cover on the surface energy balance and consequently on near-surface rock temperatures (NSRT) in steep north-west and south-east oriented rock walls using the physics-based 3d atmospheric and surface process model Alpine3D (Lehning et al., 2006). The distribution of the spatially and temporally heterogeneous snow cover in the steep terrain (up to 85°) was provided to the model using a precipitation scaling approach. This was based on a combination of snow depth measurements from the on-site flat field automatic weather station (AWS) and high-resolution (0.2 m) snow depth distribution data obtained using TLS. The challenge of integrating representative precipitation input (e.g. Imhof et al., 2000; Fiddes et al., 2015; Stocker-Mittaz et al., 2002) in the rock walls and its redistribution by wind (Mott and Lehning, 2010), as well as gravitational transport (Bernhardt and Schulz, 2010; Gruber, 2007) was thus accounted for. Model performance for simulating snow depth distribution and consequently the influence on rock temperatures was tested against a dense network of validation measurements of snow depth and NSRTs at both the point- and the spatial-scale. After quantifying model uncertainties, a sensitivity study was performed in order to assess the effects of the snow cover on the rock thermal regime. High-resolution (0.2 m) simulations were carried out, either providing snow cover distribution to the model (by precipitation scaling) or fully neglecting the presence of a snow cover in the rock walls. Thus the potential error induced by neglecting the snow cover in steep rock face thermal modelling for slope angles >50° can be estimated. This is necessary, since it has in general been assumed that wind and gravitational transport remove the snow from steep rock in slopes >50–60° (e.g. Blöschl and Kirnbauer, 1992; Gruber Schmid and Sardemann, 2003; Winstral et al., 2002) and rock temperatures were often modelled without snow for idealized rock walls >50° (e.g. Gruber et al., 2004a; Noetzli and Gruber, 2009; Noetzli et al. 2007).*

According to this the methods and results section will be reorganised. Some introductory sentences will be provided in the beginning of both the methods and results sections to better lead the reader through the manuscript (changes in the methods are answered in answer 2). It will be clearly distinguished between model results and validation data.

*Introduction to Section 4, page 8, new lines 299-309:*

*In this section only measured and modelled results are presented, while model uncertainties will be discussed in Section 5. First the measured and modelled snow cover accumulating in the rock walls is described at both the spatial- and the point-scale (4 selected locations). The accumulation of snow changes the surface energy balance of the rock walls, which is discussed in Section 4.2, where the surface energy balance is presented for both the virtually snow-free and the snow-covered scenario at two NSRT locations accumulating snow. Changes in the surface energy balance are mirrored in the rock temperatures. The rock thermal regime close to the surface is firstly presented at the 4 selected NSRT locations (Section 4.3), followed by the spatial analysis (all 22 NSRT locations) of measurements and model results of both the snow-covered and the snow-free scenario (Section 4.4). Finally the*

*accuracy of model results for coarser resolutions (1 m, 5 m) is evaluated in Section 4.5. Mean annual near-surface rock temperature (MANSRT), $r^2$, MAE and MBE are always given for the study years 2012-2013/2013-2014, separated by a slash (e.g. MANSRT for 2012-2013/MANSRT for 2013-2014).*

In addition the chronology of Section 4 was revised:

*4. 1    Spatial snow cover variability (page8, new line 311)*

*4.1.1    Measured snow cover variability (page 8, new lines 312-323)*

*4.1.2    Modelled snow cover variability (page 9, new lines 325-344)*

*4.2    Modelled surface energy balance at selected points (page 9, new lines 346-351)*

*4.2.1    Snow-free scenario (pages 9-10, new lines 353-369)*

*4.2.2    Snow-covered scenario (page 10, new lines 371-385)*

*4.3    NSRT variability at selected locations (page 10, new lines 387-392)*

*4.3.1 NSRT variability at snow-free locations (page 10, new lines 394-402)*

*4.3.2 NSRT variability at snow-covered locations (pages 10-11, new lines 404-424)*

*4.3.3 Thermal effect of snow (page 11, new lines 426-441)*

*4.4    MANSRT variability in the entire rock walls (pages 11-12, new lines 443-447)*

*4.4.1 Snow-covered scenario (page 12, new lines 449-470)*

*4.4.2 Snow-free scenario (page 12, new lines 472-477)*

*4.4.3 Modelled spatial distribution of MANSRT variability (pages 12-13, new lines 479-501)*

*4.5    Influence of grid resolution (page 13, new lines 503-524)*

Concerning the results section: first modelled and measured snow cover data will be presented since the snow strongly influences the thermal regime of the rock walls. The snow cover section (4.1) will be revised to clarify various confusing points.

*Section 4.1, pages 8-9, new lines 311-344:*

*4.1    Spatial snow cover variability*

*4.1.1 Measured snow cover variability*

*Similar inter-annual patterns of snow depth distribution were observed using TLS (Figs. 4a-c). However, the variability of the snow depth distribution and thus of snow cover onset and disappearance at certain locations was high over both the N and S facing rock walls. Areas accumulating a thick snow cover can be in the immediate vicinity of snow-free areas due to strongly varying micro-topographic effects. The snow cover was more homogeneous and thicker on the smoother S facing dip slope than on the steeper and rougher N facing scarp slope. Steep to vertical areas far above ledges or areas close to the ridge were usually snow-free, as was the case for the N3 and R2 loggers (Figs. 4a-c). Locations close to the foot of the rock wall and steep areas just above flat ledges accumulated mean snow depths up to 3.5 m. Inter-annual snow depth variations are illustrated in Fig. 5 for both the four locations discussed in detail and for the flat field AWS. Snow depths were on average 1 m lower at both the AWS and NSRT logger locations in 2013-2014 compared to 2012-2013, resulting in snow disappearance up to 4 weeks earlier in 2014.*

*4.1.2 Modelled snow cover variability*

*The evaluation of the snow depth distribution modelled using Alpine3D (Figs. 4d-f) against*
240 *data from three independent TLS revealed a reasonably well reproduced snow depth*
*distribution with $r^2$ = 0.52-0.95 (Figs. 4j-l), while absolute snow depth differences were in the*
*range of +1.5 m to -1 m (Figs. 4g-i). Considering the area around NSRT locations modelled*
*snow depths are often underestimated (Fig. 5), whereas they are overestimated while*
*averaging over the entire model domain (MBE = 0.31 – 0.81 m). The MAE of the N and S*
245 *slopes varied between 0.47 – 0.77 m and always indicated higher deviations between snow*
*depth observations and predictions in the heterogeneous N facing slope (Table 2).*

*In Fig. 5 the evolution of modelled snow depths for the four selected locations within both the*
*N (N3, N7) and the S (R2, S9) facing slopes is shown. While R2 and N3 lacked snow, snow*
*accumulated for 7.5 to 9 months per year at N7 and S9. The modelled winter snowpacks are*
250 *compared to measured TLS snow depths (markers in Fig. 5) on the dates of TLS campaigns. At*
*the shaded N7 location, the measured and modelled snow depths fit well in early winter*
*(December/January 2012-2013 and 2013-2014), while modelled snow depths are*
*underestimated by 0.55 m in early summer (2012-2013). In the S facing slope differences*
*between measured and modelled snow depths are modest in early winter (0.04 and 0.5 m in*
255 *December), while during the course of the winter and ablation period modelled snow depths*
*were underestimated by up to 0.9 m. Although absolute snow depth differences are up to 0.9*
*m in the S slope, the snow cover durations (Table 1) were satisfactory reproduced by the*
*model. The accurately modelled timing of snow cover onset and disappearance was*
*confirmed by NSRT data in the grid cells corresponding to N7 and S9 (see Figs. 7b, c; Section*
260 *4.3), as well as at all other NSRT locations (not shown).*

New Fig. 4 was correspondingly be adapted, please see answer 29 for this figure.

In a next step the NSRT data will be discussed. The usage of NSRT data of all loggers for
265 model validation will be better explained (addressed in the methods). The calculation of the
statistics between model and validation data will be addressed in the methods section for a
better understanding. Only results of the statistics will then be given in the results section.

[revised manuscript text omitted]

The references to figures in the results section will be shortened for better reading and only the most appropriate ones will be cited. In addition we will clearly distinguish between results and discussion.

The contradiction of results with previous studies (e.g. Hasler et al., 2011) will be discussed in more detail in the discussion section. We will provide an explanation of possible factors leading to this contradiction and will connect our findings to findings in literature of steep rock wall temperatures (e.g. Hasler et al., 2011; Magnin et al., 2015; Myhra et al., 2015).

[revised manuscript text omitted]

**4. The references to the existing literature are not always consistent with the text. Some examples of inconsistencies between the text and the references are given in the specific comments, but not all of them. Please, consider this comment and verify your references all along the text.**

405

The references to the literature are often too generalized. Therefore we revised the references in the introduction and checked all references throughout the manuscript.

**5. Introduction: 1st paragraph is poorly written. First two sentences focus on rock wall**

410 **permafrost (with a strange way to use references) whereas the two other sentences, apparently aligned with the first two sentence mention the need to model permafrost with example from very different alpine permafrost terrains, that are not relevant to address the questions related to "rock wall permafrost". This must be improved to be more consistent and to better settle your study in its global research field.**

415

The first paragraph, as well as the whole introduction were rewritten in order to emphasize the need to study (measure and model) rock wall permafrost.

*Section 1, pages 1-3, new lines 32-90:*

420 *1. Introduction*

[revised manuscript text omitted]

**6. The study site is made of a NW and SSE faces (according to Table 1) named N and S face. Whilst naming N and S face is not a problem, it seems that these slopes are considered as real N and S facing slopes in the study (e.g. the apparently unexpected low difference in surface temperature, which is maybe not as low as suggested given the real aspect of the slopes). During revision, this should be taken into consideration to avoid scientific imprecision and straightforward conclusions.**

An additional table giving aspect and slope measured in 'reality' and in the model domain (based on the DEM) for 22 NSRT logger locations was inserted in the appendix. Please see the new Table *Appendix: Table 1A* in answer 4.
In addition the interpretation of measured and modelled data will be improved with a special focus on the real aspect (NW/SSE) of the rock walls.

*Section 4.4.1, page 12, new lines 450-453:*
*The topography driven difference of the measured mean MANSRT between the entire N and the entire S facing rock wall were 3.6/3.2 °C. Such a small deviation is reasonable when taking into account that the rock walls are facing rather NW and SE than N and S (Fig. 1a, Appendix Table 1A), as well as considering the accumulation of a thick snow cover at 7 of 11 locations in both the N and S slopes.*

**7. Lines 20-21: is this sentence written in proper English? It seems confusing.**

The sentence was deleted.

**8. Line 29: what does "large" mean? Some rock falls affected "narrow" rock faces, pinnacles, ridges... Is this word really appropriate?**

'Large' was deleted.

**9. Line 31: Davies et al. 2001 didn't not investigate the stability of permafrost in high Alpine regions but proposed a laboratory study under very specific conditions. Gruber et al. 2004a didn't not investigate rock wall stability, but only permafrost distribution. Are these references really appropriate?**

Davies et al. (2001) and Gruber et al. (2004a) were deleted.

**10. Line 35: the reference to Gruber, 2012 doesn't seem appropriate since the sentence focus on permafrost modelling in the European Alps and Gruber's work focused on global models.**

Gruber (2012) was deleted.

**11. Line 36: there are better examples than Fiddes et al. 2015 as numerical modelling of mountain permafrost (especially in rock walls).**

The reference Fiddes et al. (2015) refers to physics-based modelling of permafrost distribution in the European Alps. Since the first paragraph was rewritten in order to point out the need to study rock wall permafrost, other references were provided.

*Section 1, page 2, new lines 42-43:*
*New references: Gruber et al. (2004a), Noetzli et al. (2007) and Noetzli and Gruber (2009)*

**12. Line 41: Harris et al. 2009 paper does not focus on modelling transient changes in rock wall permafrost. Here again, better examples could be provided (e.g. only keep Noetzli et al. 2007 and move it at the end of the sentence, other examples could be added: e.g. Noetzli and Gruber 2009).**

Harris et al. (2009) was deleted. Noetzli et al. (2007) and Noetzli and Gruber (2009) were moved at the end of the sentence.

*Section 1, page 2, new lines 49-52:*
*Beside three-dimensional (3d) subsurface heat flow and transient changes in steep bedrock thermal modelling (Noetzli et al., 2007; Noetzli and Gruber, 2009), the strongly variable spatial and temporal rock surface boundary conditions therefore also need to be taken into account.*

**13. Line 46: "However this approach cannot capture..." Is it really because of the modelling approach that the small scale variability cannot be captured or because of the spatial resolution?**

Fiddes et al. (2015) cannot capture the small scale variability because of too coarse spatial resolution of the approach used. The sentence was deleted.

**14. Line 59: Gruber et al., 2004b and Gruber and Haeberli 2007 didn't really study the Snow control. The last reference, proposed some theories and hypotheses about the snow control, but not a study dedicated to its effect.**

The three references will be replaced with more appropriate references on snow control in steep rock walls, such as Haberkorn et al. (2015a,b), Magnin et al. (2015), Mhyra et. al. (2015) and Pogliotti (2011).

*Section 1, page 2, new lines 53-54 :*
*The influence of the snow cover on the rock thermal regime has recently been studied in steep bedrock (Haberkorn et al., 2015a,b; Hasler et al., 2011; Magnin et. al., 2015).*
*And new lines 66-73:*
*Those observations emphasize the need to account for the strongly varying snow cover in thermal modelling of steep rock walls. Myhra et al. (2015) and Pogliotti (2011) simulated the potential thermal effect of snow on steep bedrock temperatures, while changing snow depths arbitrarily in one-dimensional (1d) (Pogliotti, 2011) and two-dimensional (Myhra et al., 2015) numerical model runs. Both authors provided evidence of a considerable influence of snow on the rock thermal regime, but could not verify their results with measurements due to a lack of snow depth observations in steep rock walls. Nevertheless, the relative influence of snow on the rock thermal regime was evaluated by Pogliotti (2011) by comparing point simulations without snow to those with virtual snow.*

**15. Line 63: Pogliotti, 2011 focused on the snow control in steep rock faces similarly to the here presented study, but in 1D. He only proposed a review of the existing literature stating ablation processes in steep alpine rock faces, but did not study the gravitational processes directly such as suggested by this reference.**

Pogliotti (2011) was deleted.

**16. Line 65: Gruber et al. 2004a study considered ideal rock walls, not the kind of "natural" rock walls described in the text before the reference.**

Correct. Gruber et al. (2004a) will be replaced with references to studies dealing with snow in steep rock walls, such as Haberkorn et al. (2015a), Sommer et al. (2015) or Wirz et al. (2011).

*Section 1, page 2, new lines 48-49 :*
*Rock walls are, however, often variable inclined, heterogeneous, fractured and thus partly snow-covered (Haberkorn et al., 2015a; Hasler et al., 2011; Sommer et al., 2015).*

**17. Line 82: is "However," really the right term? It connects the starting sentence with the former sentence in the sense of "Nevertheless", and opposes the new sentence to previous statement. But the smoothed temperature difference between N and S face results of the warming/cooling effect of snow, it is a consequence. Could you consider this and revise your sentence accordingly to avoid confusion? Maybe there is an opposition between two sentences but it is not clear when reading.**

The sentence was deleted.

**18. Line 82-84: References are not consistent: do you mean that thick snow smoothes the variability of MAGST compared to snow free bedrock (Gruber et al., 2004b; Noetzli et al., 2007) or compared to bedrock with thin and intermittent snow (Hasler et al., 2011)? The sentence has to be more precise and the references better used.**

Please see answer 17.

**19.  Line 89: What is the difference between NRST and the "rock thermal regime"? Do you mean the thermal regime at depth?**

The rock thermal regime close to the surface and at depth is meant. We will rewrite the sentence accordingly.

*Section 1, page 3, new lines 85-89 :*
*The 1d modelling approach used by Haberkorn et al. (2015b) to investigate the influence of the snow cover on the rock thermal regime is therefore not sufficient, although the ability of the 1d SNOWPACK model (Lehning et al., 2002a,b; Luetschg et al., 2003; Wever et al., 2015) to simulate the effect of a snow cover on rock temperatures could clearly be demonstrated.*

**20. Lines 118-119: could you explain why did you choose this reference period? Data availability?**

The study period from 1 September 2012 to 31 August 2014 was chosen in order to present 2 years of complete meteorological input, as well as validation data (TLS, NSRT). This data is not provided in the text.

**21. Lines 123-125 could you at least tell when the iButtons were installed in order that the reader doesn't have to look for essential information into the referred paper.**

The iButtons were installed on 9 July 2012. In this study we focus on the investigation period from 1 September 2012 to 31 August 2014, which is stated in section 2. The date of installation of NSRT loggers will not be provided in the text, since it is not relevant for the reader.

**22. Line 127: here also I don't understand the meaning of "However,".**

'However' will be deleted and sentence rewritten for better understanding.

*Section 3.3, page 7, new lines 277-278 :*
*In addition to the spatial analysis, an absolute point analysis between measured and modelled NSRT evolutions has been carried out for 4 loggers (Section 4.3).*

**23. Lines 131-132 and 135-136: could you be more precise with the features that you describe? What is the difference in temperature amplitude between N7 and N3? How do you see the snow influence on S9?**

Lines 127-136 will be rewritten in order to describe better the snow/ no snow influence on NSRT. In addition it will be referred to new Fig. 7 to better show the features described.

*Section 3.3, page 8, new lines 284-287 :*
*Pronounced daily NSRT amplitudes indicate that N3 and R2 were generally snow-free (Figs. 7d, e). Although logger N7 and S9 are located in steep rock, wide ledges below allow the accumulation of a thick snow cover in winter, causing strong NSRT damping during the snow-covered period, as well as a zero curtain in spring (Figs. 7b, c).*

**24. Lines 183-184: it is difficult to understand the end of the sentence: "hence a constant upward ground heat flux is applied as the lower boundary condition". Please, could you reformulate and be more precise?**

The whole section 3.3 will be shortened (please see also answer 2 to referee 2). Therefore this sentence will be rewritten and the depth, as well as the magnitude of the geothermal heat flux at the lower boundary will be provided.

*Section 3.1.2, page 5, new lines 190-193 :*
*Although the geothermal heat flux is most likely negligible in the narrow, steep and complex Gemsstock ridge due to strong topographic (Kohl, 1999) and 3d thermal effects (Noetzli et al., 2007), a constant upward ground heat flux had to be applied. $Q_{ground}$ is assumed to be 0.001 W m$^{-2}$ at 20 m depth to ensure a marginal impact of the lower boundary condition on the analysed rock thermal regime close to the surface.*

**25. Lines 190: could you give an indication of the gap proportion in the meteorological data and of the bias induced by the gap filling procedure (even if information also exists in Haberkorn et al., 2015b)? Does the gap filling procedure induce a part of the bias in model results?**

Data from the AWS Gütsch, maintained by MeteoSwiss, were used for gap filling for all parameters of the meteorological data series of Gemsstock from 22 March to 15 April 2013, as well as for correcting the erroneous ISWR measured at Gemsstock between 1 September 2012 and 15 April 2013. For the corrected ISWR in 2012 the mean absolute error was 14.4 W m$^{-2}$, the mean bias error was 8 W m$^{-2}$ and the root mean squared error was 30.2 W m$^{-2}$. Calculated errors are reasonable, since the radiation sensor accuracy is ±20 W m$^{-2}$.

In order to parameterize ILWR between 1 September 2012 and 15 April 2013, a combination of a clear-sky algorithm developed by Dilley and O'Brien (1998) and a cloud correction algorithm from Unsworth and Monteith (1975) is applied. For the all-sky ILWR the mean absolute error was 26.8 W m$^{-2}$, the mean bias error was –6.3 W m$^{-2}$ and the root mean squared error was 31.9 W m$^{-2}$. Hence, parameterization errors are reasonable compared with the error range suggested e.g. by Flerchinger et al. (2009) for the combination of the Unsworth cloud correction and the Dilley clear-sky algorithm (root mean squared error of 27.1 W m$^{-2}$).

Gaps in snow depth data were filled based on similarity with data from adjacent stations using geostatistical interpolation tools. Snow depth from 10 surrounding IMIS AWS within a distance of 20 km and from AWS Gütsch served as correction data for the snow depth of Gemsstock. Stations are located in flat terrain and cover all directions to consider different air flows at each station. Detrended weighting procedures were applied to account for elevation differences between Gemsstock and the neighbouring stations.

The presented correction methods may are inappropriate to determine ISWR and ILWR exactly at one certain point in time, but are considered to be an acceptable solution for the input of an energy balance model running on a multi-annual timescale where the conservation of natural variability of the model input variables is much more important than the projection of single time steps. We think that gap filling only induces a minor part of the bias in model results and a meteorological error analysis is not within the scope of this study. All this information will therefore not be provided in the manuscript. The correction and error analysis are well documented in Haberkorn et al. (2015b).

**26. Lines 200-201: the reason for which the thermal parameters, especially those at depth (such as 100% solid content which is unusual in modelling rock wall thermal regime) have been chosen is not clear. The utility of these parameters to simulate rock wall surface temperature is not clear either. Could you be more precise about this points?**

Down to 0.5 m depth 99% solid and 1% pore space containing ice or water was assumed to account for near-surface fracture space. Between 0.5 and 20 m depth 100% un-fractured, solid rock was assumed. Further, it is not the scope of this study to model the influence of fractures on rock temperatures, which we addressed (but did not model) in Phillips et al. (2016a).

Although the geothermal heat flux is most likely negligible in the narrow and steep Gemsstock ridge, a geothermal heat flux (here: 0.001 W m$^{-2}$) had to be applied as lower

boundary condition of the model. To ensure a marginal impact of this boundary condition on the analysed rock thermal regime close to the surface, it was important to model deep into the rock. We chose to model down to 20 m depth, since detailed rock temperatures for initializing the model were available from an on-site borehole. Model uncertainties resulting from the use of the geothermal heat flux as the lower boundary condition were evaluated in 1d SNOWPACK test simulations at the borehole location at Gemsstock. Here, modelled rock temperatures accord well with borehole rock temperatures measured at various depths down to 15 m for both NE- and SW-facing locations ($r^2$ = 0.6–0.88). Correlation decreases with increasing depth, since modelled temperatures are biased by the geothermal heat flux. Consequently, simulated rock temperatures could be considered to depths of approx. 10 m at Gemsstock. This data however, will be presented elsewhere.

The physical properties of the granodiorite bedrock used for ground modelling are discussed in Haberkorn et al. (2015b). For consistency the same bedrock properties are applied.

**27. Line 233: Wouldn't be "one" instead of "an" in "an Alpine3D run"?**

The whole sentence was changed.

*Section 3.1.2, page 6, new line 231 :*
*Thirdly, model runs were carried out using scaled precipitation of each of the four TLS campaigns.*

**28. Lines 234-235: could you provide a concise overview of the results for the three other TLS. What "coincided best with validation data" means quantitatively?**

We will provide an additional figure (histogram) to justify the choice of one TLS as precipitation scaling input. Please see also answer 3 to referee 2.

**29. Lines 278-280: it is not really easy to report the mentioned results to the figure. On which data are the R2 and MBE calculated? You report to figure 2b and c to compare snow heights measured with TLS and modelled with Alpine3D, but those figures only show the measured snow depth. Also, a scatter plot would help the reader to better see the comparison between modelled and measured values.**

The results section 4.1 will be reorganized and rewritten, since it is confusing. Please see also answer 3. Subsection 4.1.2 will be deleted. Thus the sentences in line 278-280 will be deleted. For your clarification: the $r^2$ and the MBE were calculated between the measured snow depth (TLS) at 11 December 2013 and the scaled snow depth (precipitation scaling) at the same date.

In addition Fig. 2 will be reworked in order to provide more meaningful snow depth information. Subfigures 2b, c, d, e will be replaced and instead three subfigures each will be shown on: independent TLS data, differences between the independent TLS data and model

770    results at date of the TLS campaigns, as well as scatter plots of measured and modelled snow
depth.

*New Figure 4:*

[Figure]

775 *Figure 4. Snow depth distribution: (a-c) measured based on TLS, (d-f) modelled at the same dates as the TLS campaigns, (g-i) differences Δ between modelled and measured snow depth and (j-l) measured snow depths as function of modelled snow depths. Grey dots indicate the locations of NSRT loggers and selected ones are highlighted in pink and labelled.*

780 **30. Line 284: "for each NRST logger": it is only 4 loggers, right? Why other NRST loggers were not used (except that those used are enough to represent snow cover variability according to lines 127-128)? One could easily think that using more loggers could provide more robustness to the MBE and R2 analysis.**

785 It is correct that the more validation data the better in order to provide robustness to the statistics applied (MBE, $r^2$, new also MAE ). The MBE and $r^2$, as well as MAE error analysis between measured and modelled NSRT data was performed for each of the 22 NSRT loggers (new section 4.4, new Fig. 9, Table 3), but only NSRT and snow depth data of 4 loggers (representative for typical snow conditions in the rock walls) were presented in detail (new
790 section 4.3, new Figs. 7, 8, new Table 1), since providing data from all loggers would be too much. This will be clearly stated in the text and the appropriate sections (last section in introduction> Section 1, methods new Section 3.3, results > new Section 4.3 and 4.4) will be reworked accordingly, since it seems that the use of all 22 NSRT loggers was not clear for the reader. For instance in Table 3 data of all 22 NSRT loggers are used. For changes in Section 1,
795 new Section 3.3, new Sections 4.3 and 4.4, please see answer 3.

**31. Lines 288-292: this paragraph is not clear either. "four independent TLS", but one of them was used to scale the snow accumulation (11.12.2013), right? So, is it true to say "independent". "R2=0.95": which data were used for this calculation: modelled versus**
800 **measured snow height for each grid cells and for each TLS survey? Why to show results from 11.12.13 if those data are used for scaling (and are therefore not independent)? Could you show a scatter plot or at least better illustrate the model output by e.g. replacing one of the 3D or 2D view in Figure 2?**

805 Correct. The TLS of 11 December 2013 was used to scale the precipitation for model input. Only three independent TLS are therefore available for model validation. This will be changed in the text.
The $r^2$ = 0.95 is a mean calculated between modelled and measured snow depth for each grid cell and each of the TLS. MBE, $r^2$ and MAE (please see answer 6 to referee 2) between
810 the modelled and the independent measured snow depth data of each of the three independent TLS campaigns will be provided in the revised manuscript. In addition Fig. 2 will be revised (please see answer 29 and new Figure 4).

815 *Section 4.1.2, page 9, new lines 326-332:*
*The evaluation of the snow depth distribution modelled using Alpine3D (Figs. 4d-f) against data from three independent TLS revealed a reasonably well reproduced snow depth*

*distribution with $r^2$ = 0.52-0.95 (Figs. 4j-l), while absolute snow depth differences were in the range of +1.5 m to -1 m (Figs. 4g-i). Considering the area around NSRT locations modelled snow depths are often underestimated (Fig. 5), whereas they are overestimated while averaging over the entire model domain (MBE = 0.31 − 0.81 m). The MAE of the N and S slopes varied between 0.47 − 0.77 m and always indicated higher deviations between snow depth observations and predictions in the heterogeneous N facing slope (Table 2).*

**32. Line 296: Is the term "validation" really appropriate?**

‚Validation' is correct, but only for the three independent TLS campaigns. This will be rewritten in the text and figure caption.

*Section 4.1.2, page 9, new lines 335-336:*

*The modelled winter snowpacks are compared to measured TLS snow depths (markers in Fig. 5) on the dates of TLS campaigns.*

*New Figure 5 and the adopted figure caption:*

[Figure]

*Figure 5. Snow depth evolution (lines) measured at the flat field AWS Gemsstock (AWS), as well as modelled at the NSRT locations discussed in detail (N7, S9, N3, R2). Snow depths at the NSRT locations obtained by TLS are shown as blue, red, grey dots and pink markers. The locations of N3 and R2 lack snow for the entire investigation period. Data of the TLS campaign on 19 December 2012 is also shown here, although the measured snow depth was used for precipitation scaling.*

**33. Lines 299-301: here you give a reason for misfit between modelled and measured values. The same explanation could be expected lines 290-292: where the under/overestimations are coming from? Modelling of ablation? If it is given in the discussion, the same should be done for these lines 299-301. If you make the choice to directly discuss your results, an explanation could be expected lines 290-292.**

The misfit between modelled and measured snow depths (lines 299-301) is also valid for lines 290-292. Possible explanations for model uncertainties are again presented in the discussion section. Hence, the results will be presented without any assessment or interpretation of results. Lines 299-301 will therefore be deleted.

**34. Lines 300-301: Here the modelled snow depth for the S measured point does not fit the measured values. A 1 m difference may have huge implications for the NRST simulations. How is that taken into account?**

Although measured and modelled snow depth differences were > 0.5 m (especially on the S slope), these snow depth differences do not affect the rock thermal regime as long as snow depths are > 0.2 m. Steep, bare rock is decoupled from atmospheric influences for snow depths exceeding 0.2 m (Haberkorn et al. 2015a). Amongst others, Luetschg et al. (2008) and Zhang (2005) stated that the influence of snow depth variations on ground temperatures in the presence of a thick snow cover are small, whereas snow depth variations only have strong effects on the ground thermal regime for thin snow cover (in steep, bare rock we found the threshold to be 0.2 m).

Of course such big snow depth differences might have an effect on the snow cover *duration*. However, both snow cover timing and duration were reproduced nicely by the model, which can be observed comparing measured and modelled NSRTs in new Fig. 7b, c and new Table 1 (Table1 and Table 2 have been merged > see snow cover duration in new Table 1). It has been shown repeatedly that realistically modelled snow cover duration over the winter is more important than accurately modelled snow depths at certain points in time (e.g. Fiddes et al. 2015; Marmy et al. 2013).

While absolute snow depths were underestimated by Alpine3D, the well modelled snow cover duration implies an underestimation of snow melt in the model. This may be at least partly explained with the 1d snow module which does not account for 3d heat flow between adjacent snow-free and snow-covered rock portions, as well as micro-meteorological processes due to uneven heating during the ablation period which accelerate snow melt in

reality. We will amend the manuscript regarding this issue. The whole topic is addressed in the Discussion.

*Section 5.1, page 14, new lines 535-565:*

*In this study discrepancies in modelling absolute snow depths in steep rock walls are evident (Figs. 4, 5). This is a consequence of the linear precipitation scaling algorithm used here. Snow settlement is calculated for snow depths at the AWS location and is then linearly scaled into the rock walls, but snow depths and the meteorological forcing obviously differ between the flat field AWS and the rock walls. This causes the snowpack to settle differently and in a non-linear manner. Differences in settling calculated at the AWS and for the grid points in the Alpine3D model domain therefore cause absolute snow depth errors. However, on the basis of measured NSRTs (Figs. 7b, c) it is evident that the snow cover duration (Table 1) is well reproduced by the model. The realistically modelled snow cover duration over the winter was found to be more important for modelling the ground thermal regime than accurately modelled absolute snow depths at certain points in time. This agrees with the findings of Marmy et al. (2013) and Fiddes et al. (2015). Although measured and modelled snow depth differences were >1.0 m (Figs. 4, 5), these snow depth differences do not affect the rock thermal regime since steep, bare rock is already decoupled from atmospheric influences at snow depths >0.2 m (Haberkorn et al., 2015a). Amongst others, Luetschg et al. (2008) and Zhang (2005) stated that the influence of snow depth variations on ground temperatures in the presence of a thick snow cover are small, whereas snow depth variations only have strong effects on the ground thermal regime for snow thinner than 0.2 m.*

*As a consequence of the strong snow depth variability in the rock walls snow depth comparisons at specific points are difficult. Although, verification of snow depth over the entire rock walls suggest an overestimation of snow depth (Table 2, Figs. 4g-i), snow depths were underestimated locally by Alpine3D, e.g. at NSRT locations (Fig. 5). The efficiently modelled snow cover duration at NSRT locations thus implies an underestimation of snow melt in the model. This agrees with an underestimation of surface heat fluxes (e.g. shortwave incoming radiation), reflected in too low modelled NSRTs (dT in Figs. 7d, e) and consequently MANSRTs (MBE in Table 1) at locations lacking snow and during the snow-free period. A likely explanation is that both air temperature and wind speeds, measured at the flat field AWS may be poorly representative for the prevailing conditions in the rock walls and therefore turbulent flux simulations are biased. In addition, the underestimation of snow melt may also be partly explained by the 1d snow module which does not account for lateral heat flow between adjacent snow-free and snow-covered rock portions, as well as micro-meteorological processes due to unevenly distributed heating during the ablation period which in reality accelerates snow melt. Nevertheless, the model verification showed that the overall performance of Alpine3D modelling snow depths and consequently rock temperatures in steep slopes in the current setup provides useful improvements compared to the common assumption of a lack of snow in thermal modelling of idealized rock walls exceeding 50° (e.g. Fiddes et al., 2015; Gruber et al., 2004a; Noetzli and Gruber, 2009; Noetzli et al., 2007).*

*Please see also new Table 1: Topographic characteristics of selected NSRT logger locations with different snow conditions and the distance to the nearest ledge below (DLB). In addition*

*analysis of observed (O) and predicted (P) snow cover duration, as well as observed MANSRT and predicted MANSRT for both snow-covered (PS) and snow-free (PSF) scenarios at selected NSRT locations and for the years 2012-2013 (12-13) and 2013-2014 (13-14). The MBE and MAE were calculated between observations and model predictions of the snow-covered respectively snow-free scenarios.*

| Logger (slope/aspect) | DLB (m) | Year | Snow cover duration O | P | MANSRT [°C] P | PS | PSF | MBE [°C] PS | PSF | MAE [°C] PS | PSF |
|---|---|---|---|---|---|---|---|---|---|---|---|
| N7 (90°/289°) | 0.1 | 12-13 | 10 Oct-8 Jul | 13 Oct-4 Jul | 0.1 | 0.9 | -2.8 | 0.8 | -3.0 | 1.1 | 5.2 |
| | | 13-14 | 7 Oct-11 Jun | 11 Oct-13 Jun | 0.5 | 1.2 | -1.9 | 0.8 | -2.4 | 1.0 | 3.7 |
| N3 (90°/284°) | 10 | 12-13 | - | - | -1.4 | -3.6 | -3.6 | -2.1 | -2.1 | 2.3 | 2.3 |
| | | 13-14 | - | - | -0.8 | -2.5 | -2.5 | -1.7 | -1.7 | 2.2 | 2.2 |
| S9 (72°/165°) | 0 | 12-13 | 28 Oct-6 Jul | 31 Oct-12 Jul | 2.4 | 2.1 | 0.4 | -0.3 | -2.1 | 0.6 | 5.3 |
| | | 13-14 | 4 Nov-11 Jun | 9 Nov-20 Jun | 2.7 | 2.3 | 1.3 | -0.4 | -1.4 | 0.6 | 4.0 |
| R2 (58°/164°) | 15 | 12-13 | - | - | 2.2 | -0.3 | -0.3 | -2.5 | -2.5 | 2.8 | 2.8 |
| | | 13-14 | - | - | 2.7 | 0.6 | 0.6 | -2.1 | -2.1 | 2.6 | 2.6 |

**35. Lines 342-344: why such a big bias (-2_C)? What is its implication in the overall results?**

Likely explanations of model uncertainties are given in the discussion. Especially at locations lacking snow the underestimation of modelled NSRT may result from both air temperature and wind speed differences between rock walls and the flat field AWS. Air temperature and wind speed measured at the AWS may be a poor surrogate for the prevailing conditions in steep rock. Hence the turbulent flux simulations are biased (provided in discussion lines 480-483). Further, also differences in slope and aspect between the model domain and reality can be a possible error source (Please see the new Table *Appendix: Table 1A* in answer 4).

*Section 5.1, page 14, new lines 554-558:*

*This agrees with an underestimation of surface heat fluxes (e.g. shortwave incoming radiation), reflected in too low modelled NSRTs (dT in Figs. 7d, e) and consequently MANSRTs (MBE in Table 1) at locations lacking snow and during the snow-free period. A likely explanation is that both air temperature and wind speeds, measured at the flat field AWS*

*may be poorly representative for the prevailing conditions in the rock walls and therefore*
940 *turbulent flux simulations are biased.*

*And Section 5.1, page 15, new lines 579-583:*
*Finally, difficulties in partitioning the measured incoming shortwave radiation in a direct and diffuse component, particularly for low sun angles, may explain the stronger modelled net*
945 *radiation for snow-free conditions in the shaded (Fig. 6b) than in the sun-exposed slope (Fig. 6d), which is amplified by differences in slope and aspect between the model domain and reality (Appendix Table 1A).*

**36. Lines 362-364: the difference is calculated using the 30 NRST time series?**
950

Correct. This is mentioned in original lines 357-359. The text will be clarified, since it might not be clear.

*Section 4.4, pages 11-12, new lines 444-447:*
955 *A comprehensive analysis of all 22 NSRT locations was used to evaluate the spatial performance of Alpine3D in modelling the potential effect of snow on NSRTs. Both the measured and modelled NSRT data of all 11 N facing locations and of all 11 S facing ones were used to calculate means of MANSRT, MBE and MAE over the individual N and S facing rock walls (Table 3).*
960

**37. Lines 363-364: not as high as expected for "real" North and South walls, which is not the case here, with rather NW and SE faces. This must be taken into consideration!**

We will clarify the text. Although NW and SE facing rock slopes are considered, the NSRT
965 differences are still smoothed due to thick snow.

*Section 4.4.1, page 12, new lines 450-453:*
*The topography driven difference of the measured mean MANSRT between the entire N and the entire S facing rock wall were 3.6/3.2 °C. Such a small deviation is reasonable when*
970 *taking into account that the rock walls are facing rather NW and SE than N and S (Fig. 1a, Appendix Table 1A), as well as considering the accumulation of a thick snow cover at 7 of 11 locations in both the N and S slopes.*

**38. Lines 392-399: figure 6 deserves much more description and precision. Could you be**
975 **more precise in the text with the 1.9_C? For what? Entire model domain? North face.? South face? Measurement point controls?**

To do so we will rewrite new Section 4.4.3, also for better understanding.

980 *Section 4.4.3, pages 12-13, new lines 480-501:*

*The influence of the snow cover on rock surface temperatures and the previously discussed rock temperature results are summarized in Fig. 10. Here modelled MANSRT for each grid cell of the entire model domain of the Gemsstock ridge (not just at selected NSRT locations) are shown for both the snow-free (Figs. 10a, b) and the snow-covered scenario (Figs. 10c, d) for*

985    *the year 2012-2013, as well as their differences (Figs. 10e, f). Pronounced MANSRT deviations between both scenarios are obvious.*

*Under snow-free conditions the mean MANSRT averaged over the entire N slope are -2.9 °C in 2012-2013 and -1.9 °C in 2013-2014 and thus clearly indicate a possible occurrence of permafrost in the rock walls under snow-free conditions. Mean MANSRTs averaged over the*

990    *entire S facing slope are -0.3/0.7 °C and therefore correspond to conditions at the lower fringe of permafrost occurrence. The MANSRT variability within the slopes is more homogenous compared to the snow-covered scenario, since rock temperatures mainly depend on topography and thus solar insolation.*

*In contrast to the snow-free scenario, the accumulation of a heterogeneously distributed*

995    *snow cover strongly changes the conditions at the rock surface and thus rock temperatures. In the snow-covered scenario, MANSRT variability is pronounced in steep rock walls depending on the accumulation of a continuous snow cover, on snow depth and snow cover duration. The snow depth distribution varies strongly due to the complex micro-topography in the rock walls with rock portions accumulating thick snow in close vicinity to rock portions*

1000    *lacking snow. MANSRTs were highest at the foot of both rock walls and gradually decreased from flat to steeper areas due to both snow depth decrease and low insolation in the N slope at locations without snow. MANSRT at locations shadowed by rock outcrops or in rock dihedrals were colder compared to their surrounding areas (arrows in Figs. 10c, d). The influence of the snow on rock surface temperatures is emphasized by 2.5/1.8 °C (N),*

1005    *respectively 2.3/1.3 °C (S) higher modelled MANSRTs averaged over the individual N and S facing slopes for snow-covered, than for snow-free conditions.*

The value of 1.9 °C (this value has now changed) is the difference of MANSRT averaged over the whole model domain (taking into account each pixel regardless of aspect) between the

1010    modelled snow-covered and the modelled snow-free scenario. For the snow-free scenario the precipitation input of the model was forced to be zero (explanation in new Section 3.2).

*Section 3.2, page 7, new lines 250-257:*

*A sensitivity study is performed in order to assess the bias made while neglecting snow in*

1015    *thermal modelling of steep rock walls, which has often been done for ideal, compact rock walls with slope angles >50° (e.g. Fiddes et al., 2015; Gruber et al., 2004a; Noetzli and Gruber, 2009; Noetzli et al. 2007). The sensitivity study comprises a rock temperature comparison between a model run with snow (precipitation input from precipitation scaling) and a model run without snow. For the model run without snow, precipitation input was*

1020    *forced to be zero. Alpine3D simulations were thus carried out for two contrasting scenarios in the rock walls (Fig. 2): one accounting for snow accumulation (henceforth referred to as*

*'snow-covered' scenario) and one neglecting snow (henceforth referred to as 'snow-free' scenario).*

**39. Lines 425ff: here the energy balance of snow free N7 is presented like absolutely different from snow covered N7 ("In contrast to", "differed strongly") . However, when looking at Figure 7, the pattern of Qnet seems quite similar, only the magnitude differs, Qsensible differs in a certain degree. Are the terms really appropriate?**

Comparing the energy balance of both snow-covered and snow-free (see explanation above) scenarios at location N7 reveals important differences especially in May, June and July (ablation period): for snow-covered conditions (new Fig. 6a) all available energy is used to melt the snow, indicated by the snow melt term $Q_{melt}$. In contrast during the same period all available energy is used to directly warm the rock assuming snow-free conditions (new Fig. 6b), which is then compensated by the sensible heat flux. These differences in energy fluxes between snow-covered and snow-free scenarios result in totally different NSRT evolution. In addition different albedo effects arise between both scenarios (snow versus bare rock).

**40. Line 367: "effects: In" is either "effects. In" or "effects: in"**

Changed.

**41. Line 447: why not modelling heat transfers in fractures is a limit of your model? Are you also modelling the interior of the ridge? If not, please remove, there are already enough details to discuss. If yes, it should appear clearly all along the text that you do not only model surface temperature and substantial results on the model temperature at depth must be provided!**

'Water flow along fractures' is removed. In addition the first paragraph is rewritten.

*Section 5.1, page 14, new lines 528-534:*
*Limitations in reproducing snow cover characteristics, energy balance components and rock temperatures in the simulations were introduced by uncertainties in the input data (see Section 3), as well as by the adequacy of the process representation in the Alpine3D model. Some physical processes, such as lateral heat fluxes at the rock surface (in our grid model heat fluxes are calculated perpendicular to the rock surface, all other fluxes are lateral) or through the narrow ridge and the heterogeneous wind field in extremely steep terrain, are currently insufficiently represented by our model setup. Some model uncertainties and their consequences on the modelled rock thermal regime of steep rock walls are discussed below.*

**42. Line 450: the consideration of snow cover at the ground surface is especially important to model small scale temperature variability. Some studies have shown that equilibrium**

**temperature fields and long-term changes can mainly consider air temperature and solar radiation in steep slope. Please, rework the sentence accordingly.**

It is true, that in near-vertical, ideal, snow-free rock faces air temperature and solar radiation might be sufficient to model ground surface temperatures. However, in fractured, structured and variably inclined rock faces this is not the case and the snow has to be taken into account, as already stated e.g. in Haberkorn et al. (2015a,b) or Magnin et al. (2015). We deleted the sentence in lines 450-451 and addressed this topic in the introduction.

*Section 1, page 2, new lines 42-52:*
*Numerical model studies simulating rock temperatures of idealized rock walls have been realised e.g. by Gruber et al. (2004a), Noetzli et al. (2007) and Noetzli and Gruber (2009). These studies assumed a lack of snow in steep rock exceeding slope angles of 50°, which is based on the general assumption that wind and gravitational transport (avalanching or sloughing) remove the snow from steep rock exceeding 50°-60° (e.g. Blöschl and Kirnbauer, 1992; Gruber Schmid and Sardemann, 2003; Winstral et al., 2002). They therefore suggested that air temperature and solar radiation are sufficient to model rock surface temperatures in near-vertical, compact, homogeneous rock walls. Rock walls are, however, often variable inclined, heterogeneous, fractured and thus partly snow-covered (Haberkorn et al., 2015a; Hasler et al., 2011; Sommer et al., 2015). Beside three-dimensional (3d) subsurface heat flow and transient changes in steep bedrock thermal modelling (Noetzli et al., 2007; Noetzli and Gruber, 2009), the strongly variable spatial and temporal rock surface boundary conditions therefore also need to be taken into account. The spatially variable snow cover is one of these driving factors.*

**43. Lines 456-457: the statement is interesting (it appear more important to correctly model snow timing to better represent snow effect) but could you at least provide one example in order to help the reader to connect this discussion to the results?**

We will cite new Fig. 7b, c. Here it is shown that modelled and measured NSRT are in good agreement, although absolute snow depths vary by around up to 0.9 m. Please see also answer 34.

**44. Line 461: again, the references are not adapted to the text: Gruber et al., 2004 andNoetzli et al., 2007 do not propose a "traditional snow modelling technique".**

We will rework this sentence. Gruber et al. (2004a) and Noetzli et al. (2007) do not account for snow in idealized slopes > 50°.

*Section 5.1, page 14, new lines 561-565:*
*Nevertheless, the model verification showed that the overall performance of Alpine3D modelling snow depths and consequently rock temperatures in steep slopes in the current setup provides useful improvements compared to the common assumption of a lack of snow in thermal modelling of idealized rock walls exceeding 50° (e.g. Fiddes et al., 2015; Gruber et al., 2004a; Noetzli and Gruber, 2009; Noetzli et al., 2007).*

**45. Lines 485-486: this belongs to the results, so move in another section and connect it with the presented results. On which source of data is this calculated? What is the difference with other presented MBE (e.g. MBE of -2_C line 342)?**

This sentence will be moved to the new results section 4.4.1 and will be reworked for better understanding. In addition we will refer to Table 3.

The MBE analysis was calculated between measured and modelled NSRT data for each of the 22 NSRT logger locations. The average MBE was than calculated for the entire N and S facing slopes while averaging all MBE of N facing locations and averaging all MBE of the S facing locations. Hence the average MBE error of -0.2 °C in the N slope and of -1 °C in the S facing slope include all N and S facing locations (22) and thus account for the various snow conditions in the rock walls.

The MBEs addressed in new lines 400-402 are only calculated between measured and modelled NSRT of one N (N3) and one S (R2) facing location lacking snow.

*Section 4.4.1, page 12, new lines 454-461:*

*At the corresponding 22 grid cells, the modelled mean MANSRT difference for the snow-covered scenario across the entire N and S facing slope is 2.6/2.3 °C and thus around 1.0 °C lower than the measured values (Table 3). This is mainly caused by too low modelled NSRTs and thus MANSRTs, especially in the sun-exposed rock wall during snow-free periods (Fig. 9) and at locations without snow (N and S slopes) resulting in a MBE of -1.3/-1.0 °C. These results are supported by the model verification at the single locations in Section 4.3.2, but clearly show that model uncertainties increase on the rock wall-scale due to the pronounced spatial variability. Uncertainties while applying Alpine3D to simulate NSRT in steep rough rock implies a MAE of 1.6/1.7 °C for both the entire shaded and sunny rock wall.*

**46. Lines 504-505: Isn't the difference of snow free/snow covered faces between N/S apects in the range of model uncertainty?**

The model uncertainty modelling rock temperatures averaged over the entire N and S facing rock walls result in a MAE of up to 1.7 °C.

*Section 4.4.1, page 12, new lines 460-461:*
*Uncertainties while applying Alpine3D to simulate NSRT in steep rough rock implies a MAE of 1.6/1.7 °C for both the entire shaded and sunny rock wall.*

In both slopes the MANSRT difference between snow-free and snow-covered scenarios varies between 1.3 – 2.5 °C (see also new Fig. 10e, f) and therefore close to the model uncertainty range. However, the differences given are calculated between modelled snow-free and modelled snow-covered scenarios (Fig. 10) and hence differences are relative. For

both the point and spatial scale snow-covered scenarios are always warmer than snow-free scenarios.

*Section 5.2, page 15, new lines 601-604:*

1150 *The modelled MANSRT increase of 1.3 – 2.5 °C found for both snow-covered N and S facing steep rock walls compared to snow-free simulations (Figs. 10e, f) is in the same order of magnitude than the cooling or warming effect of snow on mean-annual ground surface temperatures modelled by Pogliotti (2011).*

1155 **47. Lines 515-520: these appear as important results that would confirm recent findings in Scandinavian rock walls (Myhra et al., 2015): rock walls favour the presence of permafrost (here in the Alps, that would be especially true for North slopes?). This must be better emphasized.**

1160 This section will be better emphasized. Accordingly, the results will be improved and rewritten.

*Section 5.2, page 16, new lines 623-634:*
*In steep S facing mountain ridges up to 3000 m a.s.l., permafrost is most likely absent*
1165 *independent of the evolution of a thick snow cover, as shown in Figs. 10b and d. In contrast in steep rugged N facing rock walls the accumulation of a thick snow cover prevents a continuous permafrost distribution (Fig. 10c), while permafrost would most likely be present in areas without or with only thin snow (Fig. 10a). These results confirm recent two-dimensional numerical simulations made for east/north-east facing Scandinavian rock walls*
1170 *by Myhra et al. (2015), who found that the size of snow-free rock portions are crucial for warming or cooling a rock wall. In addition, these authors show that the existence of permafrost in steep bedrock varies strongly depending on thickness and extension of an insulating snow cover, which can lead to permafrost temperature increase and taliks in steep slopes. We therefore suggest that in recent permafrost distribution assessments in the*
1175 *European Alps based on energy balance (Fiddes et al., 2015) or statistical modelling (Boeckli et al., 2012a,b) mean annual rock surface temperatures were possibly modelled too low by around 2 °C in steep bedrock as a result of neglecting snow.*

**48. Lines 541-542: reaching that stage of the paper, the use of 30 NRST logger is still not**
1180 **clear: where the validation is shown? In figures, only 4 loggers are used and discussed. Same question as previously: is "validation" really appropriate?**

Please see answer 3 and 30. 'Validation' is correct, since independent measured NSRT data is compared to modelled NSRT data. In addition the error analysis is also based on this data.

1185

**49. Line 553: "50_", how this threshold has been defined? It appears for the first time in the conclusion.**

1190 This topic is addressed in the introduction (old lines 60-63). It was in general assumed that wind and gravitational transport remove the snow from steep rock in slopes exceeding 50 to 60°. Please see also answer 44. The threshold of 50° will be stated in the last part of the introduction for better understanding.

*Section 1, page 3, new lines 105-110:*

1195 *Thus the potential error induced by neglecting the snow cover in steep rock face thermal modelling for slope angles >50° can be estimated. This is necessary, since it has in general been assumed that wind and gravitational transport remove the snow from steep rock in slopes >50–60° (e.g. Blöschl and Kirnbauer, 1992; Gruber Schmid and Sardemann, 2003; Winstral et al., 2002) and rock temperatures were often modelled without snow for idealized*
1200 *rock walls >50° (e.g. Gruber et al., 2004a; Noetzli and Gruber, 2009; Noetzli et al. 2007).*

**50. Line 554: is "accurately" really appropriate when significant bias have been displayed?**

Sentence was reworked.
1205

*Section 6, page 17, new lines 655-658:*
*In the rough rock walls, the heterogeneously distributed snow cover was moderately well reproduced by Alpine3D with absolute snow depth differences varying between +1.5 and -1.0 m and a MAE between 0.47 and 0.77 m averaged over the entire rock walls. However, the*
1210 *snow cover duration was well reproduced by the model and proved to be most important for realistically NSRT modelling.*

**51. Lines 569-571: this is an interesting result but it has only be mentioned in the discussion. No quantitative information nor graphical results are provided for such**
1215 **statement. Either remove from the conclusion and remain as close as possible of your major findings, or develop the results related to grid-scale sensitivity analysis.**

Another short chapter will be provided in the results, as well as new Table 4 providing simulation results of 0.2 m, 1 m and 5 m in order to prove this statement. In addition the loss
1220 of topographic accuracy is additional for 1 m and 5 m grid cell size in the Appendix > new Table 1A. To avoid repetitions in the discussion chapter 5.3, this chapter was cut and merged to chapter 5.2.

[revised manuscript text omitted]

**52. Figure 1c: what are the peaks between 2930-2950 and 161750-161780 on the y and-x axis respectively? They look like artefacts in the DEM. How did you clean up the points cloud before generating the DEM? Furthermore, I the figure could be improved by including a hillshade below the elevation colour scale to improve the visibility of micro-topography.**

1280

The peaks in Fig. 1c are artefacts in the DEM due to the projection of overhanging rocks. Fig. 1c and 1d will be removed (comment 87 of referee 2) and replaced by a profile through the ridge for a better overview of the linear logger layout, elevations and slope angles. In Fig. 1a slope angle colours will be displayed in black and white to improve the visibility of the micro-topography. Fig. 1a then resembles a hillshade with an illumination angle of 90°.

1285

*New Figure 1:*

[Figure]

*Figure 1. The Gemsstock study site: (a) The Alpine3D model domain with slope angles (red rectangle) based on TLS data, as well as the locations of the AWS and the NSRT devices. The location of Gemsstock in the Swiss Alps is shown in the top left inset. (b) 3d view of the DEM*

1295  *of Gemsstock, as well as (c) the cross-section of the Gemsstock ridge with all 30 NSRT locations. Photographs showing the (d) N and (e) S rock faces and the measurement set-up. (b-e) Black dots indicate the locations of the 30 NSRT locations and selected ones, discussed in further detail are highlighted in pink and labelled.*

1300  **53. Figure 2a: it is very difficult to read the legend, could you make it bigger?**

Figure 2a is now new Figure 5. Please see this figure in answer 32.

**54. Figure 3: This figure must be improved. I propose the following modification for better**
1305  **clarity and readability. The legend: measured NRST and the measured-modelled NRST have the same line colours. Make different colour. Some lines are dashed or dotted but this does not appear in the legendOf course, the reader can then easily find out which line in the legend corresponds to which line in the graph, but it is confusing at first glance and does not support rapid overview of the Figure: make the legend consistent with the line**
1310  **style. The measured-modelled NRST is not shown at an appropriate scale. Why not displaying these differences in independent plots below the model output?**

The line colours of the measured NSRT and the measured-modelled NSRT have different blue shades. They might be difficult to distinguish. Therefore we will change line colours. In
1315  addition we will modify the legend and will provide a legend which is consistent with the line style.
The measured-modelled NSRT will not be moved to an independent plot, since these graphs shall only provide a quick overview on differences between measured and modelled results.

1320  *New Figure 7:*

[Figure]

*Figure 7. (a) Daily mean air temperature at the AWS Gemsstock. (b-e) Measured and modelled daily mean NSRT are shown for four selected locations in the N and the S facing rock walls representing typical snow conditions (snow, no snow). At locations accumulating snow (N7, S9) modelled NSRTs are shown for both the snow-covered and the snow-free scenarios, while the NSRT differences (dT) were only shown between measured and modelled snow-covered conditions. At locations without snow (N3, R2) measured and modelled NSRT differences (dT) are also shown.*

**Answers to Referee 2:**

**1. The approximate use of technical terms as well as of references (often totally wrong!) denotes the scarce attention paid by writing the introduction chapter. I suggest to the authors to deeply review this chapter by checking carefully the references (all along the paper!)**

The introduction is totally reworked and references checked all along the manuscript. Please see also answer 4 and 5 to referee 1.

**2. Sections 3.3.1 and 3.3.2 can be merged and shortened (mainly 3.3.1) by providing less detail about Alpine3D and SNOWPACK that are well known and documented models.**

Sections 3.3.1 and 3.3.2 in order to avoid repetitions. In addition we will merge sections 3.3.1 and 3.3.4. The surface energy balance is a core element of Alpine3D and belongs in the description of the energy balance model. Apart from the changes mentioned, the model description is already concise. Subsections 3.3.1 and 3.3.2 will still be treated separately for a better overview.

*Section 3.1.1 and 3.1.2, pages 4-5, new lines 135-196:*

*3. 1 Distributed energy balance modelling*

*3.1.4 The Alpine3D model*

[revised manuscript text omitted]

1425 **3. The precipitation scaling is a very promising idea but it does not seem to work very well as it is. It would be very interesting to understand why in 3 of the TLS campaign does not work providing quantitative analysis of these discrepancies (see technical comment). Moreover, looking at figure 2 is evident that it works quite well on the validation point N7 but is scarce at point S9. In my opinion a simple ratio between AWS snow-depth and TLS**
1430 **snow-depth is a too simplistic approach and represents the main limitation of the present study. I suggest the authors to put together all the TLS campaign data and AWS snow-depth data and try a more complex statistical approach which includes at least the topographical characteristics (ele, slp, asp) and doy (day of year) of the cells as scaling predictors. A first attempt could be to build a linear model with all the predictors, run a**
1435 **stepAIC on it for selecting the significant ones and use the resulting regressive model to scale the precipitation.**

We agree with the referee that the comparison of the modelled to the measured snow depth data clearly showed discrepancies in modelling absolute snow depths. However, snow
1440 depth distribution and especially snow cover duration are reproduced nicely by the model. Well reproduced snow cover duration was found to be most important for modelling the ground thermal regime (e.g. Fiddes et al. 2015; Marmy et al. 2013), which becomes obvious in new Fig. 7b, c. Please see also answer 34 to referee 1.
Although a quantitative analysis of the precipitation scaling approach is currently being
1445 evaluated (Voegeli et al., submitted) and beyond the scope of this contribution, we performed some additional analysis. This has been done since both referees expressed concerns about the discrepancies resulting from precipitation scaling. An additional figure (histogram) is provided (new Fig. 3) in the methods section in order to justify the choice of one TLS used for precipitation scaling. Please see new Fig. 3, as well as the figure attached to
1450 this response letter (Fig. 1 for revision). Here solid lines illustrate the distribution of the ratio modelled/measured snow depth for the 4 TLS available. The TLS of 11 December 2013 (20131211, pink line) is centred by 1 (since this TLS was used for precipitation scaling). Snow depth is underestimated for the other 3 TLS campaigns, while using the TLS of 11 December 2013 for precipitation scaling. Based on the solid lines in the figure attached we think it

1455 might be better to use snow depths derived from the TLS 19 December 2012 for precipitation scaling.

Dashed lines in the figure attached show an intercomparison between each TLS. First each pixel is corrected with the mean value of the TLS. Thus the relative snow depth per scan is calculated. Then the ratios of the relative snow depths of each TLS are compared to the 1460 other scans. For each pixel a ratio of 1 would imply that the ratio with the mean value is constant between TLS campaigns. Hence one can consider this to be the best possible result while building a statistical model. While comparing the envelope of the dashed and solid lines it becomes obvious that the scatter of the dashed lines is similar or larger than the precipitation scaling approach, especially for high-winter TLS. The scatter of the envelope is 1465 too wide to build a representative statistical model. We therefore come to the conclusion that the precipitation scaling is currently the best possible method to introduce varying snow depths into the rock walls. It is also clear that the method is not perfect, but we consider this future research to improve.

In addition it has been shown repeatedly (e.g. Lehning et al., 2011) that small-scale tatistical 1470 modelling of snow depth based on terrain parameters does not work very well. This is why we decided to use the scaling approach based on the measured snow distribution. We will provide a similar Figure (new Fig. 3) like those attached (Fig. 1 for revision) and additional discussion regarding this point in the revised manuscript. Further a new Table (new Table 2) is added providing the MBE and MAE for better snow depth validation. Since simulations 1475 were now performed with the precipitation scaling input based on the TLS 19 December 2012, the new Figures 4 to 10 had to be adopted and contain all new modelled data!

*Section 3.1.3, pages 6-7, new lines 231-247:*

*Thirdly, model runs were carried out using scaled precipitation of each of the four TLS* 1480 *campaigns. The modelled snow depth and NSRT data coincided best with validation data when using scaled precipitation from snow depth data based on the TLS data obtained on 19 December 2012. Henceforth, the modelled results analysed and discussed here are only based on this TLS data. The use of an early winter TLS is preferred, since the early winter snow depth distribution best represents winter snowfall events. TLS data obtained in spring* 1485 *already contain ablation processes. Fig. 3 provides justification for the choice of the TLS used. Here the distribution of the ratio modelled to measured snow depth is shown for the 4 TLS available. The TLS data measured on 19 December 2012 is centred around 1, as well as the snow depth curves on the dates of two (7 June 2013, 28 January 2014) of the other three TLS campaigns. Those TLS show satisfactory agreement between modelled and measured snow* 1490 *depths. In contrast, simulations using the TLS on 11 December 2013 overestimate snow depths and have a wider spread compared to the ratio of the other scans. This may be due to snow depth distribution differences due to varying wind conditions in the rock walls. In early winter 2013-2014 a large proportion of snowfall events occurred with southerly winds, whereas in general snowfalls were accompanied by northwesterly flows. A quantitative* 1495 *analysis of the precipitation scaling approach is currently being evaluated (Voegeli et al., submitted). The use of only one TLS is additionally justified by annually recurring snow depth*

*distribution patterns caused by the micro-topography, which have been observed in steep rock walls by Haberkorn et al. (2015a), Sommer et al. (2015) and Wirz et al. (2011).*

1500  *And new Figure 3:*

[Figure]

*Figure 3. Histogram of measured and modelled snow depth data. Solid lines denote the distribution of the ratio modelled over measured snow depth for the 4 TLS available. The TLS of 19 December 2012, 7 June 2013 and 28 January 2014 are centred by 1. The TLS of 19*
1505  *December 2012 was used for precipitation scaling and shows the best agreement between modelled and measured snow depths.*

*New Table 2:*

*Table 2. Snow depth validation (MBE, MAE) between measured and modelled snow depths*
1510  *averaged over the entire N and S facing rock walls at the dates of the independent TLS campaigns. The MBE and MAE are in [m].*

| TLS campaign | Rock wall | MBE | MAE |
|---|---|---|---|
| 7 June 2013 | N | 0.25 | 0.81 |
| | S | 0.52 | 0.74 |
| 11 December 2013 | N | 0.73 | 0.75 |
| | S | 0.47 | 0.48 |
| 28 January 2014 | N | 0.42 | 0.59 |
| | S | 0.17 | 0.31 |

**4. In my opinion the sections 3.3.4 and 4.4 are totally disconnected from other chapters, not in terms of concepts (energy balance is fundamental) but in terms of contents and**
1515  **argumentations. There are no links or references to what observed or discussed in the other sections, there is no think over possible source of modeling uncertainty, is just a**

**chronicle on the course of each component along the seasons. I suggest the authors to remove these chapters, due to the already high number of data and elements to discuss. As it is, the energy balance discussion looks a digression that distracts the reader from the main subject of the paper. Alternatively the section 4.4 must be deeply reworked in order to provide precise evidence of what is discussed in the section 5.1 (Lines 471-484).**

Section 3.3.4 will be merged to section 3.3.1. Section 4.4 is reworked and moved to new Section 4.2. In Section 5 model uncertainties are discussed also from the surface energy balance perspective. Please see also answer 2 and the reworked text.

[revised manuscript text omitted]

**5. Section 4.1.1. The description of the measured snow cover variability by TLS is interesting but useless for the purpose of the paper and has scarce relevance for the scientific community because is too detailed and site-specific. It lacks effort to outline most general patterns of snow accumulation in steep rock walls. It would be very interesting to explore if in your dataset exists a relationship between snow-depth-TLS and steepness of the grid-cell. This analysis might be, I guess for the first time, a real measure of snow-depth in steep rock walls and provide the community some indications on the snow-depth thresholds to use for modeling experiments in steep rock walls. At first, this analysis (i) could exclude the cells above ledges and (ii) could analyze NW and SE faces separately.**

Section 4.1.1 was reworked.

*Section 4.1.1, page 8, new lines 312-323:*
*Similar inter-annual patterns of snow depth distribution were observed using TLS (Figs. 4a-c). However, the variability of the snow depth distribution and thus of snow cover onset and disappearance at certain locations was high over both the N and S facing rock walls. Areas accumulating a thick snow cover can be in the immediate vicinity of snow-free areas due to strongly varying micro-topographic effects. The snow cover was more homogeneous and thicker on the smoother S facing dip slope than on the steeper and rougher N facing scarp slope. Steep to vertical areas far above ledges or areas close to the ridge were usually snow-free, as was the case for the N3 and R2 loggers (Figs. 4a-c). Locations close to the foot of the rock wall and steep areas just above flat ledges accumulated mean snow depths up to 3.5 m. Inter-annual snow depth variations are illustrated in Fig. 5 for both the four locations discussed in detail and for the flat field AWS. Snow depths were on average 1 m lower at both the AWS and NSRT logger locations in 2013-2014 compared to 2012-2013, resulting in snow disappearance up to 4 weeks earlier in 2014.*

The relationship between measured snow depth and slope angle will not be provided, since already enough methods and results are presented. Further it is not within the scope of this study and such an analysis will be presented elsewhere.

**6. Section 4.1.2. The statistics provided (R2 and MBE) are not sufficient. R2 indicates the fraction of variability (variance) in the observation that is explained by the model. Used alone it says little about model performance in strict sense because e.g. in case of temperature you can have an R2=0.99 with 10_C of bias. The modeling efficiency (ME) must be used also. MBE describes the direction of the error bias. Its value is related to the magnitude of values under investigation. A negative MBE occurs when predictions are smaller in value than observations, positive MBE occurs when predictions are greater in value than observations. In case of snow-detph has no sense to provide a mean value of MBE (-0.002 m!!) over the entire model domain because over- and under- estimations vanish each other. Mean absolute error (MAE) or root mean square error (RMSE) must be used instead. Also error bars in Fig.2 look strange, see technical comments. I suggest this paper for further detail: Mayer, D., and D. Butler (1993), Statistical validation, Ecological modelling, 68(1-2), 21–32.**

The subsection 4.1.2 was deleted. Please see also answer 3 and 29 to referee 1.
Regarding the statistics used in this manuscript: first, MBE is important in case of snow since the bias over a whole area has huge implications. Second, the modelling efficiency is approximated by the $r^2$, even if root mean squared error or MAE are more common in some communities. In general, there is no single error analysis that says it all and every one is a little different. The choice of the authors to use $r^2$ and MBE is not a bad one. However, as

requested the MAE will additionally be provided all along the text and all Tables (1-4). For snow depth validation a new Table is added (new Table 2), addressed in answer 3.

**7. Section 4.1.3. If one of the objective of this paper is (accurately) simulate the influence of snow cover on NSRT in steep rock walls I guess that differences in the order of 0.5 – 1m between observed and modeled snow depth is too much for obvious reasons. To reduce this uncertainty, as said in specific comments n.3, the precipitation scaling must be totally revised.**

Please see answer 3, as well as answer 34 to referee 1.

**8. Section 4.2.2. This section would be a validation of NSRT but is very poor under this point of view. The absence of statistical metrics to evaluate model performance is evident here (see general comments). The description of discrepancies between obs. and mod. is only qualitative, comments are limited to temperature without any reference to the modeled snow which is the main constraining factor. In particular, observing together Fig.2a and Fig.3 results that temperature modeling has better perfor mance where snow modeling has worst performance (point S9). Nothing is said about that. This section, that potentially could be the core of the paper, must be strongly improved.**

New Section 4.3.2 and 4.3.3 were rewritten. From our point of view differences between modelled and measured data are quantitatively. Please see answer 6. In addition a link to snow cover conditions in the rock walls has been done in new lines 417-420 (see below).

*Sections 4.3.2 and 4.3.3, page 11, new lines 405-441:*
*4.4.1 NSRT variability at snow-covered locations*
*At locations favouring the accumulation of a thick snowpack the NSRT evolution was strongly controlled by snow for around 7.5 to 9 months of the year in both the N and the S facing rock walls. After the onset of the continuous snow cover in October/November the rock surface was partly decoupled from atmospheric influences. In the N facing slope (N7, Fig. 7b) measured NSRT oscillations were damped, but continuously decreased down to -4 °C, thus clearly showing the occurrence of permafrost at this location, while in the S facing slope (S9, Fig. 7c) measured NSRT remained close to 0 °C. The timing of snow cover onset and disappearance were similar in both the N and the S facing slope. In 2013-2014 the snow cover onset was similar compared to 2012-2013, while the snow disappearance up to 4 weeks earlier (mid-June). The latter caused 0.3 °C (S9) respectively 0.4 °C (N7) warmer MANSRTs in 2013-2014 than in the previous year due to snow-free conditions during the weeks with most intense solar radiation (mid-June to mid-July).*
*At the locations accumulating a thick snow cover the temporal evolution of modelled NSRTs are in good accordance with the measured ones in both the shaded (N7) and the sun-exposed (S9) slopes with $r^2$ = 0.80-0.94 (Figs. 8a, d). When comparing measured and modelled NSRT evolution, the modelled timing of the snow cover onset, of the zero curtain period and of*

*snow disappearance was similar (Figs. 7b, c and dT in these). This underlines that satisfactory modelled snow cover duration is the most important factor influencing modelled NSRT evolution, rather than accurately modelled absolute snow depths. Variations between measured and modelled NSRT are small at the S facing S9 with a MAE of 0.6/0.6 °C and a MBE of -0.3/-0.4 °C, indicating too low modelled NSRTs in summer. At N7 measured and modelled NSRT fit well together during the snow-free period, while measured NSRTs are colder than modelled ones during the snow-covered period resulting in a MBE of 0.8/0.8 °C and a MAE of 1.1/1.0 °C (Table 1).*

*4.4.2 Thermal effect of snow*
*The previously discussed modulating influence of the snow cover on the surface energy balance and its effects on the ground thermal regime can be emphasized by comparing NSRTs at the snow-covered N7 and S9 to the modelled snow-free scenario at these locations (blue lines in Figs. 7b, c). Using the snow-free scenario, modelled NSRT oscillations of N7 and S9 were pronounced during the whole study period, indicating a permanent energy exchange between the atmosphere and the rock. MANSRTs were -2.8/-1.9 °C at the shaded N7 and 0.4/1.3 °C at the sun-exposed S9. This contradicts the NSRT measurements at these locations (Section 4.3.2). Measurements reveal a permanent insulation of the rock by a continuous snowpack between October/November and June/July. Neither cold atmospheric conditions in winter, nor strong insolation and warm air temperatures between May and July (all energy available used for snow melt, Figs. 6a, c) affected the rock thermal regime below the snowpack. Thus the potential thermal effect of a thick, long lasting snowpack accumulating in steep rock can locally be quantified: at locations accumulating a long lasting, insulating snow cover the measured MANSRTs were 2.9/2.4 °C higher in the shaded and 2.0/1.4 °C higher in the sun-exposed rock wall, while comparing to modelled MANSRT of the snow-free scenario (Table 1). The negligence of snow in steep rock resulted in deviations between measured and modelled (snow-free) NSRT causing the $r^2$ to decrease by 0.26/0.21 at N7 and 0.57/0.51 at S9 (Figs. 8b, e) and the MAE to increase by 4.1/2.7 °C at N7 and 4.7/3.4 °C at S9.*

It is correct that the ground thermal regime depends on snow conditions, but mainly on snow cover duration, not on absolute snow depths. Please see answer 34 to referee 1, as well as answer 3. Not only snow cover duration, but also ground conditions are important for near-surface rock temperature modelling. In the S facing slope NSRT can be simulated well since permafrost is absent in the S and most NSRT are around 0 °C below a thick snowpack. In addition the S rock surface is more homogenous (dip slope) compared to the N face (scarp slope). Thus the interaction between adjacent rock portions sticking out of the snow and rock portions covered by thick snow is reduced on the S face.

**9. Section 4.2.3. The idea of a run with forced snow-free condition is good but results are not exploited at all. This run could be used as reference to quantify the potential thermal effect of snow cover at different slope and aspect (see Pogliotti, 2011). This is a way to**

**generalize the results and valorize the dry run. Of course, the precipitation scaling must be**
1725  **improved before (specific comments 2).**

A comparison between simulations of snow-covered and snow-free scenarios was done in order to quantify errors made while neglecting snow in steep rock wall thermal modelling. The objective to run Alpine3D also with forced snow-free conditions might not have been
1730  clear. This was clarified in the text with an additional chapter in the methods > new chapter chapter 3.2. Please see therefore answer 38 to referee 1, the entire revised results section and new Fig. 10, as well as additional comments in the text:

*Section 1, page 3, new lines 103-110:*
1735  *High-resolution (0.2 m) simulations were carried out, either providing snow cover distribution to the model (by precipitation scaling) or fully neglecting the presence of a snow cover in the rock walls. Thus the potential error induced by neglecting the snow cover in steep rock face thermal modelling for slope angles >50° can be estimated. This is necessary, since it has in general been assumed that wind and gravitational transport remove the snow from steep*
1740  *rock in slopes >50–60° (e.g. Blöschl and Kirnbauer, 1992; Gruber Schmid and Sardemann, 2003; Winstral et al., 2002) and rock temperatures were often modelled without snow for idealized rock walls >50° (e.g. Gruber et al., 2004a; Noetzli and Gruber, 2009; Noetzli et al. 2007).*

1745  **10. Line 29: the term "rock avalanche" refers to big falls of earth material (of up to millions of metric tons) able to reach velocities of more than 50 meters per second and leave a long trail of destruction. In the Alps such phenomena are not "numerous" (e.g. Val Pola 1987, Tschierva 1988, Brenva 1997, Thurwieser 2005) and even less those where permafrost can be directly listed among the trigger factors. The right term is "rock falls".**
1750
Changed to 'rock fall'.

**11. Line 30: strange references, Gruber & Haeberli 2007 is better and more comprehensive than Gruber 2004b, e.g. Fisher 2012 (Nat. Hazards Earth Syst. Sci) is missing.**
1755
References were reworked, we decided to refer to the following:

*Section 1, page 2, new line 34:*
*(e.g. Fischer et al., 2012; Gruber et al., 2004b; Phillips et al., 2016b; Ravanel et al., 2010,*
1760  *2013)*

**12. Line 31: Davies et. al 2001 is wrong! Gruber et. Al 2004a is wrong! Fisher 2012 (Nat. Hazards Earth Syst. Sci) is more appropriate than Fisher 2006, Gruber & Haeberli 2007 is missing, Allen & Huggel 2013 (Glob. and Planetary Change) is missing, Saas 2012 (Nat.**

1765 **Hazards Earth Syst. Sci) is missing, Deline et al. 2015 (Snow and Ice- Related Hazards, Risks, and Disasters, chapter 15) is missing: : : and many more.**

Davies et al. (2001) and Gruber et al. (2004a) were deleted. Other references were provided, also with respect to your suggestions.

1770

*Section 1, page 2, new lines 34-38:*
*Rock fall can be attributed to various triggering factors (Fischer et al., 2012; Krautblatter et al., 2013), including a fast reaction of rock faces to climate change expressed in rapid active layer thickening and permafrost degradation (e.g. Allen and Huggel, 2013; Deline et al.,*
1775 *2015; Gruber and Haeberli, 2007; Ravanel and Deline, 2011; Sass and Oberlechner, 2012).*

**13. Line 35: Gruber 2012 is wrong! e.g. Guglielmin 2003 (Geomorphology) is missing**

Gruber (2012) was deleted, additionally the paragraph has been reworked. Please see
1780 answer 5 to referee 1.

**14. Line 36: if you cite only Fiddes et al. 2015 add "e.g." because exist more**

The paragraph has been reworked and Fiddes et al. 2015 was deleted.
1785

**15. - - - Line 37: kilometres**

Please see answer 14.

1790 **16. Line 41: transient changes… Harris et al. 2009 alone has no sense because is a big state-of-the-art of mostly all fields of research around mountain permafrost... Noetzli & Gruber 2009??**

Harris et al. (2009) was removed. Noetzli et al. (2007) and Noetzli and Gruber (2009) will be
1795 moved at the end of the sentence. Please see answer 12 to referee 1.

**17. Line 46-49: ...cannot capture… the ground thermal regime. I'm not sure of that. The Fiddes 2015 approach has not been yet validated against field measures.**

1800 Please see answer 13 to referee 1.
The model results of Fiddes et al. (2015) were validated in the same publication against a network of air temperature, ground surface temperature and snow depth measurements, as well as data loggers (PERMOS) to evaluate ground surface temperature in coarse debris and bedrock.
1805

**18. Line 56: remove "However"**

Deleted.

1810 **19. Line 56-58: this statement is too strong and do not consider that the temperature of a point in depth integrates the contribution of a certain area at surface. This area is wider as deeper is the point so the effect you are talking about is probably limited to few meters. Thus, in my opinion, to investigate the 3D subsurface heat flow is not necessary to reproduce surface temperatures with so-high spatial resolution. Please, reformulate this**
1815 **sentence considering also these aspect.**

The sentence was reworked.

*Section 1, page 2, new lines 49-52:*
1820 *Beside three-dimensional (3d) subsurface heat flow and transient changes in steep bedrock thermal modelling (Noetzli et al., 2007; Noetzli and Gruber, 2009), the strongly variable spatial and temporal rock surface boundary conditions therefore also need to be taken into account. The spatially variable snow cover is one of these driving factors.*

1825 **20. Line 59-60: Gruber 2004 is wrong!, Gruber & Haberli 2007 is a kind of review and snow control only is mentioned, remove it. Pogliotti 2011 is probably the first work that systematically investigate the thermal effect of snow cover (moreover with high affinity with the present work) even in steep rock walls and is missing. Magnin 2015, Haberkorn 2015a & 2015b are missing too!**

1830

Please see answer 14 to referee 1.

**21. Line 63: Pogliotti 2011 is wrong!**

1835 Pogliotti (2011) was removed.

**22. Line 65: Gruber et al. 2004A is wrong!**

Please see answer 16 to referee 1.
1840
**23. Lines 82-85: this sentence is not clear, explain better.**

The whole sentence was deleted.

1845 **24. Line 106: elevation range must be explicit in the site description.**

Are given.

*Section 2, page 3, new lines 120-121:*

1850 *The 40 m high slopes (2890–2930 m a.s.l.) are 40° to 70° steep, with vertical to overhanging (>90°) sections (Fig. 1a).*

**25. Line 127: Remove However. In this study, only data from…**

1855 'However' was deleted and sentence rewritten accordingly. (Answer 22 to referee 1).

**26. Lines 130-136: what you describe here is not evident neither from figure 1 nor from table 1 but just in figure 3. If you don't show a plot you have to describe better the differences you observe in the temperature fluctuations in order to justify your choices.**

1860 Please see answer 23 to referee 1.

**27. Lines 191-194: the initialization is important. Provide here, synthetically, more details about initialization without reference to another paper. Is not clear as it is.**

1865 The sentences were rewritten. However, all information regarding the initialization is given.

*Section 3.1.2, page 5, new lines 179-185:*
*Based on the DEM, the land-use classification was divided into two groups with varying rock*
1870 *properties, depending on whether the grid cells were N or S facing. The rock is simulated to 20 m depth, divided into 24 layers of varying thickness ranging from 0.02 m at the surface to 4 m at the bottom of the substrate. For each classification, the rock layers were initialized with different layer temperatures based on borehole rock temperatures measured on-site (Fig. 1c, PERMOS 2013). The rock was assumed to be 99 % solid with 1 % pore space*
1875 *containing ice (N facing grid cells) or water (S facing grid cells) to account for near-surface fracture space to a depth of 0.5 m. Unfractured rock with a solid content of 100 % was assumed between 0.5 and 20 m depth.*

**28. Line 205: remove high resolution**

1880 Removed.

**29. Line 211: Uncertainties in modeling...**

1885 Changed, but ,oved to a new section for a better understanding.

*Section 3.3, page 7, new lines 259-262:*
*3. 3 Model validation*

*Uncertainties in modelling the snow depth distribution and the near-surface rock thermal regime in steep rock walls were assessed statistically, using the mean bias error (MBE), the mean absolute error (MAE) and the coefficient of determination ($r^2$).*

**30. Line 213: R2 is the coefficient of determination! MBE is not the right statistic in this case, look at specific comments.**

Changed.
Please see also answer 6.

**31. Lines 209-213: move this paragraph as preamble of chapter 4.**

The methods and results section were completely rewritten. These sentences were deleted, but are addressed in new Sections 3.2 (please see answer 38 to referee 1) and 3.3. Please see answer 3 to referee 1 for the introduction of Section 4.

**32. - Lines 216-218: remove.**

Deleted.

**33. Lines 222-224: what is the "snow depth driving mode" of snowpack? Something that convert snowfall in liquid precipitation? By which snow density value? This is a key step of your precipitation scaling, please explicit all the detail, synthetically, without references to other papers.**

The 'snow depth driving mode' means that SNOWPACK was driven with measured snow depth as model input (not precipitation). SNOWPACK converts fresh snow falls in precipitation under consideration of snow settlement, as well as fresh snow density which are both calculated based on a statistical model.
Although this is not a key step in our precipitation scaling, but rather a common approach to calculate liquid precipitation if only snow depth is available, we will provide additional explanation on this topic. Detailed information, however, is given in Lehning et al. (1999) and Wever et al. (2015). We think providing these references in the manuscript are sufficient. As you mentioned, SNOWPACK and Alpine3D are well known and documented models.

*Section 3.1.3, page 6, new lines 217-221:*
*By using the snow depth driven mode of the SNOWPACK model, the snow depth measurements were used to determine the timing and amount of snowfall by interpreting increases in snow depth as fresh snowfall. According to Lehning et al. (1999) and Wever et al. (2015), SNOWPACK converts snowfall to precipitation while calculating both snow settlement and snow density based on a statistical model.*

**34. lines 225: "integrated" seems a mathematical term, please use a synonym.**

Changed to 'was used'.

**35. Line 228: replace "onto the DEM" with "in each grid cell".**

Changed to:

*Section 3.1.3, page 6, new lines 225-226:*
*These scaling factors were then used to scale the two-year precipitation time series for each grid cell of the DEM.*

**36. Lines 228-232: replace this sentence with "cells where TLS data were non available have been excluded from the analysis".**

Changed to:

*Section 3.1.3, page 6, new lines 228-230:*
*Data gaps in the TLS lead to data gaps in the precipitation scaling grid, resulting in erroneously modelled snow depths and rock temperatures at these locations. For the analysis of the Alpine3D grid output those grid cells have not been used.*

**37. Line 233: TLS campaign.**

Changed to:

*Section 3.1.3, page 6, new line 231:*
*Thirdly, model runs were carried out using scaled precipitation of each of the four TLS campaigns.*

**38. Lines 233-241: explain better why you choose only the TLS of December 2013 for driving the precipitation scaling and provide quantitative proofs for this choice (model performance on modeled vs. observed NSRT). Look also specific comments.**

Please see answer 3.

**39. Line 247: see specific comments 4.**

Please see answer 4.

**40. Line 262: see specific comments 5.**

Please see answer 5.

**41. Line 277: see specific comments 6.**

Please see answer 6.

**42. Line 279: MBE = -0.002 m has no sense. MBE is the wrong statistic in this case (see specific comments).**

Please see answer 6.

**43. Lines 282-283: explain the method used for calculating the error bars and exactly what they represent. Is not clear. How can I have an error bar of _0.3m and a difference obs./mod. (red dot, red line) of about 1 m?**

The error bars in the old Fig. 2a represent the errors only of the validation data itself. An error bar of ±0.3m is composed of both an error of ±0.08 m due to errors of the TLS method itself and an error of ±0.22 m inherited in the precipitation input data due to precipitation scaling.
The highest inaccuracies of validation data occurred in areas with a strongly heterogeneous surface (N face).
The error bars do not indicate differences between measured and modelled snow depth. The error bars in new Fig. 5 were omitted. New Fig 5 is presented in answer 32 to referee 1.

**44. Lines 300-301: explain/explore better the reasons of such a huge difference in S9.**

Differences up to 1 m between measured and modelled snow depths in the S facing slope are mainly due to inadequate description of snow settlement. This is explained in the discussion section 5.1 (lines 451-456).
Lines 299-301 will be removed, since the results will be presented without any assessment or interpretation of the data. Possible explanations for model uncertainties are presented in the discussion.

*Section 5.1, page 14, new lines 535-540:*
*In this study discrepancies in modelling absolute snow depths in steep rock walls are evident (Figs. 4, 5). This is a consequence of the linear precipitation scaling algorithm used here. Snow settlement is calculated for snow depths at the AWS location and is then linearly scaled into the rock walls, but snow depths and the meteorological forcing obviously differ between the flat field AWS and the rock walls. This causes the snowpack to settle differently and in a non-linear manner. Differences in settling calculated at the AWS and for the grid points in the Alpine3D model domain therefore cause absolute snow depth errors.*

**45. Line 287: see specific comments 7.**

Please see answer 3, as well as answer 34 to referee 1.

**46. Line 334: what does it means "auspicious accordance"? please try to be more adjective**

Deleted.

**47. Line 335: MBE is the wrong statistic in this case (see specific comments).**

Please see answer 6.

**48. Line 330: see specific comments 8.**

Please see answer 8.

**49. Line 346: see specific comments 9.**

Please see answer 9.

**50. Lines 363-364: this sentence is ambiguous, what does it means "not pronounced as expected"? Expected for N/S differences (?) this is not the real case. Expected for snow-free, steep, conditions(?) this is not the real case. If you average all the measures of a mountain side like the yours, the value you got is exactly what I expected.**

MANSRT differences between the NW and SE faces are smoothed due to thick snow. MANSRT differences between both faces would have been bigger if the slopes would have been snow-free, as it is often assumed in literature for steep rock faces. The text was clarified, please see therefore answer 37 to referee 1.

**51. Line 366: remove "compensating"**

Removed.

**52. Line 367: remove "In 2013-2014"**

Removed.

**53. Lines 367-370: respect the colon, merge**

Text changed to:

*Section 4.4.1, page 12, new lines 464-470:*

*For all cases, the MANSRT variability within the individual N and S slopes was higher in 2012-2013, which is the result of two effects. In 2012-2013 the mean annual air temperature was 0.8 °C lower than in 2013-2014, causing MANSRTs at snow-free locations to decrease by around 0.6 °C. In contrast, MANSRTs at snow-covered locations in the N slope increased by up to 0.4 °C due to an early onset of a long lasting, insulating snow cover. In early winter 2013-2014 the absence of a sufficiently thick, insulating snow cover resulted in effective ground heat loss at these locations (Haberkorn et al., 2015a).*

**54. Lines 374-376: the higher SD of modeled temperatures derives essentially by the scarce ability in reproduce real (in terms of thickness) snow cover conditions on both sides.**

Please see answer 3, as well as answer 34 to referee 1.

**55. Line 378: how can you say that underestimation is mainly in summer? (fig. 3?). Explicit.**

Deleted.

**56. Lines 379-380: remove "therefore", this sentence is not a direct consequence of what you said before, or only partially. This is a comparison with the 3.6_C stated at line 363. Contextualize better this sentence.**

The whole section was reworked.

*Section 4.4.1, page 12, new lines 450-458:*

*The topography driven difference of the measured mean MANSRT between the entire N and the entire S facing rock wall were 3.6/3.2 °C. Such a small deviation is reasonable when taking into account that the rock walls are facing rather NW and SE than N and S (Fig. 1a, Appendix Table 1A), as well as considering the accumulation of a thick snow cover at 7 of 11 locations in both the N and S slopes.*

*At the corresponding 22 grid cells, the modelled mean MANSRT difference for the snow-covered scenario across the entire N and S facing slope is 2.6/2.3 °C and thus around 1.0 °C lower than the measured values (Table 3). This is mainly caused by too low modelled NSRTs and thus MANSRTs, especially in the sun-exposed rock wall during snow-free periods (Fig. 9) and at locations without snow (N and S slopes) resulting in a MBE of -1.3/-1.0 °C.*

**57. Line 384: compared to what? Modeled or real snow covered conditions? It is very difficult to follow your reasoning looking at Table 3 because the number in the text are often means of values in different columns of the table and moreover rounded! If you need these numbers add columns in the table!**

Modelled MANSRT of snow-free simulations were around 2 ° C colder to both measured MANSRT and modelled MANSRT assuming snow-covered conditions. If we give rounded values, we will state this. Differences can be calculated from Table 3. However, the whole section > new Section 4.4.2 was completely reworked.

*Section 4.4.2, page 12, new lines 472-477:*
*4.4.3 Snow-free scenario*
*In the absence of a snow cover, the modelled MANSRT variability was much lower within the individual rock walls (Fig. 9). Assuming the modelled snow-free scenario in the entire rock walls, resulted in mean MANSRT of -3.3/-2.3 °C within the N and of 0.1/0.8 °C within the S facing slopes (Table 3). In correspondence to the single NSRT locations (Section 4.3.3) the mean MANSRT of snow-free simulations confirmed too low modelled MANSRT when compared with both observations and snow-covered simulations (Fig. 9).*

**58. Lines 383-390: rework this section in accordance with the previous comment. Consider also the specific comments n.9**

Please see answer 57 above. The difference in line 88 is calculated for modelled snow-free conditions between the N and the S facing slopes. Please see answer 9.

**59. Lines 392-399: very poor description. Provide more details or remove this section, figure 6 and the "grid" lines in table 3.**

Please see answer 38 to referee 1. In addition new Fig. 10 has be revised and the information on MANSRT are now given in new Fig. 10 (please see this figure in answer 3 to referee 1). Thus the 'grid lines' in new Table 3 were deleted .

*New Table 3:*
*Table 3. MANSRT, MAE and MBE [all in °C] calculated within the individual N and S facing rock walls at NSRT locations. The MAE and MBE were calculated between measurements (O) and model predictions of both the snow-covered (PS) and the snow-free scenarios (PSF) at NSRT locations. Additionally mean annual air temperature (MAAT) for the years 2012-2013 and 2013-2014 is shown.*

| Scenario | Rock wall | 2012- 2013 | | | 2013- 2014 | | |
|----------|-----------|--------|-----|-----|--------|-----|-----|
| | | MANSRT | MAE | MBE | MANSRT | MAE | MBE |
| O | N | -0.7 | | | -0.5 | | |
| PS | N | -1.0 | 1.6 | -0.4 | -0.6 | 1.7 | -0.2 |
| PSF | N | -3.3 | 3.9 | -2.6 | -2.3 | 2.7 | -1.8 |
| O | S | 2.9 | | | 2.7 | - | |
| PS | S | 1.6 | 1.6 | -1.3 | 1.7 | 1.7 | -1.0 |
| PSF | S | 0.1 | 4.7 | -2.8 | 0.8 | 3.4 | -1.9 |

| | | |
|---|---|---|
| MAAT | -3.2 | -2.4 |

**60. Line 401: see specific comments 4.**

Please see answer 4.

**61. Line 447: modeling of water flow within fractures is not relevant for reproducing surface rock temperatures. Also the influence of surface water flow is negligible in comparison to a correct simulation of snow cover thickness.**

'Water flow in fractures' was removed. Please see also answer 41 to referee 1.

**62. Line 451: check the references (see specific comments 1)**

The sentence was deleted.

**63. Lines 452: please explicit the value of snow density used (see also technical comment Lines 222-224)**

SNOPWACK calculates fresh snow density for each time step by a statistical model.  Please see answer 33 and 44.

**64. Line 453: remove "However"**

Removed.

**65. Lines 454-455: the first half of the sentence (from However to AWS) is obvious thus can be removed, the second half is not clear, explain better this concept of non-linear settling. Include also the sentence after.**

Please see answer 44.

**66. Lines 457-458: this is not evident from your data. Look the table attached (Fig.1) and justify your sentence.**

New Fig. 7b, c will be cited. In new Fig. 7b, c it is shown that modelled and measured NSRT are in good agreement, although absolute snow depths vary by around 1 m. Please see also answer 34 to referee 1. In addition, snow cover duration for the loggers shown in new Fig. 7b, c is given in new Table 1, which will be also referred to.

**67. Line 459-461: is not evident to me. Check the references (see specific comments 1)**

Please see answer 44 to referee 1.

**68. Line 462: what is the "apparent insulation"?**

2175

'Apparent' was removed.

**69. Lines 465-466: heat flux at the bottom (20m below) cannot be seen in surface in so short simulations!**

2180

Deleted.

**70. Lines 468: remove "While"**

2185    Not deleted.

**71. Lines 471-484: this is interesting but is very difficult to see the evidence of what you are saying in the plot 7 as well as find references in the text of section 4.4. See specific comment 4.**

2190

Please see also answer 4. The text is changed to:

*Section 5.1, page 15, new lines 574-583:*
*Effective ground heat loss in autumn 2013-2014 was observed and modelled at exposed*
2195    *locations due to an initially thin snow cover, but a heat exchange between adjacent locations*
*covered with thick snow was not reproducible by the model, although it was measured*
*(Haberkorn et al., 2015a). In contrast modelled and measured NSRTs in the homogenous S*
*facing slope supported the validity of the 1d heat conduction assumption at snow-covered*
*locations since here a continuous, smooth snowpack was an effective barrier to heat loss*
2200    *from the ground to the air (Fig. 7c). Finally, difficulties in partitioning the measured incoming*
*shortwave radiation in a direct and diffuse component, particularly for low sun angles, may*
*explain the stronger modelled net radiation for snow-free conditions in the shaded (Fig. 6b)*
*than in the sun-exposed slope (Fig. 6d), which is amplified by differences in slope and aspect*
*between the model domain and reality (Appendix Table 1A).*

2205

**72. Lines 485-486: move this in the results providing evidence of the source data. Keep in mind specific comments about the use of MBE.**

Please see answer 45 to referee 1.

2210

**73. Lines 489-499: in my opinion this belong to section 5.1. Check the references (see specific comments 1) all along this paragraph.**

This paragraph will not be moved to section 5.1, since model uncertainties are not discussed in this paragraph.

**74. Line 500: replace "possibly made" with "introduced"**

Changed.

**75. Lines 504-505: looking at table 3 the warming effect on MANSRT is up to 3.7_C at N7 (2012-2013) and up to 1.5_C at S9 (2012-2013). Please keep attention and precision in reference to plot and table contents!**

Lines 504-505 refer to the entire rock wall (new Fig. 10) not to single locations (new Table 1). The text is clariefied.

*Section 5.2, page 15, new lines 601-604:*
*The modelled MANSRT increase of 1.3 – 2.5 °C found for both snow-covered N and S facing steep rock walls compared to snow-free simulations (Figs. 10e, f) is in the same order of magnitude than the cooling or warming effect of snow on mean-annual ground surface temperatures modelled by Pogliotti (2011).*

**76. Lines 508-511: this obviously depends on the amount of snow. A persistent thin snow cover has always cooling effect both at N and S faces, while a thick snow cover has warming effect. Thus the reason you observe on average a warming effect of snow cover is because you allow the accumulation of thick snow. If you have a look a other cells with thin snow I'm sure you can observe cooling effect between dry and snow simulation. So change this sentence keeping in mind also these aspect.**

The influence of snow on mean annual rock temperatures close to the surface of course depends on snow depth and especially on snow cover duration. In this study snow accumulates for around 7-5-9 months a year and has a warming effect on bot NW and SE faces. The effect of thin snow on rock surface temperatures, especially on mean annual temperatures is still poorly studied. Whether thin snow has a cooling or warming effect on mean annual rock temperatures on both N and S faces strongly depends on snow cover duration. Thin snow < 0.2 m will not persist on S faces for several months, especially not during the months with most intense radiation and its effect on mean annual rock temperatures is still not clear and should be better investigated in future.
The contradiction of the presented results to previous studies (e.g. Hasler et al. 2011; Magnin et al. 2015) will be discussed more differentiated and the sentence will be reworked, please see therefore also answer 3 to referee 1.

*Section 5.2, pages 15-16, new lines 608-620:*

*In contrast, in sunny rock walls both measurements and model results at the point- and spatial scale (Tables 1,3) challenge the hypotheses presented by Magnin et al. (2015) and Hasler et al. (2011), who supposed a cooling effect of a snow cover due to the shielding of the rock surface from radiation influences during the months with most intense insolation. Discrepancies with our observations may have three reasons: (i) These authors estimated snow depths qualitatively rather than quantitatively. (ii) They adopt the widespread theory of an insulating snow cover with depths exceeding 0.6 m for blocky terrain (Hanson and Hoelzle, 2004, Keller and Gubler, 1993, Luetschg et al., 2008), while Haberkorn et al. (2015a) found the insulation effect on NSRT at smooth rock surfaces already present for snow depths exceeding 0.2 m. (iii) Their observations are a few point measurements, whereas we complemented multiple point measurements with simulations of the entire rock walls. At Gemsstock a thick snow cover accumulates in most parts of the rock walls between October and June/July. Considering snow in sunny, steep rock for shorter periods or only for the months with strongest insolation (March to June) most likely has a cooling effect on rock surface temperatures.*

**77. Lines 515-520: this sentence is very interesting but not well introduced nor supported by findings of this paper. Provide more detail, evidence and argumentations in order to support this suggestion.**

Please see answer 47 to referee 1.

**78. Line 524: this section is very interesting and useful for the modeling community, but is poor of numerical evidences. Please, provide a synthetic table (or plot) where the influence of grid resolution on the model performance becomes evident (see also specific comments for assessing model performance in the correct way).**

Please see answer 51 to referee 1.

**79. Lines 551-553: I would say, "the results of the present work help to quantify the potential error..."**

The sentence was moved to the end of the conclusions and has been reworked.

*Section 6, page 17, new lines 678-680:*
The results of this study help to quantify the potential errors in ground temperature modelling when neglecting the evolution of a snow cover in steep rock exceeding 50°, as has often been done for idealized rock walls.

**80. Line 554-556: "Alpine3D simulates near-surface rock temperatures and snow depth in the heterogeneous terrain accurately." in general this is true but is not the case of this work. The reason is that the precipitation scaling procedure is weak and provide unreliable**

**precipitation input to the model. In my opinion this conclusion does not reflect the real result of this work.**

Sentence was reworked, please see therefore answer 50 to referee 1.

**81. Line 556-558: lateral heat-flux is negligible in comparison to the effect of a bad precipitation input.**

Please see answer 3, as well as answer 34 to referee 1. The paragraph was slightly reworked.

*Section 6, page 17, new lines 663-665:*
*Remaining errors in snow depth and consequently rock temperature simulations are explained by inadequate snow settlement modelling, due to linear precipitation scaling, missing lateral heat fluxes in the rock and by errors due to shortwave radiation, air temperature and wind interpolation, which are complex in such terrain.*

**82. Line 559-561: this is true, the potential of the dataset is very high but the choice of exploring just 2 cells on the N face and 2 cells in the S face strongly constrain this potential. See also general comments.**

Please see answer 30 to referee 1. The sentence was deleted.

**83. Line 562: this sentence on the lateral heat flux is speculative. Nothing in the results provides the basis to verify this statement.**

The sentence was deleted.

**84. Lines 569-571: also in that case no numerical evidence about model performance are provided in the results hence this sentence is speculative too.**

Please see answer 51 to referee 1.

**85. Table 3: Caption (Line 812), replace "data" with "cells". How do you identified snow-free cells?**

The sentence in lines 811-812 refers to the model run considering snow (in old Table 3: modelled N grid snow & modelled S grid snow) and to the model run lacking snow (in old Table 3: modelled N grid snow-free & modelled S grid snow-free), in the latter the precipitation input was forced to be zero. Modelled results given in the respective lines of Table 3 were averaged over the entire N and S facing model domain. Thus a comparison between the run considering snow and the run without snow has been done. However, the 'grid' lines were deleted in new Table 3, please see therefore answer 59.

**86. Figure 1: The boreholes are not considered at all in this work then I suggest to remove it from the figure and caption to avoid confusion.**

The 22/30 shallow NSRT logger locations were used to validate model results. Please see answer 30 to referee 1. The horizontal borehole (points BH N and BH S in old Fig 1a, e, f), which was drilled through the whole ridge, provided rock temperature data in various depths, which were used to initialize our model (new Section 3.1.2). We removed the points BH N and BH S in the new Fig. 1a, but we will keep this information in new Fig. 1c. Please see answer 52 to referee 1 for the new Fig. 1.

**87. I suggest to replace the three colorful elevation plot by a "classic" but more readable cross-section along the logger line which easily can gives the information about elevation and steepness at one-shot.**

Fig. 1 was reworked. Please see also answer 52 to referee 1 for the new Fig. 1.

**88. Figure 2: Just figures a) and f) are relevant for the interpretation and discussion of the precipitation scaling. Remove figures b) c) d) e) that are not relevant and enlarge figure a).**

Fig. 2 > now new Fig. 4 was reworked. Please see answer 29 to referee 1 for the new Fig. 4.

**89. The range in figure f) has been constrained at _0.5m for graphical reasons, but a frequency distribution plot (barplot) of differences on the model domain should be inserted as compendium to provide a comprehensive overview of modeled snow depth uncertainties.**

Fig. 2 > now new Fig. 4 was reworked. Please see answer 29 to referee 1 for the new Fig. 4. A scatter plot was provided to show differences between measured and modelled snow depth.

**90. Figure 3: Caption: dT are present also in the plots d) and e) not only in b) and c) as stated.**

The caption was ambiguous. We meant that dT was calculated in new Fig. 7b, c between measured and modelled snow-covered conditions, although snow-free conditions were also shown. In new Fig. 7d, f dT is calculated between measured and modelled snow-free conditions. The figure caption was reworked.

*Caption new Fig. 7:*
*Figure 7. (a) Daily mean air temperature at the AWS Gemsstock. (b-e) Measured and modelled daily mean NSRT are shown for four selected locations in the N and the S facing rock walls representing typical snow conditions (snow, no snow). At locations accumulating snow (N7, S9) modelled NSRTs are shown for both the snow-covered and the snow-free*

*scenarios, while the NSRT differences (dT) were only shown between measured and modelled snow-covered conditions. At locations without snow (N3, R2) measured and modelled NSRT differences (dT) are also shown.*

**91. Figure 5: The boxplot shows the meadian but in the text and table 3 the references are always to the mean. Please modify the boxplot in accordance with the text.**

In the boxplots the means were also provided.

*New Figure 9:*

[Figure]

*Figure 9. MANSRT variability within the individual N (left) and S (right) facing rock walls for the years 2012-2013 (12-13) and 2013-2014 (13-14). The MANSRT variability in the rock walls were based on 22 measured NSRTs, 11 facing N and 11 facing S. Measured MANSRT variabilities are compared to modelled MANSRT differences calculated at the grid cells of NSRT locations, shown for both the snow-covered and the snow-free scenarios. In addition to the MANSRT differences calculated at all 22 NSRT locations, the modelled MANSRT variability of each grid cell of the entire model domain is shown, depending on whether the grid cell is N or S facing. The median is marked with a red horizontal line in each box, the mean is additionally plotted as a red asterix, the box edges are the 25th and the 75th percentiles, the*

*whiskers extend to the 2.5 % and 97.5 % quantiles and outliers are plotted as individual crosses.*

**Additional figure used in the response to referees**

[Figure]

2405    Fig. 1 for revision: Histogram for TLS data: solid lines  illustrate the distribution of the ratio modelled/measured snow depth for the 4 TLS available. The TLS of 11 December 2013 (20131211, pink line) is centred by 1 (since this TLS was used for precipitation scaling). Dashed lines show a comparison between each TLS. First each pixel is corrected with the mean value of the TLS. Thus the relative snow depth per scan is provided. Than the ratios of

2410    the relative snow depths of each TLS are compared to each other.

**Abbreviations used in the response to referees**

AWS: automatic weather station

DEM: digital elevation model

2415    ILWR: incoming longwave radiation

IMIS: Intercantonal Measurement and Information System

ISWR: incoming shortwave radiation

MAE: mean absolute error

 MANSRT: mean-annual near-surface rock temperature

2420    MBE: mean bias error

NSRT: near-surface rock temperature

NW: north-west

$r^2$: coefficient of determination

SE: south-east

2425    TLS: terrestrial laser scanning

**References used in the response to referees**

Davies, M.C.R., Hamza, O., and Harris, C.: The Effect of Rise in Mean Annual Temperature on the Stability of Rock Slopes Containing Ice-Filled Discontinuities, Permafr. Periglac. Process., 12, 137-144, doi:10.1002/ppp378, 2001.

Dilley, A.C., and O'Brien, D.M.: Estimating downward clear sky long-wave irradiance at the surface from screen temperature and precipitable water. Q. J. Roy. Meteor. Soc., 124, 1391 – 1401, doi: 10.1002/qj.49712454903, 1998.

Fiddes, J., Endrizzi, S., and Gruber, S.: Large-area land surface simulations in heterogeneous terrain driven by global data sets: application to mountain permafrost, Cryosphere, 9, 411-426, doi:10.5194/tc-9-411-2015, 2015.

Flerchinger, G.N., Xaio, W., Marks, D., Sauer, T.J., and Yu, Q.: Comparison of algorithms for incoming atmospheric long-wave radiation. Water Resour. Res., 45, W03423, doi: 10.1029/2008WR007394, 2009.

Gruber, S.: Derivation and analysis of a high-resolution estimate of global permafrost zonation, Cryosphere, 6, 221-233, doi:10.5194/tc-6-221-2012, 2012.

Gruber, S., Hoelzle, M., and Haeberli, W.: Rock-wall Temperatures in the Alps: Modelling their Topographic Distribution and Regional Differences, Permafr. Periglac. Process., 15, 299-307, doi:10.1002/ppp.501, 2004a.

Haberkorn, A., Hoelzle, M., Phillips, M., and Kenner, R.: Snow as driving factor of rock surface temperatures in steep rough rock walls, Cold Reg. Sci. Technol., 118, 64-75, doi:10.1016/j.coldregions.2015.06.013, 2015a.

Haberkorn, A., Phillips, M., Kenner, R., Rhyner, H., Bavay, M., Galos, S.P., and Hoelzle, M.: Thermal Regime of Rock and its Relation to Snow Cover in Steep Alpine Rock Walls: Gemsstock, Central Swiss Alps, Geogr. Ann.: Ser. A, 97, 579-597, doi:10.1111/geoa.12101, 2015b.

Harris, C., Arenson, L.U., Christiansen, H.H., Etzelmüller, B., Frauenfelder, R., Gruber, S., Haeberli, W., Hauck, C., Hölzle, M., Humlum, O., Isaksen, K., Kääb, A., Kern-Lütschg, M.A., Lehning, M., Matsuoka, M., Murton, J.B., Nötzli, J., Phillips, M., Ross, N., Seppälä, M., Springman, S.M., and Vonder Mühll, D.: Permafrost and climate in Europe: Monitoring and modelling thermal, geomorphological and geotechnical responses, Earth-Sci. Rev., 92, 117-171, doi:10.1016/j.earscirev.2008.12.002, 2009.

Hasler, A., Gruber, S., and Haeberli, W.: Temperature variability and offset in steep alpine rock and ice faces, Cryosphere, 5, 977-988, doi:10.5194/tc-5-977-2011, 2011.

Lehning, M., Bartelt, P., Brown, B., Russi, T., Stöckli, U., and Zimmerli, M.: SNOWPACK model calculations for avalanche warning based upon a new network of weather and snow stations, Cold Reg. Sci. Technol., 30, 145-157, doi:10.1016/S0165-232X(99)00022-1, 1999.

Lehning, M., Grünewald, T., and Schirmer, M.: Mountain snow distribution governed by an altitudinal gradient and terrain roughness, Geophys. Res. Lett., 38, L19504, doi:10.1029/2011GL048927, 2011.

Luetschg, M., Lehning, M., and Haeberli, W.: A sensitivity study of factors influencing warm/thin permafrost in the Swiss Alps, J. Glaciol., 54, 696-704, doi:103189/002214308786570881, 2008.

Magnin, F., Deline, P., Ravanel, L., Noetzli, J., and Pogliotti, P.: Thermal characteristics of permafrost in the steep alpine rock walls of the Aiguille du Midi (Mont Blanc Massif, 3842 m a.s.l.), Cryosphere, 9, 109-121, doi:10.5194/tc-9-109-2015, 2015.

Marmy, A., Salzmann, N., Scherler, M., and Hauck, C.: Permafrost model sensitivity to seasonal climatic changes and extreme events in mountainous regions, Environ. Res. Lett., 8, 035048 9pp, doi:10.1088/1748-9326/8/3/035048, 2013.

Myhra, k.S., Westermann, S., and Etzelmüller, B.: Modelled Distribution and Temporal Evolution of Permafrost in Steep Rock Walls Along a Latitudinal Transect in Norway by CryoGrid 2D, Permafr. Periglac. Process., doi: 10.1002/ppp.1884, 2015.

Noetzli, J., and Gruber, S.: Transient thermal effects in Alpine permafrost, Cryosphere, 3, 85-99, doi:10.5194/tc-3-85-2009, 2009.

Noetzli, J., Gruber, S., Kohl, T., Salzmann, N., and Haeberli, W.: Three-dimensional distribution and evolution of permafrost temperatures in idealized high-mountain topography, J. Geophys. Res., 112, F02S13, doi:10.1029/2006JF000545, 2007.

Phillips, M., Haberkorn, A., Draebing, D., Krautblatter, M., Rhyner, H., and Kenner, R.: Seasonally intermittent water flow through deep fractures in an Alpine Rock Ridge: Gemsstock, Central Swiss Alps, Cold Reg. Sci. Technol., 125, 117-127, doi:10.1016/j.coldregions.2016.02.010, 2016.

Pogliotti, P.: Influence of Snow Cover on MAGST over Complex Morphologies in Mountain Permafrost Regions, Ph.D. thesis, 79 pp., University of Torino, Torino, Italy, 2011.

Sommer, C.G., Lehning, M., and Mott, R.: Snow in a very steep rock face: accumulation and redistribution during and after a snowfall event, Front. Earth Sci., 3, Article 73, doi:10.3389/feart.2015.00073, 2015.

Unsworth, M.H., and Monteith, J.L.: Long-wave radiation at the ground I. Angular distribution of incoming radiation. Q. J. Roy. Meteor. Soc., 101, 13 – 24, doi: 10.1002/qj.49710142703, 1975.

[revised manuscript text omitted]

~~Traditional two-dimensional (2d) permafrost maps, based on statistical models, can serve as indicators of potential permafrost occurrence, but are limited in their ability to represent physical processes such as snow redistribution by avalanching and wind (Hoelzle et al., 2001), three-dimensional (3d) topographical effects in the ground (Noetzli et al., 2007) and transient changes (Harris et al., 2009).complex,faces with complex micro-topography, of atmosphere-surface interactionsand their modulating factors (e.g. local terrain shading, snow) that influence subsurface properties in complex topography,h. Fiddes et al. (2015) used the physics-based land surface model GEOtop (Endrizzi et al., 2014) to successfully model both snow cover and ground temperature evolution over the entire European Alps with a resolution of 30 m. However, this approach cannot capture the small-scale variability of the local surface energy balance and consequently of the ground thermal regime in complex, moderately inclined terrain (Gubler et al., 2011; Riseborough et al., 2008), as well as in steep~~

rock faces with complex micro topography (Haberkorn et al., 2015a; Hasler et al., 2011). The coarse grid resolution cannot account for phenomena such as lateral heat fluxes at the rock surface, which are defined here as heat fluxes with differing directions (parallel, vertical, as well as perpendicular to the rock surface) caused by pronounced temperature gradients induced by rock outcrops, by adjacent snow covered and snow free rock portions or by water flow at the rock surface. Lateral heat fluxes cause strongly variable ground surface temperatures in rock slopes (Haberkorn et al., 2015a), as well as between mountain sites (Gruber et al., 2004c; Wegmann et al., 1998) leading to substantial 3d thermal effects at depth of steep rock walls with convex topography (Noetzli et al., 2007; Noetzli and Gruber, 2009).

However, before a suitable investigation of 3d subsurface heat flow is feasible in steep bedrock, the strongly variable spatial and temporal ground surface boundary conditions and their driving factors need to be modelled realistically and with a high spatial resolution. One of these driving factors is the snow cover and its influence on the rock thermal regime, which has only recently been studied in steep bedrock (Gruber et al., 2004b; Gruber and Haeberli, 2007; Hasler et al., 2011). In steep rock walls exceeding 50° snow was often neglected for modelling purposes (Fiddes et al., 2015; Gruber et al., 2004a; Mittaz et al., 2002; Noetzli et al., 2007). In general it was assumed that wind and gravitational transport (avalanching or sloughing) remove the snow from steep rock (Blöschl and Kirnbauer, 1992; Gruber Schmid and Sardemann, 2003; Pogliotti, 2011; Winstral et al., 2002). We argue that rock temperature modelling cannot be done realistically without taking snow into account, since rock slopes in the Alps are generally heterogeneous, fractured and thus partly snow covered (Gruber et al., 2004a; Hasler et al., 2011).

Recent studies based on terrestrial laser scanning (TLS) have shown that snow accumulates in steep, rough rock walls (Haberkorn et al., 2015a; Sommer et al., 2015; Wirz et al., 2011), especially due to micro topographic effects, such as short distances to rock ledges on which snow can accumulate (Haberkorn et al., 2015a). The highly variable spatial and temporal distribution of the snow cover strongly influences the ground thermal regime of steep rock faces (Haberkorn et al., 2015a, b; Magnin et al., 2015) due to high surface albedo and low thermal conductivity of the snow cover, as well as energy consumption during snow melt (Bernhard et al. 1998; Keller and Gubler, 1993; Zhang, 2005). In gently inclined, blocky terrain effective ground surface insulation from cold atmospheric conditions were observed and modelled for snow depths exceeding 0.6 to 0.8 m (Hanson and Hoelzle, 2004; Keller and Gubler, 1993; Luetschg et al., 2008), while cooling effects on rock temperatures were found for snow depths smaller 0.15 m (Keller and Gubler, 1993). In contrast, Haberkorn et al. (2015a) found that snow depths exceeding 0.2 m were enough to have an insulating effect in steep, bare bedrock. Such amounts are likely to accumulate in steep, high rock walls with a certain degree of fracturing. Indeed, a warming effect of the snow cover on mean annual ground surface temperature (MAGST) was observed by Haberkorn et al. (2015a) and Magnin et al. (2015) in shaded rock walls, whilst in moderately inclined (45° 70°) sun exposed rock walls Hasler et al. (2011) suggest a reduction of MAGST of up to 3 °C compared to estimates in near-vertical, compact rock, due to snow persistence during the months with most intense radiation. However, a thick snow cover smoothes the variability of MAGST between N S oriented ridges by around 4 °C (Haberkorn et al., 2015a) compared to compact, near-vertical bedrock (Gruber et al., 2004a; Hasler et al., 2011; Noetzli et al., 2007).

In order to assess contrasting influences of heterogeneous snow distribution on the ground thermal regime, the integration of the snow cover in distributed energy balance modelling of steep bedrock is necessary. The one-dimensional (1d) modelling approach used by Haberkorn et al. (2015b) to investigate the influence of the snow

cover on near surface rock temperature (NSRT) and the rock thermal regime demonstrated that the SNOWPACK model (Lehning et al., 2002a,b; Luetschg et al., 2003) is able to simulate the effect of a snow cover on rock temperature. However, to account for the complex micro topography in rough rock walls and their influence on local shading effects, small scale snow distribution and rock temperatures, a spatially distributed energy balance approach is necessary.

[revised manuscript text omitted]

~~SNOWPACK simulates the temporal evolution of the vertical transport of mass and energy, as well as phase-change processes for a variety of layers within the seasonal snowpack and in the ground (Luetschg et al., 2003, 2008; Wever et al., 2015) for each single grid cell, solving the heat transport equation using a finite element method (Bartelt and Lehning, 2002). The albedo formulation for snow is calculated by a statistical model for~~

**3.3.2 3.1.7 Model setup**

For this application tThe model was driven by meteorological data measured by the on-site AWS Gemsstock (Fig. 1a, 2869 m a.s.l.). Measured aAir temperature, relative humidity, wind speed and direction, as well as incoming short- and longwave radiation data and (indirectly) snow depth were used as input data, pre-processed, as well as spatially interpolated and parameterized with the MeteoIO library (Bavay and Egger, 2014). Precipitation was provided to the model as described in Sect. 3.1.3. located at the foot of the northern rock slope (Fig. 1a). The AWS provided meteorological data during most of the investigation period 2012-2014. Measured air temperature, relative humidity, wind speed and direction, incoming short- and longwave radiation and (indirectly) snow depth were used as input data. Meteorological data and gGaps in meteorological data were corrected according to Haberkorn et al. (2015b).

The DEM is derived from the high-resolution TLS. data carried out at Gemsstock in the snow-free N and S facing rock walls in summer using a RIEGL *VZ6000* scanner at a grid resolution of 0.2 m and with a domain size of 4460 $m^2$ (Figs. 1a, b). On the basis of the DEM, Based on the DEM, the land-use classification was divided into two groups with varying rock properties, depending on whether the grid cells were N or S facing. The rock ground is simulated to 20 m depth, divided into 24 layers of varying thickness ranging from 0.02 m at the surface to 4 m at the bottom of the substrate. For each classification, the rock layers were initialized with different layer temperatures based on borehole rock temperatures measured on-site (Fig. 1c, PERMOS 2013). The rock was assumed to be 99 % solid with 1 % pore space containing ice (N facing grid cells) or water (S facing grid cells) to account for near-surface fracture space to a depth of 0.5 m. Unfractured rock with a solid content of 100 % was assumed between 0.5 and 20 m depth. The appropriate typical physical properties of the granodiorite bedrock were based on obtained from Cermák and Rybach (1982): with the a rock density of is 2600 kg $m^{-3}$, a the specific heat capacity of is 1000 J $kg^{-1}K^{-1}$, a the thermal conductivity of is 2.8 W $m^{-1}K^{-1}$ (on the S facing grid cells) side respectively and 1.9 W $m^{-1}K^{-1}$ (on the N facing grid cells) slope, as discussed in Haberkorn et al. (2015b). The rock albedo is assumed to be 0.15 and an aerodynamic roughness length of 0.002 m over snow is used for simulations. Down to 0.5 m depth the rock was assumed to be 99 % solid and 1 % pore space containing ice (N slope) or water (S slope) to account for near-surface fracture space. A solid content of 100 % was assumed between 0.5 m and 20 m depth. the rock layers of the SNOWPACK model were initialized with different layer temperatures based on measured borehole rock temperatures (Haberkorn et al., 2015b), depending on whether the grid cell was N or S facing. The ground is simulated to 20 m depth, divided into 24 layers of varying thickness ranging from 0.02 m at the surface to 4 m at the bottom of the substrate. The appropriate typical physical properties of the granodiorite bedrock were obtained from Cermák and Rybach (1982): the rock density is 2600 kg $m^{-3}$, the specific heat capacity is 1000 J $kg^{-1}K^{-1}$, the thermal conductivity is 2.8 W $m^{-1}K^{-1}$ on the S side and 1.9 W $m^{-1}K^{-1}$ on the N slope, as discussed in Haberkorn et al. (2015b). The rock albedo is assumed to be 0.15 and an aerodynamic roughness length of 0.002 m over snow is used for simulations. Down to 0.5 m

 Although The geothermal heat flux is most likely negligible in the narrow, steep and complex Gemsstock ridge due to strong topographic (Kohl, 1999) and 3d thermal effects (Noetzli et al., 2007), a constant upward ground heat flux had to be applied.  $Q_{ground}$ is  assumed to be 0.001 W m$^{-2}$ at 20 m depth.  to ensure a marginal impact of the lower boundary condition on the analysed rock thermal regime close to the surface.

All  simulations were run in parallel mode on the same computer cluster as a 32 core process, requiring around 15 days for a two-year simulation. Simulations were also performed for coarser resolutions (1 m, 5 m) to analyse the loss of model accuracy for lower computational costs.

~~The Alpine3D output at grid points corresponding to the 30 NSRT locations have a temporal resolution of two hours, according to the time resolution of the measured validation data. A detailed point validation of modelled and measured snow depth (derived by TLS) and NSRT data was therefore feasible. Errors made in modelling the distributed ground thermal regime in steep rock walls were assessed statistically for both the snow-covered and the snow-free approach using the mean bias error (MBE) and the correlation of determination (r²).~~

**3.1.8  Precipitation  input for Alpine3D**

*Terrestrial laser scanning (TLS)*

Snow depths acquired from TLS were used as input data for the precipitation scaling approach.  Snow depth distribution asthesinceRiegland snow depthsthe(Sommer et al., 2015)Parts of the NSRT measurement line and Alpine3D modelling domain were not visibledue tois assumedSnow depths acquired from TLS were used as input data for the precipitation scaling approach discussed in detail in Sect. 3.3.3.~~

**Precipitation scaling**

To model the snow cover in steep rock walls  the

availability of high-resolution spatially explicit snow depth distribution data provided by TLS were used. A precipitation scaling algorithm was used to drivefor driving the Alpine3D model, which, a precipitation scaling algorithm was applied. Alpine3D only uses precipitation as input data. As thisprecipitation data was not available for Gemsstock, precipitation was first calculated from the snow depth measured at the on-site AWS using a stand-alone SNOWPACK simulation. By using the snow depth driven mode of the SNOWPACK model, the snow depth measurements were used to determine the timing and amount of snowfall by interpreting increases in snow depth as fresh snowfall. Aaccording to Lehning et al. (1999) and Wever et al. (2015), SNOWPACK converts snowfall to precipitation while calculating both snow settlement and snow density based on a statistical model. To complete the resulting precipitation series, summer liquid precipitation was integrated used from the nearby MeteoSwiss AWS Gütsch (2287 m a.s.l., 6 km north of Gemsstock; Haberkorn et al. 2015b).

Secondly, for each grid cell, scaling factors were calculated based on tThe ratio between measured snow depth at the AWS and the distributed snow depth of each grid cell measured by TLS at the date of the TLS campaign.was used These scaling factors were then used to scale the two-year precipitation time series derived from the SNOWPACK simulation for the AWS Gemsstock for each grid cell ofonto the DEM. We refer to this method as precipitation scaling, which provides grids of spatially distributed precipitation amounts for Alpine3D input. However, d Data gaps in the TLS (Fig. 2b,c) lead to data gaps in the precipitation scaling grid. For grid cells lacking a precipitation scaling factor, the precipitation measured at the AWS Gemsstock was assumed, resulting in erroneously modelled snow depths and ground rock temperatures at these locations. For the analysis of tThe Alpine3D grid output with incomplete TLS wasthose grid cells have not been used therefore extracted to avoid errors in the analysis.

Thirdly, Precipitation scaling and an Alpine3D model runs were carried out using scaled precipitation offor each of the four TLS campaigns. However, tThe modelled snow depth and NSRT data coincided best with validation data when using scaled precipitation and from snow depth data based on the TLS data obtained onof 191 December 2012. Henceforth, the modelled results analysed and discussed here are only based on this TLS data. The use of an early winter TLS is preferred, since the early winter snow depth distribution best represents winter snowfall events. TLS data obtained in spring already contain ablation processes. Fig. 3 provides justification for the choice of the TLS used. Here the distribution of the ratio modelled to measured snow depth is shown for the 4 TLS available. The TLS data measured on 19 December 2012 is centred around 1, as well as the snow depth curves on the dates of two (7 June 2013, 28 January 2014) of the other three TLS campaigns. Those TLS show satisfactory agreement between modelled and measured snow depths. In contrast, simulations using the TLS on 11 December 2013 overestimate snow depths and have a wider spread compared to the ratio of the other scans. This may be due to snow depth distribution differences due to varying wind conditions in the rock walls. In early winter 2013-2014 a large proportion of snowfall events occurred with southerly winds, whereas in general snowfalls were accompanied by northwesterly flows. A quantitative analysis of the precipitation scaling approach is currently being evaluated (Voegeli et al., submitted). 3 (Fig. 2b, c). The reason for this is unclear but early winter TLS best represent winter snow fall events. TLS carried out in spring already contain processes modelled by Alpine3D, such as ablation. Henceforth, the modelled results analysed and discussed here are based solely on the TLS of 11 December 2013. The use of only one TLS is is additionally justified by annually recurring micro-topography driven snow depth distribution and snow depth patterns caused by the micro-topography, which have been observed in steep rock walls by Haberkorn et al. (2015a), Wirz et al. (2011),

Sommer et al. (2015) and Wirz et al. (2011)Haberkorn et al. (2015a). We refer to this method as precipitation scaling, which provides grids of spatially distributed precipitation amounts for Alpine3D input.

**3.2 Sensitivity study**

[revised manuscript text omitted]

~~In contrast to the radiation budget of the snow-covered steep N facing location of N7, the monthly evolution of the modelled surface energy balance components for snow free conditions (Fig. 7b) differed strongly. $Q_{net}$ increased uniformly from negative values in winter to maximum values in summer. During the months with low solar elevation (November-January) the energy loss by $Q_{net}$ was smaller on bare rock due its lower albedo and emissivity. The sensible heat flux towards the surface was smaller for snow free conditions, but the ground heat~~

released in fall and winter was increased due to the absence of snow. $Q_{ground}$ was directed into the rock during spring and summer. From March to July the energy gained by $Q_{net}$ was considerably higher compared to the snow-covered case due to the absence of snow, which was mainly compensated by $Q_{sensible}$. $Q_{ground}$ was low due to the insulating snowpack and positive, since the measured permafrost signal could not be reproduced by the model.

**4.3.24.2.1 S facing slope**

Although the monthly evolution of the energy fluxes in the steep, snow-covered and sun-exposed location of S9 (Fig. 7c) were similar to those of N7, variations in the magnitude of the fluxes were observed. During November and March energy was lost by $Q_{net}$ and $Q_{latent}$. However, $Q_{net}$ was less negative than in the N face due to strong insolation in winter. During the snowmelt period from April to June/July the energy budget was positive. However, the magnitudes of the fluxes were smaller in the slightly less inclined S slope compared to the steeper north west exposed location of N7.

The temporal evolution of the modelled energy fluxes at locations lacking snow (Fig. 7d) varied strongly. $Q_{net}$ was positive throughout the whole year, displaying a sinusoidal cycle with minimum values in winter and maxima in summer. The strong $Q_{net}$ input in winter is caused by stronger direct solar radiation input on steep S facing slopes due to the low solar elevation and perpendicular angle of incoming solar radiation.

**4.32 NSRT variability at selected points locations**

The measured and simulated NSRT evolution atin 0.1 m depth in theat four selected NSRT logger locations with differing snow conditions (no snow, snow; Table 1) in both the N and the S facing rock walls are illustrated in Fig. 37. PAoint bias assessmentverification (r², MBE, MAE) was performed between measured and modelled NSRT for each individual location. First the NSRT evolution at for both the snow-free locations is described, and then the modulating effect of the snow cover on NSRT is emphasizedand the snow covered case (Table 2, Fig. 4).

**4.4.4 Measured NSRT variability at snow-free locations**

At NSRT locations lacking snow, measured NSRTs closely followed air temperature in the shaded N face (N3, in Table 1. Fig. 37d) while pronounced daily NSRT amplitudes of up to 10 °C could be observed inat the sun-exposed locationsrock wall (R2 in Table 1, Fig. 37e) during the whole investigation period. The topography driven MANSRT difference between the N (N3) and the S (R2) facing locations was 3.6 °C. Similar to the snowy locations, the MANSRT was up to 0.6 °C higher here in 2013 2014 compared to 2012 2013, but here induced by the 0.9 °C warmer mean annual air temperature in 2013-2014 (Table 3).

At logger locations lacking snow (N3 and, R2 in Table 2, Fig. 3d, e) the modelled NSRT evolution was in good accordance with measured NSRT with r² = 0.82-0.94 (Figs. 8c, f). pronounced daily NSRT amplitudes were measured and modelled during the whole investigated period (up to 12 °C). Measured and modelled NSRT evolution was in good accordance (r² up to 0.92, Fig. 4c, f). However,Although NSRT evolution was successfully reproduced by Alpine3D, the MAE between measured and modelled NSRT were 2.2/2.3 °C at N3 and 2.6/2.8 °C at R2 (Table 1). tThe MBE between measured and modelled NSRT was up to -2.01/-1.7 °C for N3 and -2.5/-2.1 °C for R2for both the N and the S facing rock slopes, indicating alwayspersistently colder modelled NSRT conditions, which is also The largest differences between measured and modelled daily NSRT amplitudes were mainly observed during winter in both rock wallsillustrated by dT in Figs. 7e and d.

**4.4.5 NSRT variability at snow-covered locations**

4.3.3

At locations favouring the accumulation of a thick snowpack the NSRT evolution was strongly controlled by snow for around 7.5 to 9 months of the year in both the N (N7 in Table 1, Fig. 3b) and the S (S9 in Table 1, Fig. 3c) facing rock walls. After the onset of the continuous snow cover in October/November the rock surface was partly decoupled from atmospheric influences. In the N facing slope (N7, Fig. 7b) measured NSRT oscillations were damped, but continuously decreased down to -4 °C, and thus clearly showing the occurrence of a permafrost signal at this location, while in the S facing slope (S9, Fig. 7c) measured NSRT remained close to 0 °C. Although tThe timing of snow cover onset and disappearance were similar in both the N and the S facing slope (Table 2). the start of the zero curtain period was delayed by up to 1.5 months in the N facing slope (beginning of May). Mean annual near-surface rock temperature (MANSRT) differences were only up to 2.4 °C between the N (N7) and the S (S9) facing locations due to the thick snow cover. In 2013-2014 Tthe snow cover onset during both years was similar compared to 2012-2013, while the onset of the zero curtain period was two weeks earlier and the snow disappearance up to 4 weeks earlier (mid-June) in 2013-2014. The latter caused up to 0.3 °C (S9) respectively 0.4 °C (N7) warmer MANSRTs in 2013-2014 than in the previous year due to the snow-free conditions during the weeks with most intense solar radiation (mid-June to mid-July).

At the locations accumulating a thick snow cover Tthe temporal evolution of modelled NSRTs and their temporal evolution are in good accordance with the measured ones at locations accumulating a thick snow cover (Fig. 4a, d) in both the Nshaded (N7) and the sun-exposedthe S (S9) facing slopes with r$^2$ = 0.80-0.94 (Figs. 8a, d). When comparing measured and modelled NSRT evolution, the modelled timing of the snow cover onset, of the zero curtain period and of snow disappearance was similar (Figs. 7b, c and dT in these). This underlines that satisfactory modelled snow cover duration is the most important factor influencing modelled NSRT evolution, rather than accurately modelled absolute snow depths. Daily vVariations between measured and modelled NSRT are small at the S facing S9 with a MAE of 0.6/0.6 °C and a MBE of -0.3/-0.4 °C, indicating too low modelled NSRTs in summer. below 4 °C in summer (dT in Fig. 3b, e). At N7 measured and modelled NSRT fit well together Dduring the snow-coveredfree period. measured and modelled NSRT and therefore MANSRT are in auspicious accordance in the S facing slope (MBE 0.15 °C), while mwhile measured NSRTs, as well as MANSRT are on average 1.2 °C colder than modelled ones in the N facing slopeduring the snow-covered period (Figs. 3b, 4a, Table 2)resulting in a MBE of 0.8/0.8 °C and a MAE of 1.1/1.0 °C (Table 1).

However, in both the shaded and sun-exposed rock slopes the modelled timing of the snow cover onset, the zero curtain period and the snow disappearance was similar to the measured ones (Table 2).

At NSRT locations lacking snow, NSRTs closely followed air temperature in the shaded N face (N3 in Table 1, Fig. 3d) while pronounced daily NSRT amplitudes up to 10 °C could be observed at sun-exposed locations (R2 in Table 1, Fig. 3e) during the whole investigation period. The topography-driven MANSRT difference between the N (N3) and the S (R2) facing locations was 3.6 °C. Similar to the snowy locations, the MANSRT was up to 0.6 °C higher here in 2013-2014 compared to 2012-2013, but here induced by the 0.9 °C warmer mean annual air temperature in 2013-2014 (Table 3).

**4.3.4 Modelled NSRT variability**

The modelled NSRTs and their temporal evolution are in good accordance with the measured ones at locations accumulating a thick snow cover (Fig. 4a, d) in both the N (N7) and the S (S9) facing slopes. Daily variations

between measured and modelled NSRT are below 4 °C in summer (dT in Fig. 3b, c). During the snow-covered period measured and modelled NSRT and therefore MANSRT are in auspicious accordance in the S facing slope (MBE 0.15 °C), while measured NSRT, as well as MANSRT are on average 1.2 °C colder than modelled ones in the N facing slope (Figs. 3b, 4a, Table 2). However, in both the shaded and sun exposed rock slopes the modelled timing of the snow cover onset, the zero curtain period and the snow disappearance was similar to the measured ones (Table 2).

At logger locations lacking snow (N3, R2 in Table 2, Fig. 3d, e) pronounced daily NSRT amplitudes were measured and modelled during the whole investigated period (up to 12 °C). Measured and modelled NSRT evolution was in good accordance ($r^2$ up to 0.92, Fig. 4e, f). However, the MBE between measured and modelled NSRT was up to 2.0 °C for both the N and the S facing rock slopes, indicating always colder modelled NSRT conditions. The largest differences between measured and modelled daily NSRT amplitudes were mainly observed during winter in both rock walls.

**4.3.5 Modelled NSRT variability with forced snow-free conditions**
**4.4.6 Thermal effect of snow**

The previously discussed modulating influence of the snow cover on the surface energy balance and its effects on the ground thermal regime can be emphasized by comparing NSRTs at the snow-covered N7 and S9 to the modelled snow-free scenario at these locations (blue lines in Figs. 7b, c). Although measured NSRT confirm the accumulation of a thick snow cover at the locations N7 and S9, NSRT were also modelled here assuming snow-free conditions. Using the snow-free scenario, Mmodelled NSRT oscillationsevolutions of N7 and S9 were similar to the snow-free locations of N3 and R2, discussed previously. MANSRTs were up to -2.5 °C in the shaded and up to 1.8 °C in the sun exposed slopes. Modelled NSRT oscillations were pronounced during the whole study period, indicating a permanent energy exchange between the atmosphere and the rock. MANSRTs were up to -2.8/-1.9 °C at the shaded N7 and up to 0.4/1.31.8 °C atin the sun-exposed S9slopes. This contradicts the NSRT measurements at these locations (Section 4.3.2). Measurements reveal a permanent insulation of the rock by a continuous snowpack between October/November and June/July. Neither cold atmospheric conditions in winter, nor strong insolation and warm air temperatures between May and July (all energy available used for snow melt, Figs. 6a, c) affected the rock thermal regime below the snowpack. Thus the potential thermal effect of a thick, long lasting snowpack accumulating in steep rock can locally be quantified: at locations accumulating a long lasting, insulating snow cover Therefore, mthe measured MANSRTs were 2.9/2.4 °C higher in the shaded and 2.0/1.4 °C higher in the sun-exposed rock wall, while comparing to modelled MANSRT forof the snow-free scenario (Table 1).conditions were up to 3.7 °C colder in the shaded and up to 1.6 °C colder in the sun-exposed rock wall locations, when comparing to measured and modelled MANSRT accumulating an insulating snow cover. MANSRTs were up to -2.5 °C in the shaded and up to 1.8 °C in the sun-exposed slopes. Therefore, modelled MANSRT for snow-free conditions were up to 3.7 °C colder in the shaded and up to 1.6 °C colder in the sun-exposed rock wall locations, when comparing to measured and modelled MANSRT accumulating an insulating snow cover. The negligence of snow in steep rock resulted in apparent deviations between measured and modelled (snow-free) NSRT causeingd the $r^2$ to decrease by 0.26/0.21 at N7 and 0.57/0.51 at S9( (Figs. 4b8b, e) and the MBE the MAE to increase by 4.1/2.7 °C at N7 and 4.7/3.4 °C at S9.(Table 2).

**4.44.5 MANSRT variability in the entire rock walls**

A comprehensive analysis of all 22 NSRT locations was used to evaluate the spatial performance of Alpine3D in modelling the potential effect of snow on NSRTs. Both the measured and modelled NSRT data of all 11 N facing locations and of all 11 S facing ones were used to calculate means of MANSRT, MBE and MAE At the 30 NSRT locations measured and modelled NSRT data were used to calculate MANSRT means, as well as the variability (standard deviation) of MANSRT overwithin the individual N and S facing rock walls (Table 3). Modelled MANSRT were also averaged for both slopes over the entire model domain.

**4.4.14.5.1 Measured MANSRT variabilitySnow-covered scenario**

The topography driven difference ofComparing the measured mean MANSRT between theaveraged over the entire entire N facing rock wall (up to 0.7 °C) with the MANSRT ofand the entire S facing rock wall (up to 2.9 °C) resulted in awere MANSRT difference of 3.6/3.2 °C. Such a small deviation is reasonable when taking into account that the rock walls are facing rather NW and SE than N and S (Fig. 1a, Appendix Table 1A), as well as considering the This was not as pronounced as expected in steep rock due to the accumulation of a thick snow cover at 7 of 11most locations in both the N and S slopes.

At the corresponding 22 grid cells, the modelled mean MANSRT difference for the snow-covered scenario across the entire N and S facing slope is 2.6/2.3 °C and thus around 1.0 °C lower than the measured values (Table 3). This is mainly caused by too low modelled NSRTs and thus MANSRTs, especially in the sun-exposed rock wall during snow-free periods (Fig. 9) and at locations without snow (N and S slopes) resulting in a MBE of -1.3/-1.0 °C. These results are supported by the model verification at the single locations in Section 4.3.2, but clearly show that model uncertainties increase on the rock wall-scale due to the pronounced spatial variability. Uncertainties while applying Alpine3D to simulate NSRT in steep rough rock implies a MAE of 1.6/1.7 °C for both the entire shaded and sunny rock wall.

The measured and modelled small-scale variability of MANSRT at all 22 NSRT locations and corresponding grid cells separated forwithin the individual N and S facing rock walls are illustrated in Fig. 9, as well as the modelled MANSRT variations for the entire model domain, depending on whether the grid cells are N or S facing. For all cases, the MANSRT variability within the individual N and S slopes was was highergreater in 2012-2013 (Fig. 5, Table 3), which is the result of two effects. Smaller MANSRT variability in 2013-2014 resulted from two compensating effects: In 20132-20134 both winter andthe mean annual air temperatures wasere 0.8 °C lowerwarmer, than in 2013-2014, causing MANSRTs at snow-free locations to increase decrease by around 0.6 °C. In contrast, MANSRTs at snow-covered locations in the N slope decreased increased by up to 0.4 °C due to an early onset of a long lasting, insulating snow cover. In early winter 2013-2014 the absence of a sufficiently thick, insulating snow cover in the beginning of winter 2013-2014 (Haberkorn et al., 2015a), which resulted in effective ground heat loss at these locations at snow-covered locations in the N slope(Haberkorn et al., 2015a).

This resulted from a combination of artificially low modelled MANSRT at locations lacking snow and artificially high modelled MANSRT at snow-covered locations during winter.

**4.4.2 Modelled MANSRT variability**

The modelled and measured MANSRTs and their annual trends coincide well in the N facing slope (Table 3). However, modelled MANSRT variability was higher (Table 3, Fig. 5) than measured data. This resulted from a

**4.5.2    Snow-free scenario**

[revised manuscript text omitted]

The implementation of 3d advective heat fluxes influencing already the rock surface and not just ground temperatures at depth (Noetzli et al., 2007) will be a crucial further step for modelling the ground thermal regime in steep bedrock.

**5.3 Influences of grid resolution**

The model performance was tested at different scales ranging from 0.2 m, 1m to 5 m. At locations with rough micro-topography the loss of information was big due to the aggregation of the initial DEM (0.2 m resolution) to 1 m and 5 m. Slope angles were only sampled <70° (1 m resolution) and <60° (5 m resolution), while in reality the rock was vertical. Aspects were displaced by up to 90°. This has a strong effect of the precipitation scaling and the modelled energy- and mass balance of the rock walls. Shortwave incoming radiation was inadequately modelled at locations with strongly varying micro-topography. However, on a monthly basis, errors in net radiation were smoothed. In both the N and the S facing rock walls modelled NSRTs confirm that a grid resolution of 1 m is acceptable to accurately model the snow cover and ground surface temperatures in steep rugged rock faces. The decrease in computational time by reducing the grid resolution from 0.2 m to 1 m, is significant (25 times smaller). Additionally, a DEM resolution of 1 m is considered to be precise enough to detect ledges within the rock face, which are essential for snow accumulation in steep rock (Haberkorn et al., 2015a; Sommer et al., 2015). At a resolution of 5 m the loss of topographic, as well as accurate snow depth information is huge and consequently snow distribution and the rock thermal regime were inadequately modelled in such complex terrain.

Mismatches of scaleing issues in distributed permafrost modelling arise often while validating the model results based on grids of tens to hundreds10s – 100s of metresers to point measurements (e.g. Gubler et al., 2011; Gupta et al., 2005; Schlögel et al., 2016). Here, a point- and spatial to point model validation of NSRTs was performed

at a dense network of 30 NSRT measurement locations. Additionally TLS derivedand snow depths grids were performed at different grid cell sizes (0.2 m, 1 m, 5 m; Table 4).compared to modelled snow depth grids at TLS recording date both with a resolution of 0.2 m. The data validation revealed similar MANSRT and MBE for both, the point- and the rock wall scale implying that fewer validation locations in the rock walls are sufficient. In both the N and the S facing rock walls, Tthe point- and spatial validation withto data atwith 1 m resolution is reasonable, to accurately model the snow cover and ground surface temperatures in steep rugged rock faces. The decrease in computational time by reducing the grid resolution from 0.2 to 1 m, is significant (25 times lower). Additionally, a DEM resolution of 1 m is considered to be precise enough to detect ledges within the rock face, which are essential for snow accumulation in steep rock (Haberkorn et al., 2015a; Sommer et al., 2015). At a resolution of 5 m the loss of topographic, as well as accurate snow depth information results in an inadequately modelled rock thermal regime.while 5 m resolution is already insufficient. Model runs at coarser spatial-scales are thus assumed to be unsuitable for modelling temperatures in complex steep rock walls, such as the Gemsstock ridge. Variations of surface processes due to micro-topographic inhomogeneity occur at small-scales, providing the motivation for high-resolution numerical modelling in complex topography in order to establish a basis of proper validation of grid-based model results.

**6  Conclusions**

The potential to model the strongly heterogeneous snow cover and its influence on the rock thermal regime on two rugged, steep mountain rock walls has been studied at the Gemsstock ridge (central Swiss Alps) over a two year period here. The results were obtained using the spatially distributed physics-based model Alpine3D in combination with a precipitation scaling approach. Modelling the impact of snow on ground temperatures in steep rock revealed potential errors made in recent ground temperature modelling when neglecting the evolution of the snow cover in terrain exceeding 50°.

Alpine3D simulates near surface rock temperatures and snow depth in the heterogeneous terrain accurately. The fine scale resolution of the model domain (0.2 m) and of the validation data allow to consider the strongly varying micro topography occurring in the rock walls, and thus the accumulation of In the rough rock walls, the a heterogeneously distributed snow cover was moderately well reproduced by Alpine3D with absolute snow depth differences varying between +1.5 and -1.0 m and a MAE between 0.47 and 0.77 m averaged over the entire rock walls. However, the snow cover duration was well reproduced by the model and proved to be most important for realistically NSRT modelling.

. The correction of winter precipitation input using a precipitation scaling method based on TLS greatly improves simulations of snow distribution and duration and thus of the rock thermal regime. Rock temperatures are convincingly modelled, although modelled NSRTs and thus MANSRTs are somewhat too low during snow-free periods and at locations without snow, as indicated by a MBE varying between -0.2 and -1.3 °C in the rock walls. Model verification suggests an MAE of 1.6/1.7 °C in both the entire shaded and sunny rock walls.

Remaining errors in snow depth and consequently rock temperature simulations are explained by inadequate snow settlement modelling, due to linear precipitation scaling, missing lateral heat fluxes in the rock, inadequate snow settlement and by errors due to shortwave radiation, air temperature and wind interpolation, which are very complex in such terrain.

The fine scale resolution of the model domain (0.2 m) and of the validation data allow to consider the strongly varying micro topography occurring in the rock walls, and thus the accumulation of a heterogeneously

[revised manuscript text omitted]

**Table 12. Topographic characteristics of selected NSRT logger locations with different snow conditions and the distance to the nearest ledge below (DLB). In addition analysis of observed (O) and predicted (P) snow cover duration, as well as observed MANSRT and predicted MANSRT for both snow-covered (PS) and snow-free (PSF) scenarios at selected NSRT locations and for the years 2012-2013 (12-13) and 2013-2014 (13-14). The MBE and MAE were calculated between observations and model predictions of the snow-covered respectively snow-free scenarios.**

| Logger (slope/aspect) | DLB (m) | Year | Snow cover duration O | Snow cover duration P | MANSRT [°C] P | MANSRT [°C] PS | MANSRT [°C] PSF | MBE [°C] PS | MBE [°C] PSF | MAE [°C] PS | MAE [°C] PSF |
|---|---|---|---|---|---|---|---|---|---|---|---|
| N7 (90°/289°) | 0.1 | 12-13 | 10 Oct-8 Jul | 13 Oct-4 Jul | 0.1 | 0.9 | -2.8 | 0.8 | -3.0 | 1.1 | 5.2 |
|  |  | 13-14 | 7 Oct-11 Jun | 11 Oct-13 Jun | 0.5 | 1.2 | -1.9 | 0.8 | -2.4 | 1.0 | 3.7 |
| N3 (90°/284°) | 10 | 12-13 | - | - | -1.4 | -3.6 | -3.6 | -2.1 | -2.1 | 2.3 | 2.3 |
|  |  | 13-14 | - | - | -0.8 | -2.5 | -2.5 | -1.7 | -1.7 | 2.2 | 2.2 |
| S9 (72°/165°) | 0 | 12-13 | 28 Oct-6 Jul | 31 Oct-12 Jul | 2.4 | 2.1 | 0.4 | -0.3 | -2.1 | 0.6 | 5.3 |
|  |  | 13-14 | 4 Nov-11 Jun | 9 Nov-20 Jun | 2.7 | 2.3 | 1.3 | -0.4 | -1.4 | 0.6 | 4.0 |
| R2 (58°/164°) | 15 | 12-13 | - | - | 2.2 | -0.3 | -0.3 | -2.5 | -2.5 | 2.8 | 2.8 |
|  |  | 13-14 | - | - | 2.7 | 0.6 | 0.6 | -2.1 | -2.1 | 2.6 | 2.6 |

**Table 2. Snow depth validation (MBE, MAE) between measured and modelled snow depths averaged over the entire N and S facing rock walls at the dates of the independent TLS campaigns. The MBE and MAE are in [m].**

| TLS campaign | Rock wall | MBE | MAE |
|---|---|---|---|
| 7 June 2013 | N | 0.25 | 0.81 |
| | S | 0.52 | 0.74 |
| 11 December 2013 | N | 0.73 | 0.75 |
| | S | 0.47 | 0.48 |
| 28 January 2014 | N | 0.42 | 0.59 |
| | S | 0.17 | 0.31 |

3795

**Table 3.** MANSRT, MAE and MBE [all in °C] calculated within the individual N and S facing rock walls at NSRT locations. The MAE and MBE w calculated between measurements (O) and model predictions of both the snow-covered (PS) and the snow-free scenarios (PSF)  at NSRT locations.  Additionally mean annual air temperature (MAAT) for the years 2012-2013 and 2013-2014 is shown.

| Scenario | Rock wall | 2012 2013 (2012-2013) | | | 2013 2014 (2013-2014) | | |
|---|---|---|---|---|---|---|---|
| | | MANSRT | MAE | MBE | MANSRT | MAE | MBE |
| North | N | -0.7 |  | | -0.5 |  | |
| PS | N | 1.0 | 1.6 | -0.4 | -0.6 | 1.7 | 0.1 |
| PSF | N | 3.3 | 3.9 | -2.3 | 2.3 | 0.7 | 1.8 |
| O | S | 2.9 |  | | 2.7 |  | |
| PS | S | 1.6 | 1.6 | -0.9 | 2.7 | 0.8 | -0.7 |
| PSF | S | 0.1 | 4.7 | 2.8 | 1.0.8 | 0.8 | 1.9 |
| MAAT | | -3.2 | | | -2.4 | | |

3800

3805

**Table 4. Differences in grid resolution: MANSRT, MAE and MBE [all in °C] calculated within the individual N and S facing rock walls at NSRT locations. The MAE and MBE were calculated between measurements (O) and model results of the snow-covered scenario (PS) at NSRT locations for 0.2 m, 1 m and 5 m grid resolution.**

| | | | 1 September 2012-31 August 2013 (2012-2013) | | | 1 September 2013-31 August 2014 (2013-2014) | | |
|---|---|---|---|---|---|---|---|---|
| Location Scenario | Rock wall | Resolution [m] | MANSRT | MAE | MBE | MANSRT | MAE | MBE |
| Measured North O | N | | -0.7 | | | -0.5 | | |
| Modelled North snow PS-0.2m | N | 0.2 | -1.0 | 1.6 | -0.4 | -0.6 | 1.7 | -0.2 |
| Modelled North snow 1m PS | N | 1 | -0.6 | 1.7 | 0.0 | -0.2 | 1.5 | 0.3 |
| Modelled North snow 5m PS | N | 5 | 0.5 | 1.8 | 1.1 | 0.7 | 1.9 | 0.9 |
| Measured South O | S | | 2.9 | | | 2.7 | | |
| Modelled South snow 0.2m PS | S | 0.2 | 1.6 | 1.6 | -1.3 | 1.7 | 1.7 | -1.0 |
| Modelled South snow 1m PS | S | 1 | 1.7 | 1.7 | -1.2 | 2.1 | 1.8 | -0.6 |
| Modelled South snow 5m PS | S | 5 | 2.4 | 1.8 | -0.4 | 2.4 | 1.7 | -0.3 |

3810

[Figure]

**Figure 1. The Gemsstock study site: (a) The  Alpine3D model domain with slope angles (red rectangle) based on TLS data, as well as the locations of the AWS and the NSRT devices. The location of Gemsstock in the Swiss Alps is shown in the top left inset. (b) 3d view of the DEM of  Gemsstock, as well as (c) the cross-section of the**

3815

**Gemsstock ridge with** **all 30 NSRT locations** . **Photographs** **showing the (****d) N and (****e) S rock faces and the measurement set-up. (b-****e) Black dots indicate the locations of the 30 NSRT locations and selected ones, discussed in further detail are highlighted in pink and labelled.**

3820

[Figure]

3825 **Figure 2. Flow chart of the methods applied in order to run the numerical model Alpine3D and to validate the model output at both the point- and the spatial-scale.**

3830

[Figure]

**Figure 3. Histogram of measured and modelled snow depth data. Solid lines denote the distribution of the ratio modelled over measured snow depth for the 4 TLS available. The TLS of 19 December 2012, 7 June 2013 and 28 January 2014 are centred by 1. The TLS of 19 December 2012 was used for precipitation scaling and shows the best agreement between modelled and measured snow depths.**

3835

[Figure]

**Figure 42**.  **Snow depth** distribution: (a-c) measured based on TLS, (d-f) modelled at the same dates as the TLS campaigns, (g-i)  **differences Δ between**

3840

3845 modelled and measured snow depth and (j-l) measured snow depths as function of modelled snow depths. distribution on 11 December 2013: 3d TLS data for (b) the N facing and (c) the S facing slopes, as well as 2d data for both the N and the S facing slopes (d) based on TLS, (e) modelled with Alpine3D and (f) their differences (measured - modelled). For better visualization differences are only illustrated here in the range of ±0.5 m, although variations are up to -1 m (only few grid cells). (b-f) GreyRed dots indicate the locations of NSRT loggers and selected ones are highlighted in pink and labelled.

3850

[Figure]

**Figure 5.** Snow depth evolution (lines) measured at the flat field AWS Gemsstock ( AWS), as well as modelled  at the NSRT locations discussed in detail (N7,  S9, N3, R2). Snow depths at the NSRT locations obtained by TLS  are shown as blue, red,  grey  dots and pink markers.  The locations of N3 and R2 lack snow  for the entire investigation period  Data of the TLS campaign on 19 December 2012 is also shown here, although the measured snow depth was used for precipitation scaling.

[Figure]

**Figure 6. Modelled monthly means of all energy balance components for two selected NSRT locations. N7 (top) faces north-west and S9 (bottom) south-east. To illustrate the influence of the snow cover on the surface energy balance, the energy fluxes are shown for the snow-covered (left) and the snow-free scenarios (right). Energy fluxes are considered positive when directed towards the snowpack surface. $Q_{snow}$ is the energy available to melt the isothermal snowpack and is thus illustrated here as an energy sink.**

[Figure]

**Figure 7. (a) Daily mean air temperature at the AWS Gemsstock. (b-e)Measured and modelled daily mean NSRT are shown for four selected locations in the N and the S facing rock walls representing typical snow conditions (snow, no snow). At locations accumulating snow (N7, S9)  modelled NSRTs are shown for both the snow-covered and the snow-free scenarios, while the NSRT differences (dT) were only shown between measured and modelled  snow-covered conditions. At locations without snow (N3, R2) measured and modelled NSRT differences (dT) are also shown.**

3870

[Figure]

**Figure 84. Two year data showing the relation between measured and modelled NSRT data for both (a,d) snow-covered (a, d) and (b,e) forced snow-free conditions scenarios(b, e), as well as for (c,f) generally snow-free NSRT locations (e, f). The mean annual r$^2$, as well as the linear relation between measured and modelled NSRT data are shown.**

[Figure]

**Figure 5̶9.** MANSRT variability within the individual N (left) and S (right) facing rock walls for the years 2012-2013 (12-13) and 2013-2014 (13-14). The MANSRT variability in the rock walls were based on 22 measured NSRTs, 11 facing N and 11 facing S. Measured MANSRT variabilities are compared to modelled MANSRT differences calculated at the grid cells of̶ ̶c̶a̶l̶c̶u̶l̶a̶t̶e̶d̶ ̶a̶t̶ NSRT locations̶ ̶o̶n̶ ̶b̶a̶s̶i̶s̶ ̶o̶f̶ ̶m̶e̶a̶s̶u̶r̶e̶d̶ ̶N̶S̶R̶T̶, shown for both the snow-covered and the snow-free scenarios. ̶a̶s̶ ̶w̶e̶l̶l̶ ̶a̶s̶ ̶m̶o̶d̶e̶l̶l̶e̶d̶ ̶N̶S̶R̶T̶ ̶d̶a̶t̶a̶ ̶f̶o̶r̶ ̶b̶o̶t̶h̶ ̶s̶n̶o̶w̶-̶c̶o̶v̶e̶r̶e̶d̶ ̶a̶n̶d̶ ̶s̶n̶o̶w̶-̶f̶r̶e̶e̶ ̶c̶o̶n̶d̶i̶t̶i̶o̶n̶s̶ ̶f̶o̶r̶ ̶t̶h̶e̶ ̶y̶e̶a̶r̶s̶ ̶2̶0̶1̶2̶-̶2̶0̶1̶3̶ ̶(̶1̶2̶-̶1̶3̶)̶ ̶a̶n̶d̶ ̶2̶0̶1̶3̶-̶2̶0̶1̶4̶ ̶(̶1̶3̶-̶1̶4̶)̶,̶In addition to the MANSRT differences calculated at all 22 NSRT locations, the modelled MANSRT variability of each grid cell of the entire model domain is shown, depending on whether the grid cell is N or S facing.̶-̶ ̶a̶s̶ ̶w̶e̶l̶l̶ ̶a̶s̶ ̶m̶o̶d̶e̶l̶l̶e̶d̶ ̶M̶A̶N̶S̶R̶T̶ ̶v̶a̶r̶i̶a̶b̶i̶l̶i̶t̶y̶ ̶w̶i̶t̶h̶i̶n̶ ̶t̶h̶e̶ ̶e̶n̶t̶i̶r̶e̶ ̶N̶ ̶a̶n̶d̶ ̶S̶ ̶f̶a̶c̶i̶n̶g̶ ̶m̶o̶d̶e̶l̶ ̶d̶o̶m̶a̶i̶n̶.̶ The median is marked with a red horizontal line in each box, the mean is additionally plotted as a red asterix, the box edges are the 25[th] and the 75[th] percentiles, the whiskers extend to the 2.5 % and 97.5 % quantiles and outliers are plotted as individual crosses.

[Figure]

**Figure 6̶10. Modelled MANSRT distribution in the N (left) and the S (right) facing slopes for  snow- free  scenario (top) and the snow-coveredfree  one (middle), as well as their differences (bottom; snow-covered – snow-free). Arrows indicate rock outcrops and rock dihedrals partly shadowing the NSRT locations, which are marked by  grey dots (selected locations in pink and**

3900  labelled). The  model results is only shown for the year 2012-2013, but MANSRT averaged over the individual N respectively S facing rock walls are given for both study years, as well as the difference between the MANSRTs of the snow-covered and snow-free scenarios (dMANSRT).

 **Appendix: Table 1A. Slope angle (slope) and aspect [both in °] measured at the 22 NSRT locations, as well as their topography in the model domain with varying grid cell size.**

| Location | Measured | | Cell size 0.2 m | | Cell size 1 m | | Cell size 5 m | |
|---|---|---|---|---|---|---|---|---|
| | slope | aspect | slope | aspect | slope | aspect | slope | aspect |
| N1 | 34 | 4 | 53 | 9 | 52 | 8 | 28 | 288 |
| N2 | 47 | 23 | 53 | 6 | 66 | 341 | 50 | 309 |
| N3 | 90 | 284 | 83 | 281 | 70 | 288 | | |
| N4 | 84 | 296 | 36 | 264 | 62 | 282 | | |
| N5 | 72 | 226 | 55 | 250 | 57 | 284 | | |
| N6 | 68 | 324 | 75 | 288 | 61 | 266 | 52 | 289 |
| N7 | 90 | 289 | 69 | 267 | 59 | 268 | | |
| N8 | 74 | 204 | 56 | 228 | 44 | 282 | 56 | 292 |
| N9 | 80 | 340 | 77 | 313 | 68 | 303 | | |
| N10 | 81 | 289 | 80 | 280 | 69 | 286 | 53 | 282 |
| N11 | 89 | 349 | 75 | 323 | 69 | 289 | | |
| S1 | 40 | 132 | 42 | 138 | 5 | 189 | 11 | 124 |
| S2 | 67 | 173 | 67 | 167 | 59 | 160 | | |
| S3 | 79 | 147 | 65 | 142 | 62 | 138 | 41 | 140 |
| S4 | 60 | 122 | 55 | 125 | 58 | 124 | 57 | 143 |
| S5 | 50 | 125 | 62 | 127 | 59 | 130 | | |
| S8 | 57 | 132 | 64 | 143 | 64 | 146 | 55 | 146 |
| S9 | 72 | 165 | 50 | 161 | 61 | 158 | | |
| S10 | 39 | 128 | 38 | 143 | 41 | 161 | 52 | 146 |
| S11 | 42 | 139 | 38 | 146 | 42 | 157 | 48 | 154 |
| S15 | 53 | 184 | 51 | 184 | 64 | 162 | 43 | 158 |
| R2 | 58 | 164 | 64 | 153 | 70 | 151 | 18 | 186 |

---

## Referee Report (RR1)

Considerable efforts were made by the authors to improve the paper taking into account the different comments.

This results in a high quality paper which is pleasant to read and is easier to follow than it was. Therefore, I only have a few specific comments before paper is accepted for publication.

1. L15: "point- and spatial-scale**s**"
2. L43-46: the content of this sentence is repeated below L105-110. Please, avoid repetitions in the introduction.
3. L47: wouldn't "variab**ly** inclined" be more accurate than "variable inclined"?
4. L130: "**is** shown" since it refers to the synopsis
5. L132: Remove "in order", "is essential for" is enough
6. L139: Add a comma after "To do this"
7. L151: I personally prefer "3D" rather than "3d", it is more consistent with common expression and with "Alpine3D"
8. L400: I may have missed something but I don't understand want 2.2 and 2.3°C are respectively referring to. The same is true for all the following expressions of this kind
9. L413: Please consider that a temperature is not "warm" or "cold", but "low" or "high": so this would be "higher" MAN
10. L530: is it "adequacy"? or "inadequacy"??
11. L661: "a" MAE

---

## Author Response (AR2)

**Authors' responses to the Editor and Reviewer (Ref. No.: tc-2016-73)**

The authors thank the Editor and the Reviewer for their final useful suggestions. We have made most of the requested changes to the manuscript, plus two other modifications and have listed all these and our explanations in the table below. Please also see the document below the table, with our modifications marked using *track changes*.

| Referee comment                                                                                                                                                                     | Authors' response                                                                                                                                                                                                                                                                                                                                                                                                                                   |
|-------------------------------------------------------------------------------------------------------------------------------------------------------------------------------------|-----------------------------------------------------------------------------------------------------------------------------------------------------------------------------------------------------------------------------------------------------------------------------------------------------------------------------------------------------------------------------------------------------------------------------------------------------|
| L. 15: 'point- and spatial-scales'                                                                                                                                                  | Done (throughout document)                                                                                                                                                                                                                                                                                                                                                                                                                          |
| L.43-46: the context of this sentence is repeated
below L.105-110. Please, avoid repetitions in the
introduction.                                                             | We have left lines 43-46 as they were and have
shortened the old lines 105-110 to 'This is
necessary, since rock temperatures were often
modelled without snow for idealized rock walls
>50° (e.g)' (new lines 106-108)
We feel that this short sentence should remain
as the reviewers both asked why the threshold
of >50° was chosen.                                                                                       |
| L47: wouldn't "variably inclined" be more accurate than "variable inclined"?                                                                                                        | Yes, changed to 'variably'.                                                                                                                                                                                                                                                                                                                                                                                                                         |
| L130: "is shown" since it refers to the synopsis                                                                                                                                    | Done.                                                                                                                                                                                                                                                                                                                                                                                                                                               |
| L132: Remove "in order", "is essential for" is enough                                                                                                                               | Done.                                                                                                                                                                                                                                                                                                                                                                                                                                               |
| L139: Add a comma after "To do this"                                                                                                                                                | Done.                                                                                                                                                                                                                                                                                                                                                                                                                                               |
| L151: I personally prefer "3D" rather than "3d",
it is more consistent with common expression
and with "Alpine3D"                                                             | Done throughout document (also for 1D and 2D).                                                                                                                                                                                                                                                                                                                                                                                                      |
| L400: I may have missed something but I don't
understand want 2.2 and 2.3°C are respectively
referring to. The same is true for all the
following expressions of this kind | This is explained in new lines 305-307: 'Note that
mean annual near-surface rock temperature
(MANSRT), r2, MAE and MBE are always given
for the study years 2012-2013/2013-2014,
separated by a slash (e.g. MANSRT for 2012-
2013/MANSRT for 2013-2014)
We added 'Note that' to alert the reader (306-
307).
We originally did this because Reviewer 2
remarked that we didn't always give the values
for both years. |
| L413: Please consider that a temperature is not
"warm" or "cold", but "low" or "high": so this
would be "higher" MAN                                                          | Done.                                                                                                                                                                                                                                                                                                                                                                                                                                               |
| L530: is it "adequacy"? or "inadequacy"??                                                                                                                                           | Thank you - we meant 'inadequacy'.                                                                                                                                                                                                                                                                                                                                                                                                                  |
| L661: "a" MAE                                                                                                                                                                       | Done                                                                                                                                                                                                                                                                                                                                                                                                                                                |
| Additional changes by the authors:                                                                                                                                                  | New L. 410: added 'was'                                                                                                                                                                                                                                                                                                                                                                                                                             |
|                                                                                                                                                                                     | New L. 243 The paper by Voegeli et al. has now been accepted, so we have adapted the reference here and in the reference list.                                                                                                                                                                                                                                                                                                                      |

[revised manuscript text omitted]